# A Comprehensive Survey on 3D Deep Point Cloud Models

## Abstract

Recently, point cloud data has attracted the attention of researchers as a promising data representation model for a wide range of applications. As unlike 2D data, point clouds are unordered, irregular, and often large in scale, they might impose severe challenges when designing deep learning models. Over the past decade, substantial progress has been made in proposing architectures that address permutation invariance, geometric reasoning, scalability, and robustness, leading to rapid expansion across diverse 3D data oriented applications. The main aims of this paper are to present a comprehensive survey on existing literature and to analyze how different 3D representations have shaped the design and performance of deep learning models. In contrast to prior surveys that have emphasized on limited task subsets or specific model families, this survey reviews deep point cloud models through representation- and architecture-centric perspective. As such, beyond (1) core tasks such as classification, segmentation, detection, tracking, this survey systematically provides insight into recent progress in broader directions, including (2) geometric modeling, alignment, and pose estimation, (3) foundation models and scene understanding, and (4) robustness, generalization, and reliability. Furthermore, this survey presents commonly used datasets and evaluation metrics, and finally summarizes challenges and future directions toward robustness, efficiency, and generalizability of 3D point cloud systems.

## 1 Introduction

In recent years, the use of 3D data has grown rapidly across various domains such as robotics, virtual reality, augmented reality (Wang et al., 2024h). The increasing availability of advanced sensors like LiDAR, RGB-D cameras, and multi-view stereo systems has made it possible to capture rich, detailed 3D information about real-world environments (Guo et al., 2020). Compared with 2D images, 3D data describes the real world in a more complete and natural way. A 2D image shows the scene on a flat surface, so information about depth and true spatial structure is lost. In contrast, 3D data keeps important geometric information such as the shape of objects, their size, and the distances between them. Because of this, 3D data is more suitable for tasks that need spatial understanding, such as recognizing objects, understanding scenes, reconstructing surfaces, and interacting with the physical world. In addition, 3D data is less affected by changes in lighting, camera angle, and surface texture, which often cause problems in image-based methods. For these reasons, 3D representations are a strong and natural choice for understanding real-world environments.

A variety of representations have been proposed to model 3D information, including voxel grids, polygonal meshes, multi-view projections and point clouds. Voxel-based approaches discretize space into regular 3D grids that are well suited to convolutional processing but often suffer from high memory consumption and limited resolution (Riegler et al., 2017). Meshes also come with challenges such as handling complex topology, managing high-resolution meshes with large memory and computation costs (Wang et al., 2024h). In contrast, a point cloud represents a scene or object as an unordered set of 3D points, each associated with attributes such as coordinates, color, intensity, or surface normals (Qi et al., 2017a). As a result, point clouds have become a preferred representation in many 3D tasks, including classification, detection, and segmentation (Qi et al., 2017a; Shi et al., 2019).

Despite these advantages, point clouds introduce unique challenges for data processing and learning. Unlike images or voxels, point clouds are inherently unordered, irregular, and unstructured, lacking a fixed grid or neighborhood definition (Bello et al., 2020). The density of points may vary significantly across regions, and large-scale point clouds can contain millions of points, posing serious computational and memory challenges. Additionally, real-world point cloud data is often noisy, incomplete, and dynamically acquired, further complicating robust feature extraction and learning (Quan et al., 2024). Designing models that are permutation-invariant, scalable, and capable of capturing both local geometric details and global context remains a central research problem.

Due to the rapid growth of research on 3D point cloud and its extensive use in a wide range of tasks and application domains, several survey papers have been published to review and categorize existing 3D data representations and methods from different perspectives. One of the early and influential works in the field, Guo et al. (2020) provided a foundational overview of deep learning approaches for point cloud processing, with a primary focus on fundamental tasks such as classification and segmentation. The survey categorized representative methods and highlighted the inherent challenges posed by the irregular structure and unordered nature of point cloud data.

Building upon earlier works, Lu & Shi (2020) expanded the scope of analysis beyond fundamental tasks by reviewing additional 3D point cloud understanding tasks such as flow estimation, registration, augmentation, and completion, alongside core tasks like classification, segmentation, detection, and tracking. With the increasing adoption of attention mechanisms and large-scale models, Lu et al. (2022) delivered the first dedicated and systematic review of transformer-based architectures for 3D point cloud analysis.

More recently, Vinodkumar et al. (2023) presented a focused review of deep learning approaches for 3D object recognition, covering state-of-the-art models for classification, segmentation, and detection across multiple 3D data formats and benchmark datasets. In 2024, Sarker et al. (2024) provided a focused and detailed review of deep learning methods specifically for 3D shape classification and semantic segmentation. This work analyzed recent progress in these two core tasks and discussed commonly used benchmark datasets and evaluation metrics.

Another contemporary contribution, Wang et al. (2024h), explored the adoption of diffusion-based generative models in 3D vision, examining how iterative denoising frameworks originally developed for 2D image generation can be adapted to 3D tasks such as object generation, shape completion, point cloud reconstruction, and scene synthesis.

Despite the breadth and depth of these surveys at their respective times of publication, several limitations persist. Prior works often emphasize specific subsets of tasks and do not systematically integrate recent architectural advances, such as transformer-based frameworks, large-scale pretraining, and foundation models across the full spectrum of 3D point cloud analysis (Guo et al., 2020; Lu & Shi, 2020). Importantly, key research areas including point cloud compression, reconstruction, robustness, and scene understanding evaluation are not jointly examined within a single survey. In contrast, this survey covers a broader and more diverse set of 3D point cloud tasks, and it provides dedicated discussions of benchmark datasets, standardized evaluation metrics, and comparisons of classification, segmentation, and detection tasks. Beyond core tasks such as classification, segmentation, detection, and tracking, this survey covers several broader directions in deep point cloud learning. These include geometric modeling, alignment, and pose estimation, encompassing generation, reconstruction, registration, and 6DoF pose estimation; foundation models and scene understanding; and robustness. The literature is further organized along two complementary axes—representation choices and architectural paradigms—and their interactions, strengths, and limitations are systematically analyzed. Together, these features provide a more comprehensive and up-to-date overview of the field, as summarized in Table 1.

This survey examines deep point cloud models through two central design dimensions: point cloud representations and architectural paradigms. A representation defines how 3D information is organized, while an architecture determines how that information is processed into task-specific features. By emphasizing their interaction, this survey shows how model behavior is shaped by representation-architecture combinations across different tasks. Following this perspective, we review core tasks, geometric modeling, spatial alignment and pose estimation, foundation models and scene understanding, and robustness.

Table 1: Comparison of existing comprehensive survey papers on deep learning for 3D point cloud analysis.

| Paper Name | Year | Application Tasks | Architecture & Representation Taxonomy | Datasets | Evaluation Metrics | Challenges & Open Problems |
|---|---|---|---|---|---|---|
| (Guo et al., 2020) | 2020 | Classification, Segmentation, Detection, Tracking | Point-Based (MLP-Based, CNN-Based, Graph-Based), Voxel-Based, Projection-Based, Hybrid | ✓ | ✓ | ✓ |
| (Lu & Shi, 2020) | 2020 | Classification, Segmentation, Detection, Tracking, Flow Estimation, Registration, Augmentation, Completion | Point-Based (MLP-Based, CNN-Based, Graph-Based), Voxel-Based, Projection-Based, Hybrid | ✓ | ✓ | ✗ |
| (Lu et al., 2022) | 2022 | Classification, Segmentation, Detection, Tracking, Registration, Video Understanding, Sampling, Denoising, Completion | Point-Based (Transformer-Based), Voxel-Based (Transformer-Based) | ✗ | ✗ | ✗ |
| (Vinodkumar et al., 2023) | 2023 | Classification, Segmentation, Detection | Point-Based (MLP-Based, CNN-Based, Graph-Based, Transformer-Based), Voxel-Based, Projection-Based, Hybrid | ✓ | ✗ | ✗ |
| (Sarker et al., 2024) | 2024 | Classification, Semantic Segmentation | Mesh-Based, Point-Based (MLP-Based, CNN-Based, Graph-Based, RNN-Based, Transformer-Based), Voxel-Based, Projection-Based, Hybrid | ✓ | ✓ | ✓ |
| (Wang et al., 2024h) | 2024 | Generation, Depth Estimation | Point-Based (Diffusion-Based) | ✓ | ✓ | ✓ |
| **This Survey** | **2026** | Classification, Segmentation, Detection, Tracking, **Compression**, **Generation**, **Reconstruction**, Registration, **6DoF Pose Estimation**, **Robustness**, **Scene Understanding** | Point-Based (MLP-Based, CNN-Based, Graph-Based, Transformer-Based, Diffusion-Based), Voxel-Based, Projection-Based, Hybrid | ✓ | ✓ | ✓ |

From the **task perspective**, this survey organizes deep point cloud learning into several task families rather than treating individual tasks as isolated problems. Core tasks include classification, segmentation, detection, and tracking, while geometric and spatial tasks include compression, generation, reconstruction, registration, and 6DoF pose estimation. Beyond these classical settings, particular emphasis is placed on robustness and scene understanding, which are increasingly important for real-world 3D applications. Specifically, robustness-oriented studies address 3D ML pipeline security, noise and occlusion resilience, rotation invariance and equivariance, domain adaptation, and anomaly detection, alongside scene-level reasoning tasks such as 3D question answering, 3D grounding, 3D captioning, and 3D reasoning. Across these task families, the survey emphasizes how representation choices and architectural paradigms influence model design, performance, and generalization.

From a **methodological standpoint**, this survey structures deep point cloud models along two complementary axes: representation choices and architectural paradigms. The representation axis includes point-based, voxel-based, projection-based, graph-based, token-based, and hybrid representations, while the architecture axis includes MLP-based, convolution-based, graph-based, transformer-based, diffusion-based, and multimodal-based architectures. This organization clarifies how different representation–architecture

combinations shape model behavior across tasks and provides a more systematic taxonomy than task-only or model-family-specific surveys.

In addition, this survey introduces a dedicated **evaluation section** that reviews commonly used benchmark datasets and evaluation metrics across point cloud tasks, with performance analysis focused on classification, segmentation, and detection. Since point cloud tasks differ in datasets, evaluation protocols, and metrics, numerical comparisons are reported only when methods are evaluated under comparable settings. Classification, segmentation, and detection are selected for performance analysis because they have widely used benchmark datasets and many methods report results with compatible metrics. For other tasks, where benchmarks and metrics are less standardized, the discussion focuses on datasets, evaluation criteria, methodological trends, and remaining challenges rather than direct numerical ranking.

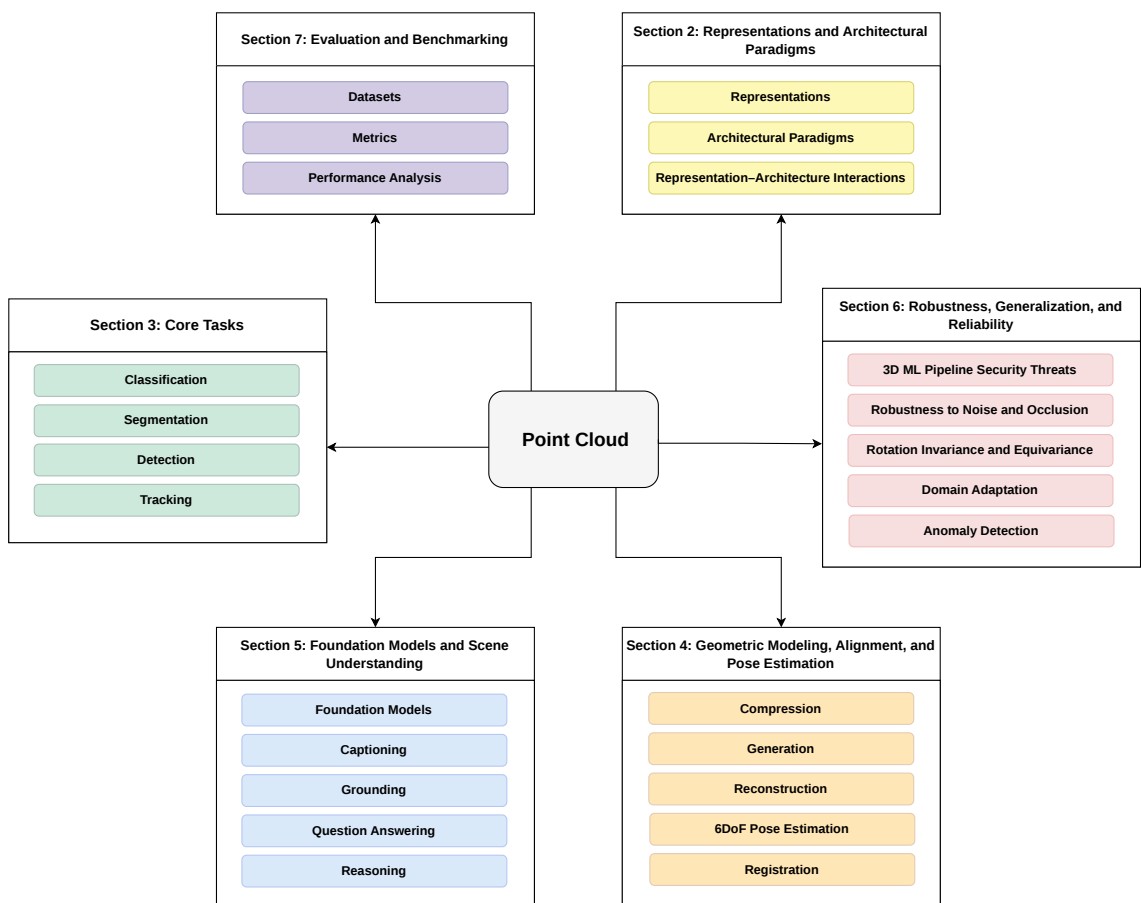

Figure 1: Overview of paper contents. This paper reviews 3D point cloud data across representations, architectural paradigms, and various tasks.

Figure 1 illustrates the overall organization of this survey. In summary, the main contributions of this work are as follows:

- A comprehensive survey of deep methods for 3D point clouds, covering a wide range of tasks, representations, and architectural paradigms within a unified framework.

- A representation-architecture taxonomy that connects point-based, voxel-based, projection-based, graph-based, token-based, and hybrid representations with major architectural families such as MLPs, CNNs, GNNs, transformers, diffusion models, and multimodal foundation models.

- A task-family structure that groups deep point cloud learning into core, geometric modeling, spatial alignment and pose estimation, foundation models and scene understanding, and robustness.

- A focused review of robustness and scene understanding in 3D point cloud learning, including studies on 3D ML pipeline security, noise and occlusion resilience, rotation invariance and equivariance, domain adaptation, anomaly detection, 3D question answering, grounding, captioning, and reasoning.

- A dedicated evaluation section that reviews commonly used datasets and metrics across point cloud tasks, with performance analysis focused on classification, segmentation, and detection where comparable benchmarks and compatible metrics are widely reported.

## 2 Point Cloud Representations and Architectural Paradigms

Despite the diverse tasks targeted by point cloud methods, they often share similar representations and architectural choices. Deep point cloud models are built upon two fundamental components: the representation used to encode 3D data and the architecture used to process it. This perspective is useful because the same 3D task can be approached through different representations, while a given representation can be paired with different neural architectures. This section reviews these common foundations and clarifies how different representation–architecture combinations shape model behavior across tasks.

### 2.1 Point Cloud Representations

A central design question in point cloud learning is how 3D data should be represented before being processed by a model. The chosen representation determines how geometric structure, point attributes, spatial relationships, and semantic cues are encoded, thereby influencing model expressiveness, computational cost, scalability, and performance across tasks.

#### 2.1.1 Point-Based Representations

Point-based representations operate directly on raw point sets, typically expressed as

$$\mathcal{P} = \{(x_i, y_i, z_i, f_i)\}_{i=1}^N,$$

where $(x_i, y_i, z_i)$ denotes the 3D coordinates of the $i$-th point, and $f_i$ represents optional point attributes such as color, intensity, surface normal, timestamp. Point-based representations preserve fine geometric details and avoid discretization artifacts introduced by voxelization or projection. However, because point clouds lack a canonical ordering, point-based methods require permutation-invariant or permutation-equivariant operations and often rely on sampling, grouping, or hierarchical aggregation to handle large-scale scenes. These representations are widely used various tasks with representative models including PointNet (Qi et al., 2017a), PointNet++ (Qi et al., 2017b), PointMLP (Ma et al., 2022), PointNeXt (Qian et al., 2022), and Point Transformer (Zhao et al., 2021a).

#### 2.1.2 Voxel-Based Representations

Voxel-based representations discretize continuous 3D space into a grid of volumetric cells, or voxels, allowing point clouds to be processed using 3D convolutional networks. By converting irregular point sets into a regular spatial structure, voxelization enables local feature extraction with established convolutional operators.

However, voxelization may introduce quantization errors and reduce fine geometric detail, while dense voxel grids often require substantial memory and computation. Sparse voxel representations mitigate these limitations by applying operations only to occupied or relevant grid regions, improving efficiency for sparse point cloud data. Representative voxel-based models include VoxNet (Maturana & Scherer, 2015), VoxelNet (Zhou & Tuzel, 2018), SECOND (Yan et al., 2018), and VoxelNeXt (Chen et al., 2023f).

### 2.1.3 Projection-Based and Multi-View Representations

Projection-based representations transform irregular 3D point clouds into structured 2D domains, allowing 2D CNNs and vision transformers to be applied for feature extraction. They are especially efficient for LiDAR range-view processing and related sensor setups, where projection naturally preserves much of the acquisition geometry. However, projection can introduce geometric distortion and information loss, especially under occlusion, viewpoint variation, and incomplete spatial coverage. Consequently, model performance is often influenced by the selected projection strategy, the choice of views, and the effectiveness of cross-view feature fusion.

### 2.1.4 Pillar and Bird's-Eye-View (BEV) Representations

Pillar and BEV representations have become central to 3D object detection. Points are organized into vertical columns (pillars) that are projected onto a 2D bird's-eye-view plane, enabling efficient 2D convolutional processing. This representation is particularly strong for capturing road-scene layout but loses fine vertical structure information. It is widely used in real-time detection and tracking systems. Representative models include PointPillars (Lang et al., 2019), BEVFusion (Liang et al., 2022).

### 2.1.5 Graph-Based Representations

Graph-based representations model a point cloud as a graph, where points are treated as nodes and edges encode spatial or geometric relationships between neighboring points. These neighborhoods are commonly constructed using k-nearest-neighbor graphs, radius-based graphs, or dynamic graphs that are updated during feature learning. By explicitly encoding local connectivity, graph-based representations support relational reasoning and capture fine-grained geometric structures that may be difficult to model with independent point-wise processing.

However, their effectiveness depends strongly on how neighborhoods are defined, and graph construction can introduce additional computational overhead. For large-scale point clouds, repeated neighborhood search and dynamic graph updates may also create scalability challenges. Representative graph-based models include DGCNN (Wang et al., 2019c).

### 2.1.6 Token-Based, and Foundation Model Representations

Token-based, and foundation-model representations encode point clouds into learned embeddings, tokens, or intermediate feature spaces rather than relying only on raw coordinates or explicit spatial grids. These representations have become increasingly important with the adoption of transformers and large-scale pretraining in 3D vision. Common strategies include grouping points into local patches, learning query or object tokens, applying masked point modeling for self-supervised pretraining, aligning point cloud features with language embeddings, and training large-scale 3D encoders for transferable representation learning.

Such representations support global context modeling, scalable pretraining, and semantic alignment with images, language, and other modalities. However, because information is encoded in latent spaces, these representations may reduce direct geometric interpretability and make it harder to trace predictions back to explicit point-level structures.

### 2.1.7 Hybrid Representations

Hybrid representations integrate two or more forms of point cloud encoding to combine their complementary advantages. Common examples include point–voxel fusion, point–BEV representations, LiDAR–camera integration, transformer tokenization of points or voxels. Such representations aim to preserve fine geometric detail while improving scalability, contextual modeling, and semantic expressiveness. However, because hybrid methods often involve multiple processing stages and heterogeneous inputs, they can be harder to analyze, reproduce, and compare fairly.

## 2.2 Architectural Paradigms

The architectural paradigm specifies how a point cloud representation is processed by a neural network. It determines how features are extracted, aggregated, and transformed across layers, and strongly influences a model's ability to capture local geometry, model global context, scale to large scenes, and adapt across tasks.

### 2.2.1 MLP-Based Architectures

MLP-based architectures use shared point-wise multilayer perceptrons to process each point independently, followed by symmetric pooling operations such as max pooling to aggregate global information. Hierarchical variants further introduce grouping and sampling strategies to capture local structures at multiple scales. Their main strengths are simplicity, permutation invariance, and computational efficiency for direct point processing. However, they have limited ability to model local geometry unless they are augmented with explicit grouping or neighborhood mechanisms. Representative models include PointNet (Qi et al., 2017a), PointNet++ (Qi et al., 2017b), and PointMLP (Ma et al., 2022).

### 2.2.2 Convolution-Based Architectures

Convolution-based architectures apply convolutional operators adapted to different representations, including voxel-based 3D convolutions, sparse 3D convolutions, point convolutions, and 2D CNNs on BEV or range-view images. These methods exploit the strong inductive bias of convolution to perform local feature extraction efficiently. Their main strengths are strong local feature learning and high efficiency when paired with structured representations such as voxels, BEV, or range images. However, their performance often depends on discretization quality or careful neighborhood design, and they may lose fine geometric detail in voxelized or projected forms. Representative models include VoxNet (Maturana & Scherer, 2015), KPConv (Thomas et al., 2019), PointCNN (Li et al., 2018), SECOND (Yan et al., 2018).

### 2.2.3 Graph Neural Network Architectures

Graph neural networks model point clouds as graphs and use message passing to propagate information between neighboring points. They incorporate edge features, support dynamic graph updates, and explicitly model relational structures. Their main strengths are effective capture of local geometry and relational patterns, as well as suitability for irregular and unordered data. However, they introduce graph construction overhead, have limited scalability in very large scenes, and can be sensitive to neighborhood definition and hyperparameter settings. Representative models include DGCNN (Wang et al., 2019c).

### 2.2.4 Transformer-Based Architectures

Transformer-based architectures apply self-attention over points, patches, voxels, or BEV tokens, enabling both local and global interactions within the same framework. They support masked modeling for self-supervised pretraining and are highly suitable for large-scale transfer learning. Their main strengths are the ability to capture long-range dependencies and global context effectively, flexibility across tasks and modalities, and strong suitability for foundation-model development. However, they also have notable weaknesses, including high computational cost, especially under dense attention, and the need for efficient tokenization, sparse attention, or hierarchical designs to scale well. Representative models include Point Transformer (Zhao et al., 2021a), PCT (Guo et al., 2021), PTv2 (Wu et al., 2022), and PTv3 (Wu et al., 2024e).

### 2.2.5 Diffusion-Based Architectures

Diffusion-based architectures model point clouds through iterative denoising, learning a probabilistic generative path from noise to structured 3D output. They are mainly used for generative tasks such as shape completion, reconstruction, and synthesis. Their key strengths are strong generative modeling capability and the ability to naturally handle uncertainty and multimodal outputs. However, they also have notable weaknesses, including slow inference due to iterative sampling, difficult evaluation and benchmarking.

Table 2: Common interactions between point cloud representations, architectural paradigms, and task usage.

| Representation | Common Architectures | Typical Tasks |
|---|---|---|
| Point-based | MLPs, GNNs, Point Transformers | Classification, Segmentation, Registration, 6DoF Pose Estimation |
| Voxel-based | 3D CNNs, Voxel Transformers | Detection, Segmentation, Reconstruction, Scene Understanding |
| Projection-based | 2D CNNs, Transformers | Segmentation, Detection, Tracking |
| Pillar/BEV | 2D CNNs, BEV Transformers, fusion networks | Detection, Tracking |
| Graph-based | GNNs, Graph Networks, Dynamic Graph Networks | Classification, Segmentation, Registration |
| Token/Foundation Model | Transformers, VLMs, LLMs | Segmentation, Grounding, Captioning, Question Answering, Reasoning |
| Hybrid | Fusion CNNs, Point Voxel Networks, Multimodal Transformers | Detection, Segmentation, Scene Understanding, 6DoF pose estimation |

### 2.2.6 Multimodal and Foundation Model Architectures

Multimodal and foundation-model architectures integrate 2D vision foundation models, vision-language models, large language models, and dedicated 3D vision-language models to align point clouds with language and other modalities. This enables reasoning-centric tasks and open-vocabulary understanding. Their main strengths include open-vocabulary 3D understanding, strong semantic reasoning and grounding capabilities, support for question answering, captioning, and reasoning tasks, and the benefits of large-scale pretraining and transfer. However, these models also have limitations, including heavy dependence on external priors from pretrained models, possible loss of fine geometric precision in favor of semantic alignment, and evaluation protocols and benchmarks that are still maturing.

### 2.3 Representation–Architecture Interactions

Representations and architectures are closely connected in deep point cloud learning. A representation defines how 3D information is organized, while the architecture determines how that information is processed and transformed into task-specific features. As a result, model performance is often shaped by the interaction between the two rather than by either component alone. For example, point-based representations preserve fine geometric details but often require efficient sampling and aggregation to scale to large scenes. Voxel- and BEV-based representations improve computational regularity and scalability, but may introduce quantization or projection artifacts. Graph-based methods explicitly capture local relationships, but neighborhood construction can be computationally expensive. Transformer-based and token-based approaches improve global context modeling, while foundation-model representations enhance semantic generalization, but both require careful design to preserve geometric fidelity and support reliable evaluation. Table 2 summarizes common representation–architecture pairings and their typical task usage.

## 3 Core Tasks

Core tasks refer to the fundamental problems through which point cloud models identify, localize, and organize objects or regions in 3D data. In this section, four core tasks are reviewed: classification (Section 3.1), segmentation (Section 3.2), detection (Section 3.3), and tracking (Section 3.4). Classification assigns a global label to an object or scene, segmentation predicts point-level or region-level labels, detection localizes objects with 3D bounding boxes, and tracking extends detection over time by associating objects across consecutive

frames. These tasks are grouped together because they form the basic pipeline for understanding static and dynamic 3D scenes.

## 3.1 Classification

This section presents a structured taxonomy of 3D point cloud classification approaches and reviews several representative recent studies.

Most methods follow a two-stage pipeline: first, learning local geometric embeddings and then aggregating them into a global shape embedding, which is finally fed into a classifier to produce the final label. Depending on the type of 3D data representation, existing deep learning based classification methods can be broadly divided into three categories: projection-based or multi-view methods; volumetric or voxel-based methods; and point-based methods. While this survey focuses primarily on point-based methods due to their effectiveness, flexibility, and growing popularity, representative projection-based and voxel-based methods are also included to provide a complete picture of the design space and trade-offs.

### 3.1.1 Projection-based Methods

Projection-based methods have been widely adopted for 3D point cloud classification by rendering multiple 2D views and leveraging mature 2D convolutional networks for feature extraction. The primary challenge lies in effectively fusing multi-view descriptors into a compact global representation that preserves the overall 3D structure. Early works addressed this fusion problem with different strategies. MVCNN (Su et al., 2015) aggregated view features using max-pooling, though this approach often discarded complementary information. GIFT (Bai et al., 2016) accelerated the multi-view framework through GPU-based projection and hashing. GVCNN (Feng et al., 2018) enhanced representation capability by grouping views based on discrimination scores and modeling intra- and inter-group relationships. MHBN (Yu et al., 2018b) employed harmonized bilinear pooling to capture multiplicative feature interactions, while MVACPN (Wang et al., 2022c) incorporated attention mechanisms with convolutional pooling to better preserve informative viewpoints during fusion.

Subsequent research introduced adaptive viewpoint selection, stronger fusion mechanisms, and integration with other 3D representations. MVTN (Hamdi et al., 2021) replaced fixed viewpoints with learnable camera parameters via differentiable rendering. CLIP2Point (Huang et al., 2023a) transferred image-based knowledge into 3D understanding by projecting point clouds into depth maps and leveraging CLIP pre-training. Later works expanded projection modalities and multimodal learning, including DTV-CNN (Xia, 2023), which fused depth and thickness images; PointOfView (Ren et al., 2024a), which combined multi-view projections with raw point features for few-shot learning; and self-supervised approaches such as Multi-View Representation Is What You Need (Yan et al., 2023a).

### 3.1.2 Volumetric-based Methods

Voxel-based methods convert point clouds into regular 3D voxel grids, enabling geometric feature learning through 3D CNNs. Early works such as VoxNet (Maturana & Scherer, 2015) applied 3D CNNs to occupancy grids for object recognition. To alleviate the high computational cost of dense 3D convolutions, FPNN (Li et al., 2016) introduced a field-probing mechanism that samples continuous 3D fields with learned probes.

To further improve efficiency, later methods adopted hierarchical and sparse representations. OctNet (Riegler et al., 2017) utilized unbalanced octrees for sparse voxel storage, O-CNN (Wang et al., 2017) restricted convolution to surface-adjacent octants, Kd-Network (Klokov & Lempitsky, 2017) replaced voxel grids with Kd-tree partitions, and MSNet (Wang et al., 2018b) leveraged multi-scale voxelization with CRF-based optimization to better capture structural details.

Subsequent research enhanced voxel encoding and robustness. VV-Net (Meng et al., 2019) incorporated variational autoencoder-based local embeddings to address sparse and noisy data, while MRCNN (Ghadai et al., 2019) proposed multi-level voxelization. $(AF)^2$-S3Net (Cheng et al., 2021) introduced attentive feature fusion and adaptive feature selection within 3D CNNs to improve semantic classification accuracy.

In recent years, purely voxel-based classification methods have become less common, with many studies shifting toward hybrid approaches. PV-Ada (Zhu et al., 2022) combines voxel and point level encodings with transformers and adaptive pooling for improved robustness. Overall, dense voxel-only pipelines have largely been replaced by hybrid, sparse, or structured representations in modern classification research.

### 3.1.3 Point-based Methods

Point-based methods operate directly on raw 3D point clouds, representing shapes as unordered sets of points in Euclidean space. Unlike voxel-based or projection-based approaches, they avoid discretization artifacts while preserving precise geometric information. Early work showed that deep networks can learn from point sets using permutation-invariant operations. Since then, point-based methods have developed into MLP-based, convolution-based, graph-based, and transformer-based architectures, mainly differing in how they model neighborhood relationships and aggregate features. The following sections discuss each category in more detail.

**MLP-based Methods**  Point-wise MLPs first became viable for 3D point cloud learning with PointNet (Qi et al., 2017a), which solved permutation invariance using a shared per-point MLP and global max pooling, supported later by the DeepSets (Zaheer et al., 2017) framework. It's T-Net alignment further improved invariance, but the model was limited by it's inability to capture local geometry. PointNet++ (Qi et al., 2017b) addressed this by introducing a hierarchical set abstraction module using FPS sampling, ball query grouping, and mini-PointNet blocks to extract multi-scale local features, establishing the foundational architecture for many later methods.

This family inspired numerous extensions aimed at improving feature quality. Examples include Mo-Net (Joseph-Rivlin et al., 2019), which fed geometric moments into a PointNet backbone; PointWeb (Zhao et al., 2019), which enhanced local neighborhoods using adaptive feature adjustment; and SRN (Duan et al., 2019b), which explicitly modeled structural dependencies through MLPs.

Recently, the field has seen a renewed focus on efficient point-wise MLP architectures, largely driven by PointNeXt (Qian et al., 2022) and PointMLP (Ma et al., 2022). PointNeXt demonstrated that the Point-Net++ hierarchy could reach state-of-the-art accuracy when paired with modern training strategies, while PointMLP introduced a pure residual MLP design supplemented by a simple geometric affine module showing that MLPs alone can outperform complex graph or transformer models while remaining fast and hardware-friendly.

Building on this resurgence, newer works refine MLP-based local feature extraction. DualMLP (Paul et al., 2024) employs parallel dense and sparse MLP streams, PointGL (Li et al., 2024a) fuses global MLP features and local graph pooling, and high-dimensional positional encoding (Zou et al., 2024b) improves MLP representational power by enriching raw coordinates. At the frontier, POINTMIL (De Vries et al., 2025) integrates Multiple Instance Learning into MLP backbones for better interpretability, while Point-KAN (Shi et al., 2025) replaces traditional MLP layers with Kolmogorov–Arnold Networks, marking the next step in efficient and expressive point-wise architectures.

**Convolution-based Methods**  Convolution-based point cloud methods aim to extend CNNs to irregular 3D point sets, where defining kernels is more difficult than on regular 2D grids. These methods are commonly divided into continuous and discrete convolution approaches. Continuous methods define kernels in continuous space and aggregate local features based on relative geometry, as seen in PCNN (Atzmon et al., 2018), SPHNet (Poulenard et al., 2019), RS-CNN (Liu et al., 2019d), KPConv (Thomas et al., 2019), DensePoint (Liu et al., 2019c), and ConvPoint (Boulch, 2020), with techniques such as radial basis functions, spherical harmonics, learnable kernel points, and Monte Carlo estimation to handle varying point densities. Discrete methods impose local structure by partitioning neighborhoods into bins or grids, including PointCNN (Li et al., 2018), A-CNN (Komarichev et al., 2019), RIConv (Zhang et al., 2019a), ShellNet (Zhang et al., 2019b), and PointWiseCNN (Hua et al., 2018), as well as geometry-enhanced operators like DeltaConv (Wiersma et al., 2022).

Fewer purely CNN-based point methods were introduced compared to MLP and transformer based models. Recent works such as CompositeNets (Floris et al., 2024), RepSurf-based convolution (Ran et al., 2022), and PointCNN++ (Li et al., 2025d) refine earlier point convolution designs for better efficiency and geometric modeling. DC-CCNN (Dang et al., 2026) introduces a primary-visual-cortex-inspired framework that combines brain-inspired spiking/continuous-coupled neural computation with convolutional feature extraction to improve efficiency and robustness for point cloud classification.

**Graph-based Methods**   Graph-based point cloud methods model a point cloud as a graph, where points are vertices and edges encode geometric relationships, enabling explicit local and global feature learning. Early works such as ECC (Simonovsky & Komodakis, 2017) and SpecGCN (Wang et al., 2018a) applied edge-conditioned and spectral graph convolutions, while RGCNN (Te et al., 2018), PointGCN (Zhang & Rabbat, 2018) leveraged Laplacian-based filters. Spatial domain graph models like DGCNN (Wang et al., 2019c) introduced the influential EdgeConv module, followed by LDGCNN (Zhang et al., 2021a), which streamlined hierarchical graph linkage. Variants such as PointNGCNN (Lu et al., 2020), KCNet (Shen et al., 2018), and 3D-GCN (Lin et al., 2020) capture neighborhood structure through graph filters and kernel correlations. Extensions including PointView-GCN (Mohammadi et al., 2021) and PointManifold (Yang & Gao, 2020) enhances feature continuity and manifold structure in learned representations, while adaptive graph convolutions such as PAConv (Xu et al., 2021b) dynamically generate kernels from point positions. CurveNet (Xiang et al., 2021a) further augments relational modeling by aggregating sequences of point segments.

Recent years have seen new graph-based classifiers that emphasize efficiency and scalability. PointViG (Zheng et al., 2024b) introduces a lightweight GNN with adaptive dilated neighborhood graph convolutions to expand receptive fields and improve efficiency, achieving high accuracy with a compact parameter budget. Point-SkipNet (Saeid et al., 2025) explores lightweight graph architectures tailored for efficient classification on refined datasets.

**Transformer-based Methods**   Since the introduction of the Transformer architecture by (Vaswani et al., 2017), their ability to model long range dependencies has strongly influenced point cloud classification, where self-attention provides a natural way to relate unordered points. Early point cloud Transformers like Point Transformer (Zhao et al., 2021a), PCT (Guo et al., 2021), PAT (Yang et al., 2019b), PTv2 (Wu et al., 2022), LCPFormer (Huang et al., 2023b), and SPoTR (Park et al., 2023a) adapted the transformer idea by building permutation-invariant attention blocks and improving geometric encoding through local attention, offset/relative modeling, or structured sparsity, while general vision advances such as ViT (Dosovitskiy, 2020) and DETR (Carion et al., 2020) helped popularize tokenization and set-based prediction designs. In the following years, supervised point cloud transformers increasingly emphasized efficiency, hierarchy, and scalable context modeling: PTv3 (Wu et al., 2024e) simplified the backbone to scale receptive fields and speed. In recent years, patch-based and domain-focused designs expanded the toolbox, including PPT (Wang et al., 2025i) for explicit point patch tokenization, SPT (Wu et al., 2025a) for spiking transformer classification, and task-specific transformers like PTMF (Pan et al., 2025a) reflecting a clear trend toward making point cloud transformers more scalable, efficient, and application-ready.

### 3.1.4   Conclusion, Challenges, and Future Directions of Classification

In summary, 3D point cloud classification has evolved from projection- and voxel-based approaches toward direct point-based models, including MLP-based, convolution-based, graph-based, and transformer-based architectures. Projection-based methods can reuse mature 2D backbones but depend on view selection and may lose geometric information. Voxel-based methods provide structured representations for 3D convolutions but suffer from quantization and computational cost. Point-based methods preserve raw geometry and have become dominant, while graph and transformer models further improve local relationship modeling and global context reasoning.

Despite strong progress, several challenges remain. Many methods perform well on clean synthetic datasets but are less reliable on real-world scans with noise, occlusion, clutter, density variation, and partial observations. Efficiently capturing both local geometry and global shape structure is still difficult, especially

for sparse or large-scale point clouds. In addition, graph- and transformer-based models often improve representation power at the cost of higher computation.

Future research should focus on classification methods that are robust, efficient, and generalizable across real-world scenarios. Important directions include lightweight architectures, better local-global feature aggregation, improved robustness to noisy and incomplete data, self-supervised or weakly supervised learning, and stronger domain generalization from synthetic to real-world datasets. More realistic benchmarks and standardized evaluation protocols are also needed to better measure practical classification performance.

## 3.2 Segmentation

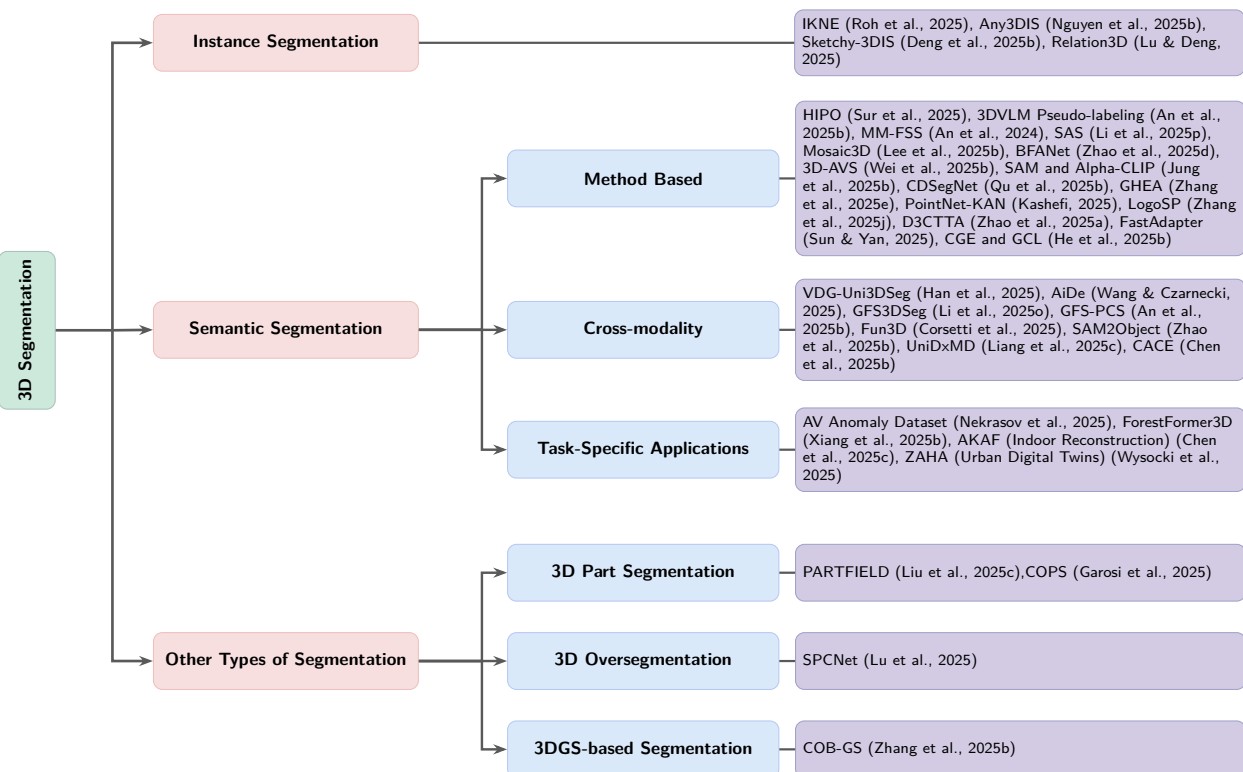

Figure 2: Taxonomy of 3D segmentation methods.

3D point cloud segmentation aims to partition a point cloud into meaningful subsets based on geometric properties or semantic content. By assigning semantic labels or grouping points according to their spatial relationships and contextual information, segmentation serves as a fundamental component of higher-level scene understanding and supports a broad range of downstream applications, including scene reconstruction and object recognition. While earlier surveys (Vinodkumar et al., 2023; Sarker et al., 2024) mainly focused on the historical development of 3D point cloud segmentation methods, this survey emphasizes recent advances in the field. In recent years, research has increasingly explored new directions, such as utilizing vision-language and large language models to enhance semantic representation (Han et al., 2025), as well as virtual camera-based formulations aimed at improving feature learning in 3D segmentation tasks (Zhang et al., 2025a). Figure 2 summarizes the main categories of recent 3D point cloud segmentation methods.

### 3.2.1 Instance Segmentation

Instance segmentation extends 3D point cloud segmentation by partitioning points into distinct object instances while preserving geometric coherence. This requires reasoning about object-level structures in addition to local point features, which is particularly challenging in complex scenes due to occlusions, clutter, and partial observations.

Recent advances address these challenges by incorporating higher-level reasoning and enhanced relation modeling. Object-centric approaches explicitly encode instance-level features to capture holistic geometric and structural properties, improving instance grouping and boundary consistency (Roh et al., 2025). Multi-view strategies leverage 2D segmentation models and enforce cross-view instance consistency through mask tracking, thereby reducing fragmented 3D proposals (Nguyen et al., 2025b). Weakly supervised methods show that sparse annotations, such as sketch-based points and 3D bounding boxes can guide accurate instance segmentation while lowering labeling costs (Deng et al., 2025b). More recently, relation modeling frameworks have enhanced 3D instance segmentation by explicitly encoding geometric and spatial relationships between scene points and instance queries, using adaptive superpoint aggregation, contrastive refinement, and relation-aware self-attention to improve instance discrimination and consistency (Lu & Deng, 2025).

### 3.2.2 Semantic Segmentation

This section categorizes 3D point cloud semantic segmentation methods into three main directions: method-based approaches, cross-modal approaches, and task-specific applications.

**Method-Based**  Several studies in semantic segmentation have proposed novel methods to address key challenges in the field; this section provides an overview of these contributions.

Several recent works have tackled the problem of few-shot learning. Sur et al. (2025) argues that the Euclidean space commonly used in the field is not optimal, as it provides insufficient embedding capacity, causing models to forget previously learned knowledge when adapting to new classes. To address this issue, they propose the use of hyperbolic spaces, whose volume expands exponentially.

In An et al. (2025b), the authors argue that, in many cases, a few samples cannot provide sufficient information to capture the complex features of each class. They propose using 3D VLMs as the backbone to leverage their extensive open-world knowledge for generating dense pseudo-labels, thereby compensating for the sparsity of few-shot samples. Similarly, An et al. (2024) proposes a multimodal framework that utilizes textual labels and 2D images. Crucially, the paper uses 2D images only during pretraining to align 3D features, allowing the model to operate without explicit image inputs during inference.

Several works address the problem of undefined classes, particularly in open-vocabulary settings. Li et al. (2025p); Yen et al. (2026) aim to train and use 3D models by distilling knowledge from 2D models into 3D representations. By combining the outputs of 2D models, SAS (Li et al., 2025p) reduces the inherited limitations of the prior models. To reduce computational cost and latency, Lee et al. (2025b) introduces Mosaic3D , which is based on inherently 3D backbones. After applying SparseUNet, the CLIP-aligned point embeddings are compared with the input text embeddings from Recap-CLIP. Zhao et al. (2025d) moves away from CLIP to capture features beyond the object level. By creating a 3D object masks, the paper selects the best viewpoint for each object and uses context-aware representations together with a Multimodal Large Language Model (MLLM) to perform segmentation based on visual and environmental context. Wei et al. (2025b) uses an off-the-shelf image captioner and introduce Sparse Masked Attention Pooling (SMAP) for point captioning to improve open-vocabulary segmentation. The generated captions are fed into a Caption2Tag module to create point-level tags, which are then used to infer semantic segmentation. Jung et al. (2025b) uses a double pass of the Segment Anything Model (SAM) to mask objects and Alpha-CLIP to obtain accurate embeddings. These embeddings are then cross-referenced with the input text. However, their study is focuses on 3DIS problems.

To improve latency, Qu et al. (2025b) moves beyond the conventional multi-step process of diffusion models and propose CDSegNet, a model based on the Conditional-Noise Framework (CNF). This model generates segmentation mask in a single step and improves robustness. To mitigate the challenge of data scarcity, Zhang et al. (2025e) proposes GHEA, a framework for generating new samples. They use generative models to produce new samples from existing data and then weight and mix the samples according to their hardness.

To explore the effect of Kolmogorov-Arnold Networks (KANs), Kashefi (2025) proposes PointNet-KAN, which replaces the shared MLPs of the original PointNet with shared KAN layers constructed using Jacobi polynomials. The study also includes PointNet-KAN-MLP which combines a KAN encoder with an MLP

decoder to mitigate overfitting in segmentation tasks, and PointNet-KAN-MLP++, which integrates KANs into the set abstraction modules of PointNet++ while retaining MLPs for the classification head.

Zhang et al. (2025j) proposes LogoSP, a method for unsupervised 3D semantic segmentation that distills self-supervised 2D foundation models into 3D point features. The points are then progressively grouped into larger superpoints, which are organized into a graph from which global patterns are extracted to assign pseudo-labels.

To address continual test-time adaptation for 3D LiDAR, Zhao et al. (2025a) proposes D3CTTA, a backpropagation-free framework that manages density variations and feature correlations. Its novelty lies in combining distance-aware prototype learning with domain-specific feature decorrelation to mitigate category confusion across changing domains without gradient updates.

Sun & Yan (2025) introduces FastAdapter, a general module designed to mitigate the geometric information degradation typically caused by fast downsampling methods. To achieve this, the paper uses two components, Point-to-Anchor (P2A) and Anchor-to-Point (A2P), which create anchor points and inject them into the downsampled points.

To tackle the problem of domain generalization, He et al. (2025b) proposes Category-level Geometry Embedding (CGE) and Geometric Consistent Learning (GCL). Unlike global approaches, CGE maps features into a geometric space specifically Wasserstein space to capture specific structural properties unique to each semantic category. Furthermore, GCL attempts to inject physics-inspired awareness into the model.

**Cross-Modality**   Recent advances in 3D semantic segmentation increasingly adopt cross-modal learning, integrating language, images, and geometry, to overcome limited annotations, domain gaps, and closed-set semantics, thereby enabling more scalable and generalizable scene understanding. Vision language priors drive much of this progress: VDG-Uni3DSeg (Han et al., 2025) and AiDe (Wang & Czarnecki, 2025) align 2D VLM features with 3D geometry for open-vocabulary recognition, while GFS3DSeg (Li et al., 2025o) and GFS-PCS (An et al., 2025b) extend these ideas to generalized few-shot segmentation using language-driven relational cues. Beyond semantics, Fun3D (Corsetti et al., 2025) shows that natural language actions can guide functional 3D understanding.

For instance-level generalization, SAM2Object (Zhao et al., 2025b) consolidates multi view SAM2 (Ravi et al., 2025) masks for zero-shot 3D instance segmentation. Cross-modal learning also enhances robustness: UniDxMD (Liang et al., 2025c) unifies 2D–3D representations for unsupervised domain adaptation, while CACE (Chen et al., 2025b) improves sim-to-real transfer through context aware augmentation and 2D–3D consistency.

**Task-Specific Applications**   Beyond general segmentation, several works focus on task-specific domains such as forestry, and indoor reconstruction. To address the challenge of out-of-distribution objects, (Nekrasov et al., 2025) introduces a high-resolution 3D LiDAR dataset for anomaly detection, benchmarking five uncertainty-based baselines adapted from 2D segmentation.

In environmental monitoring, (Xiang et al., 2025b) proposes ForestFormer3D, an end-to-end framework for semantic instance segmentation of forest LiDAR data. By utilizing instance-aware query selection in a 5D embedding space and a block-merging strategy for large-scale inference, the model effectively separates intertwined tree canopies in complex forest environments.

For indoor reconstruction, (Chen et al., 2025c) improves structural component identification such as beams and columns, by introducing Architectural-Knowledge-Aware features (AKAFs). Their two-stage framework fuses explicit global coordinates with relative semantic positions, consistently improving the accuracy of various backbones.

Finally, in the context of urban digital twins, (Wysocki et al., 2025) tackles the heterogeneity of building facades. They introduce the ZAHA dataset, the largest of its kind, and the Level of Facade Generalization (LoFG) taxonomy based on CityGML standards.

### 3.2.3 Other Types of Segmentation

Beyond semantic and instance segmentation, other types of 3D segmentation focus on different structural levels. 3D part segmentation aims to divide a single object into meaningful components such as legs, handles, or wings, enabling fine-grained shape understanding. Recent works explore different approaches to this problem: PARTFIELD (Liu et al., 2025c) learns a continuous 3D feature field that groups points into consistent parts directly in 3D space, while COPS (Garosi et al., 2025) uses multi-view 2D features and projects them into 3D to perform part segmentation without 3D training data. Another related task is 3D oversegmentation, which divides a point cloud into many small, coherent regions to simplify later processing; SPCNet (Lu et al., 2025) improves this step by ordering points along a space-filling curve to preserve spatial locality and efficiently form high-quality superpoints. In scene representation, 3DGS-based segmentation methods, such as COB-GS (Zhang et al., 2025b) focus on improving object boundaries by adaptively splitting Gaussians near edges, producing cleaner and sharper segmentation results.

### 3.2.4 Conclusion, Challenges, and Future Directions of Segmentation

Recent advancements in 3D point cloud segmentation mark a significant shift toward robust, multi-modal frameworks, yet quantitative benchmarks reveal an ongoing trade-off between open-world versatility and raw accuracy. While fully supervised and augmented methods like Mosaic3D (Lee et al., 2025b) and GHEA (Zhang et al., 2025e) achieve state-of-the-art performance, zero-shot and open-vocabulary approaches such as SAM2Object (Zhao et al., 2025b) and Any3DIS (Nguyen et al., 2025b) currently exhibit lower accuracy. This difference highlights the primary challenge facing the field: overcoming the prohibitive costs of dense 3D annotations without sacrificing high-fidelity scene understanding. Moving forward, future research should focus on bridging this performance gap by refining self-supervised and cross-modal architectures that minimize reliance on explicit 2D or textual priors during inference, ultimately driving toward low-latency, accurate segmentation across complex, real-world environments.

## 3.3 Detection

3D object detection estimates the 3D pose and extent of objects by predicting oriented 3D bounding boxes and class labels from point clouds such as LiDAR or RGB-D data, sometimes in combination with images (Mousavian et al., 2017; Shi et al., 2019; Yin et al., 2021). The task is challenging because point clouds are irregular and sparse, have range-dependent density, and suffer from occlusions and partial observations, making discriminative feature learning and accurate localization difficult (Qi et al., 2019; Shi et al., 2020a).

Because the number of proposed 3D detection methods is now very large, a single linear or purely chronological review is difficult to navigate. We therefore adopt a multi-axis taxonomy that reflects the most common design decisions and the trade-offs they introduce, helping readers compare methods within comparable settings and quickly identify approaches suited to their constraints. Specifically, we categorize detectors along six complementary criteria: **data representation** (point, voxel, pillar, projection, hybrid), **detection stage** (one-stage, two-stage), **anchor strategy** (anchor-based, anchor-free), **backbone type** (CNN, transformer, GNN, MLP-based), **scene type** (indoor, outdoor), and **supervision type** (fully, weakly, semi-, self-supervised).

Table 3 presents a comprehensive taxonomy of 3D detection methods organized according to these six complementary criteria. Because these axes overlap, many papers fit multiple categories. We therefore use the taxonomy as an overlapping tagging scheme rather than a strict partition, which better reflects the modular nature of 3D detectors and keeps the review easy to navigate while we discuss each criterion in its own subsection.

### 3.3.1 Data Representation

A core design choice in 3D object detection is the data representation used to convert sparse, irregular point clouds into learnable features, which directly affects geometric fidelity, receptive field, and runtime (Shi et al., 2019; Zhou & Tuzel, 2018; Lang et al., 2019). This section groups detectors into five common representations: **point-based** methods operate on raw (or RoI-sampled) points to preserve fine geometry, at

Table 3: Summary of 3D detection methods.

| Axis | Subcategory | Methods |
|------|-------------|---------|
| Data Representation | Point | PointRCNN (Shi et al., 2019), VoteNet (Qi et al., 2019), 3DSSD (Yang et al., 2020b), Point-GNN (Shi & Rajkumar, 2020a), 3DETR (Misra et al., 2021), Group-free 3D (Liu et al., 2021b), IA-SSD (Zhang et al., 2022c), Clusterformer (Pei et al., 2023), DetZero (Ma et al., 2023), OV-3DET (Lu et al., 2023), PTT (Huang et al., 2024b), FASTer (Dang et al., 2025), GeoFormer (Jin et al., 2025b), Part-A$^2$ net (Shi et al., 2020b), PPC (Goyal et al., 2025) |
| | Voxel | VoxelNet (Zhou & Tuzel, 2018), SECOND (Yan et al., 2018), Voxel R-CNN (Deng et al., 2021), VoTr (Mao et al., 2021b), FocalSparseConv (Chen et al., 2022e), SST (Fan et al., 2022b), DSVT (Wang et al., 2023a), VoxelNeXt (Chen et al., 2023f), SAFDNet (Zhang et al., 2024c), UniMamba (Jin et al., 2025a), ViKIENet (Yu et al., 2025b), FSHNet (Liu et al., 2025e) |
| | Pillar | PointPillars (Lang et al., 2019), PillarNet (Shi et al., 2022a), PillarNeXt (Li et al., 2023c), PillarHist (Zhou et al., 2025f), 3DPillars (Noh et al., 2025) |
| | Projection | MV3D (Chen et al., 2017), PIXOR (Yang et al., 2018a), AVOD (Ku et al., 2018), LaserNet (Meyer et al., 2019), RangeDet (Fan et al., 2021), BEVFormer (Li et al., 2024j), BEVDepth (Li et al., 2023j), SparseBEV (Liu et al., 2023b), BEVNeXt (Li et al., 2024i), EVT (Lee et al., 2025d), OpenM3D (Hsu et al., 2025) |
| | Hybrid | PV-RCNN (Shi et al., 2020a), PV-RCNN++ (Shi et al., 2023a), PointAugmenting (Wang et al., 2021a), VISTA (Deng et al., 2022), CVFNet (Gu et al., 2022), TransFusion (Bai et al., 2022), Bridged Transformer (Wang et al., 2022d), FUTR3D (Chen et al., 2023c), GAFusion (Li et al., 2024e), RCBEVDet (Lin et al., 2024d), LiDAR–4D Radar Fusion (Chae et al., 2024), LiRaFusion (Song et al., 2024), Ev-3DOD (Cho et al., 2025), V2X-R (Huang et al., 2025f) |
| Detection Stage | One Stage | SA-SSD (He et al., 2020a), SST (Fan et al., 2022b), BEVFormer (Li et al., 2022d), DSVT (Wang et al., 2023a), SparseBEV (Liu et al., 2023b), BEVNeXt (Li et al., 2024i) |
| | Two Stage | PV-RCNN (Shi et al., 2020a), PV-RCNN++ (Shi et al., 2023a), Voxel R-CNN (Deng et al., 2021), LiDAR R-CNN (Li et al., 2021e), Pyramid R-CNN (Mao et al., 2021a), CT3D (Sheng et al., 2021), DiffRefine (Shin et al., 2025) |
| Anchor Strategy | Anchor-based | VoxelNet (Zhou & Tuzel, 2018), PointPillars (Lang et al., 2019), SECOND (Yan et al., 2018), PV-RCNN (Shi et al., 2020a) |
| | Anchor-free | PointRCNN (Shi et al., 2019), CenterPoint (Yin et al., 2021), VRVP (Deng et al., 2023), ONE (Wang et al., 2024i) |
| Backbone Type | CNN | VoxelNeXt (Chen et al., 2023f), Rethinking Backbone Design for Lightweight 3D Object Detection in LiDAR (Chandorkar et al., 2025), Mixed Precision PointPillars for Efficient 3D Object Detection with TensorRT (Fuengfusin et al., 2026) |
| | Transformer | Voxel Transformer (VoTr) (Mao et al., 2021b), SST (Fan et al., 2022b), DSVT (Wang et al., 2023a), LitePT (Yue et al., 2025), GeoFormer (Jin et al., 2025b) |
| | GNN | Point-GNN (Shi & Rajkumar, 2020b), PC-RGNN (Zhang et al., 2021c), VoxT-GNN (Zheng et al., 2025) |
| Scene Type | Indoor | ImVoteNet (Qi et al., 2020a), H3DNet (Zhang et al., 2020b), 3DETR (Misra et al., 2021), Group-free 3D (Liu et al., 2021b), V-DETR (Shen et al., 2023), FCAF3D (Rukhovich et al., 2022) |
| | Outdoor | CIA-SSD (Zheng et al., 2021b), MPPNet (Chen et al., 2022d), FSD (Fan et al., 2022c), CenterFormer (Zhou et al., 2022c), PointPainting (Vora et al., 2020), 3D-CVF (Yoo et al., 2020), BEVFusion (Liang et al., 2022), SparseLIF (Zhang et al., 2024e), MambaFusion (Wang et al., 2025e) |
| Supervision | Weakly-supervised | WS3D (Meng et al., 2020), VS3D (Qin et al., 2020), (Meng et al., 2021), Back to Reality (Xu et al., 2022), ViT-WSS3D(Zhang et al., 2023b), VG-W3D (Huang et al., 2024c), Prompt3D (Zhang et al., 2024k), (Han et al., 2024) |
| | Semi-supervised | Sess (Zhao et al., 2020a), 3dioumatch (Wang et al., 2021b), ProficientTeachers (Yin et al., 2022a), NoiseDet (Chen et al., 2023g), Ssda3d (Wang et al., 2023f), PTPM (Wu et al., 2024d), DPL (Zhang et al., 2024f), Sp3d (Zhao et al., 2025c), PieAug (Lee et al., 2025a), GA-BEVFusion (Hazra et al., 2025), (Park et al., 2025) |
| | Self-/Unsupervised | Pointcontrast (Xie et al., 2020), (Liang et al., 2021), Randomrooms (Rao et al., 2021), GCC-3D (Erçelik et al., 2022), Proposalcontras (Yin et al., 2022b), Union (Lentsch et al., 2024), Lise (Zhang et al., 2024j), Liso (Baur et al., 2024), Patchcontrast (Shrout et al., 2025), Cmae-3d (Zhang et al., 2025g), DOtA (Xia et al., 2025a) |

the cost of irregular computation and expensive neighborhood aggregation (Shi et al., 2019; Qi et al., 2019; Shi et al., 2020b; Yang et al., 2020b; Shi & Rajkumar, 2020a; Misra et al., 2021; Liu et al., 2021b; Zhang et al., 2022c; Pei et al., 2023; Lu et al., 2023; Huang et al., 2024b; Dang et al., 2025; Jin et al., 2025b); **voxel-based** methods discretize space and learn from sparse voxel tensors, enabling scalable context modeling with sparse 3D convolutions or sparse transformers but introducing quantization effects (Zhou & Tuzel, 2018; Yan et al., 2018; Deng et al., 2021; Mao et al., 2021b; Chen et al., 2022e; Fan et al., 2022b; Wang et al., 2023a; Chen et al., 2023f; Zhang et al., 2024c; Jin et al., 2025a; Yu et al., 2025b; Liu et al., 2025e); and **pillar-based** methods collapse height into vertical columns and process BEV pseudo-images efficiently, typically trading some vertical detail for speed and simplicity (Lang et al., 2019; Shi et al., 2022a; Li et al., 2023c; Zhou et al., 2025f; Noh et al., 2025). **Projection-based** approaches map points to 2D views (e.g., BEV or range images) to leverage mature 2D backbones, benefiting efficiency and dense reasoning while inheriting view-specific information loss and occlusion artifacts (Chen et al., 2017; Yang et al., 2018a; Ku et al., 2018; Meyer et al., 2019; Fan et al., 2021; Li et al., 2024j; 2023j; Liu et al., 2023b; Li et al., 2024i; Lee et al., 2025d; Hsu et al., 2025). Finally, **hybrid** designs explicitly combine multiple encodings and/or modalities to balance accuracy and efficiency, for example through point-voxel interaction, cross-view fusion, or multi-sensor fusion (Shi et al., 2020a; 2023c; Wang et al., 2021a; Deng et al., 2022; Gu et al., 2022; Bai et al., 2022; Wang et al., 2022d; Chen et al., 2023c; Li et al., 2024e; Lin et al., 2024d; Chae et al., 2024; Song et al., 2024; Cho et al., 2025; Huang et al., 2025f). Since these choices are modular, many detectors naturally mix representations, so we treat this taxonomy as overlapping tags rather than a strict partition.

### 3.3.2 Detection Stage

Most modern 3D detectors follow either a **one-stage** or **two-stage** design. **One-stage** methods predict 3D bounding boxes directly from dense scene features (e.g., BEV, voxel, or point features) using a single detection head. This design usually provides a simpler pipeline and lower latency, which is suitable for real-time applications (He et al., 2020a; Fan et al., 2022b; Li et al., 2022d; Wang et al., 2023a; Liu et al., 2023b; Li et al., 2024i). These methods include convolution-based and query-based detectors, and some approaches also use lightweight refinement without a separate proposal refinement stage (Fan et al., 2022b; Liu et al., 2023b).

In contrast, **two-stage** detectors first generate object proposals and then refine them with RoI feature extraction and a second refinement module. This strategy can improve localization accuracy and recall, especially in sparse or long-range scenes, but it usually requires more computation (Shi et al., 2020a; 2023a; Deng et al., 2021; Li et al., 2021e; Mao et al., 2021a; Sheng et al., 2021; Shin et al., 2025). Recent two-stage methods further improve RoI features by using multi-scale features, point-voxel fusion, better proposal refinement, and iterative correction modules to reduce false positives and improve box alignment (Shi et al., 2023a; Mao et al., 2021a; Sheng et al., 2021; Shin et al., 2025).

### 3.3.3 Anchor Strategy

The design of object proposals in 3D detection heavily depends on the chosen anchor strategy, which determines how initial bounding box estimates are formulated. Historically, the field has been dominated by **anchor-based** methods (Zhou & Tuzel, 2018; Lang et al., 2019), where a dense grid of pre defined 3D bounding boxes is placed across the point cloud or its proxy representation such as voxels or pillars. The network then learns to predict class probabilities and regress residual offsets from these pre defined anchors to the ground-truth boxes. While this paradigm benefits from stable training and high recall, defining anchor sizes, ratios, and orientations requires extensive dataset-specific hyperparameter tuning. Furthermore, evaluating thousands of empty anchors in sparse 3D space introduces significant computational overhead.

To mitigate these drawbacks, a prominent shift has occurred toward **anchor-free** methods (Shi et al., 2019; Qi et al., 2019; Yin et al., 2021). These approaches eliminate the need for heuristic anchor generation by either modeling 3D objects as keypoints at their spatial centers and regressing bounding box properties directly from these features, or by utilizing bottom-up point-wise voting mechanisms. Accordingly, this section compares these two primary paradigms by evaluating their computational efficiency, flexibility in handling diverse object scales, and overall localization performance.

### 3.3.4 Backbone Type

The architectural choice inherently governs the trade-off among inference speed, robustness to point sparsity, and spatial localization precision. Historically, **CNN-based** architectures have been the most prevalent, typically applied after the data is rasterized into a regular grid. Standard 2D CNNs provide high efficiency for pillar-based representations, while sparse 3D convolutions dominate voxel-based methods by explicitly avoiding computations in empty space (Chandorkar et al., 2025; Shi et al., 2022b; Fuengfusin et al., 2026). The primary advantage of CNN backbones is their highly optimized inference speed, making them the standard for real-time applications. However, the requisite voxelization or projection processes inevitably discard fine-grained geometric details. This loss of resolution can degrade precise bounding box boundary estimation and limit the detection accuracy for small objects, such as pedestrians, especially at long ranges.

More recently, **transformer-based** backbones have gained significant traction in the 3D domain (Mao et al., 2021b; Fan et al., 2022b). By leveraging self-attention mechanisms, architectures such as Voxel Transformer (VoTr) and Single Stride Transformer (SST) capture long-range dependencies and global contextual information across the sparse 3D scene, addressing the inherent receptive field limitations of standard local convolutions. The global receptive field enables transformers to excel under challenging conditions like severe occlusion and distant point sparsity. By utilizing scene-wide contextual cues, they can reliably infer object presence even when local evidence is fragmented. The main drawback is their high computational footprint and memory requirements, which can bottleneck the frames-per-second (FPS) rate in practical deployments.

Finally, Graph Neural Network **(GNN)** backbones offer a distinct paradigm by representing the point cloud directly as a graph (Shi & Rajkumar, 2020b). In this formulation, points serve as nodes and local geometric neighborhoods define edges, allowing the network to natively preserve and exploit topological structures without prior voxelization. By directly processing raw points, GNNs achieve superior geometry preservation, which translates to highly precise 3D localization and accurate orientation (heading) estimation for complex shapes. Nonetheless, the substantial computational cost of dynamic neighborhood search across millions of points in large-scale outdoor scenes often restricts GNNs to localized bounding-box refinement stages or necessitates aggressive point downsampling.

### 3.3.5 Scene Type

Scene type plays an important role in 3D object detection, as it affects point cloud characteristics, occlusion levels, and computational requirements (Misra et al., 2021; Li et al., 2023c; Fan et al., 2022c). In this section, we focus only on methods that use point clouds as input, either independently or in combination with other sensors.

Indoor scenes usually contain denser point clouds because objects are observed at shorter distances. However, these environments are often cluttered, and objects may occlude one another (Qi et al., 2020a; Zhang et al., 2020b). Therefore, indoor detection methods commonly emphasize local geometric features and object-level reasoning (Zhang et al., 2020b; Liu et al., 2021b; Shen et al., 2023). Many approaches use point-based processing (Qi et al., 2020a; Zhang et al., 2020b) or transformer-based set prediction (Misra et al., 2021; Liu et al., 2021b; Shen et al., 2023), while several effective baselines are built on sparse voxel features with anchor-free detection heads (Rukhovich et al., 2022). When RGB-D data are available, some methods combine image features with point cloud information to improve semantic understanding in complex indoor environments (Qi et al., 2020a).

Outdoor driving scenes, in contrast, generate sparse and non-uniform point clouds over large areas, and point density decreases rapidly as distance increases (Yang et al., 2020b; Fan et al., 2021; 2022c). As a result, outdoor detection methods often rely on scalable representations such as bird's-eye-view (BEV) maps, voxel grids, range-view projections, and fully sparse representations, which allow large receptive fields and efficient inference (Yin et al., 2021; Deng et al., 2021; Fan et al., 2021; Li et al., 2023c; Chen et al., 2023f; Fan et al., 2022c). Common architectures include fast single-stage or center-based detectors (Zheng et al., 2021b; Yang et al., 2020b; He et al., 2020a; Yin et al., 2021; Li et al., 2023c), as well as two-stage, temporal, or hybrid models that refine object localization, especially for distant objects (Deng et al., 2021; Li et al., 2021e; Chen et al., 2022d). Sparse attention mechanisms and transformer-based backbones are also used to capture broader context from sparse point clouds (Mao et al., 2021b; Fan et al., 2022b; Zhou et al., 2022c; Wang

et al., 2023a). Moreover, LiDAR-camera fusion is widely applied in outdoor settings, since camera features provide useful semantic information and can improve detection in challenging cases such as distant, partially visible, or occluded objects (Vora et al., 2020; Yoo et al., 2020; Bai et al., 2022; Liang et al., 2022; Li et al., 2024e; Zhang et al., 2024e; Wang et al., 2025e).

In summary, indoor 3D object detection usually focuses on local geometry and object-centered modeling (Zhang et al., 2020b; Misra et al., 2021; Liu et al., 2021b; Shen et al., 2023), whereas outdoor detection places more emphasis on large-scale context, efficient sparse processing, and the frequent multi-sensor integration (Yin et al., 2021; Fan et al., 2022b; Wang et al., 2023a; Fan et al., 2022c; Liang et al., 2022; Zhang et al., 2024e).

### 3.3.6 Supervision

Most 3D point cloud object detectors are developed and evaluated in the fully-supervised setting (Mousavian et al., 2017; Shi et al., 2019; Yin et al., 2021; Pu et al., 2025), where large-scale datasets provide precise 3D bounding boxes for training. While this remains the dominant paradigm and the main source of state-of-the-art results, 3D box annotation is costly and time-consuming, motivating a growing body of work that reduces the need for dense labels. Accordingly, this section reviews limited-supervision 3D detection along three settings: **weakly-supervised** methods that replace full 3D boxes with weaker cues (e.g., 2D/image signals, clicks, tags, or partial labels) to enable 3D localization (Meng et al., 2020; Qin et al., 2020; Meng et al., 2021; Xu et al., 2022; Zhang et al., 2023b; Huang et al., 2024c; Zhang et al., 2024k; Han et al., 2024); **semi-supervised** methods that combine a small labeled set with abundant unlabeled data using teacher–student learning, pseudo-labels, or consistency regularization (Zhao et al., 2020a; Wang et al., 2021b; Yin et al., 2022a; Chen et al., 2023g; Wang et al., 2023f; Wu et al., 2024d; Zhang et al., 2024f; Zhao et al., 2025c; Lee et al., 2025a; Hazra et al., 2025; Park et al., 2025); and **self-supervised or unsupervised** approaches that learn transferable 3D representations from unlabeled point clouds (e.g., reconstruction or masked modeling) for downstream detection, or attempt object discovery without explicit box annotations (Xie et al., 2020; Liang et al., 2021; Rao et al., 2021; Erçelik et al., 2022; Yin et al., 2022b; Lentsch et al., 2024; Zhang et al., 2024j; Baur et al., 2024; Shrout et al., 2025; Zhang et al., 2025g; Xia et al., 2025a).

### 3.3.7 Conclusion, Challenges, and Future Directions of Detection

In summary, each axis introduces fundamental trade-offs. Point-based representations preserve fine geometry but suffer from irregular memory access; voxel/pillar methods enable efficient convolution at the cost of quantization artifacts; hybrid designs balance both but increase system complexity. Two-stage detectors achieve higher accuracy through refined region proposal, while one-stage methods prioritize speed and simplicity. Anchor-based strategies offer stable training but require heuristic design, whereas anchor-free approaches reduce hyperparameter tuning but may struggle with oriented objects. Multi-modal fusion (LiDAR–camera) currently sets state-of-the-art performance on benchmarks like nuScenes and Waymo, yet introduces sensor synchronization and calibration challenges. Transformer backbones are increasingly replacing CNNs for long-range context, but at higher computational cost. Outdoor detectors must handle extreme sparsity and scale variation, while indoor methods focus on occlusion and fine-grained layout. Finally, the supervision axis reveals a clear spectrum: fully supervised methods remain the accuracy leader, but weak, semi-, and self-supervised approaches are rapidly closing the gap, driven by the high cost of 3D annotation.

**Remaining Challenges**: (1) robustness to domain shift and sensor corruption — most methods degrade sharply on unseen weather or LiDAR configurations; (2) open-vocabulary and zero-shot 3D detection, where current models are tightly bound to fixed class vocabularies; (3) efficient long-range detection for high-resolution sensors (e.g., 128-beam LiDAR or 4D radar); and (4) truly unsupervised object discovery in point clouds without any human labels.

**Promising Future Directions**: unifying the six axes into learnable architectures that adapt representation and stage count per scene; leveraging large-scale pre-training (e.g., masked autoencoding) specifically for 3D geometry; integrating language and 3D for instruction-based detection; and developing benchmark suites that systematically evaluate trade-offs along each axis separately, enabling practitioners to select methods based on their specific constraints (e.g., speed vs. accuracy vs. label budget).

### 3.4 Tracking

3D object tracking is a fundamental computer vision task that continuously locates objects across consecutive frames in 3D space. Based on the number of targets, tracking methods can be categorized into Single-Object Tracking (SOT) and Multi-Object Tracking (MOT). Figure 3 illustrates this taxonomy.

#### 3.4.1 Single-Object Tracking

SOT specifies a target in the initial frame and tracks it across subsequent frames. While most research in the past decade has focused on deep neural network (DNN) approaches (Wang et al., 2024c), recent studies have shifted toward multi-modal trackers to combine the strengths of different modalities. Therefore, this survey categorizes SOT methods as either LiDAR or RGB-LiDAR trackers.

**LiDAR Trackers**   LiDAR-based tracking works with 3D point clouds captured by LiDAR sensors. These sensors are robust to lighting changes and weather conditions, but working with this data is challenging. LiDAR-based tracking methods are categorized into three frameworks: siamese-based, motion-based, and transformer-based.

- **Siamese-based**

  A shared Siamese backbone extracts features from both the template and the search area. The similarity between these features is then computed using a metric such as cosine similarity. SC3D (Giancola et al., 2019) combines a Siamese network with an autoencoder to encode sparse point clouds, after which a decoder reconstructs the 3D shapes. However, this method requires an extensive search process. 3D-SiamRPN (Fang et al., 2020) uses a region proposal network (RPN) to directly predict 3D bounding boxes by introducing two cross-correlation modules. The Siamese paradigm can be combined with other architectural techniques such as VoteNet, voxelization, and transformers, particularly for handling 3D point clouds.

  - **VoteNet-based:** These methods combine siamese feature matching with VoteNet-style proposal generation. P2B (Qi et al., 2020b) improves SC3D by an end-to-end learning approach that avoids extensive search and uses Hough voting (Qi et al., 2019) to guide proposal generation and verification. MLVSNet (Wang et al., 2021g) uses multi-level Hough voting and target-guided attention with vote-cluster modeling to achieve better localization accuracy. BAT (Zheng et al., 2021a) improves P2B (Qi et al., 2020b) by using 3D box information from the first frame with a new BoxCloud method and Box-Aware Feature Fusion.
  - **Voxel-based:** These methods handle unstructured and sparse LiDAR data through voxel-based processing combined with the siamese paradigm. V2B (Hui et al., 2021) uses a siamese network to learn shape-aware features and then projects the voxelized features into a dense bird's-eye-view representation.
  - **Siamese-Transformer:** These methods enhance feature representation and model the relationship between the template and the search region. PTT (Shan et al., 2021) introduces a transformer-based module leveraging self-attention, making P2B (Qi et al., 2020b) more robust to sparsity and disorder. PTTR (Zhou et al., 2022a) proposes a coarse-to-fine approach, first performing broad localization to estimate the target region and then refining the target's position and orientation.

- **Motion-based**

  Motion modeling trackers process consecutive frames, allowing them to learn movement patterns that improve tracking efficiency. M2-Track (Zheng et al., 2022) is a two-stage approach that first localizes targets and then refines bounding boxes, which is effective in cluttered scenes. DMT (Xia et al., 2023) consists of two modules: the Motion Prediction Module provides an initial target estimate, which is then refined by the Voting Module. SiamCUT and MoCUT (Nie et al., 2024) introduce a unified network, AdaFormer, which encodes geometric variations across different object types. FocusTrack (Zhou et al., 2025e) proposes a one-stage tracker that jointly models motion

and semantics. Its inter-frame motion-modeling module captures differences between frames and then guides a Focus-and-Suppress attention module. TrajTrack (Fan et al., 2025) is a two-frame tracker with trajectory-based motion modeling that uses TrajFormer (a transformer architecture) to predict future trajectories.

- **Transformer-based**

  With the success of transformer architectures, researchers have increasingly adopted transformers for LiDAR tracking. PTT-Net (Shan et al., 2022) uses an attention mechanism with two stages: voting and proposal generation. CXTrack (Xu et al., 2023a), instead of cropping frames, processes full point clouds to keep context, which improves robustness to occlusions and appearance changes. MBPTrack (Xu et al., 2023b) addresses appearance changes with two innovations: a transformer-based memory module and a box-prior localization network. CorpNet (Wang et al., 2023c) simplifies tracking by using a single encoder that both extracts and finds correlated template-search features using multi-level self- and cross-attention. HVTrack (Wu et al., 2024c) consists of three modules: Relative-pose-aware memory, base-expansion cross-attention, and contextual point self-attention. This method handles large frame-to-frame variations.

**RGB-LiDAR Trackers**  RGB-LiDAR trackers leverage the complementary advantages of both modalities. there has been limited research in this area. MMF-Track (Li et al., 2023l) provides more reliable tracking by creating textured pseudo-points, then uses attention modules to connect modalities at different scales. PTTR++ (Luo et al., 2024b) improves PTTR (Zhou et al., 2022a) accuracy by fusing bird's-eye-view and raw point features. MVCTrack (Hu et al., 2025) segments the target and uniformly samples virtual points, then assigns depths and projects them into 3D space, which improves tracking of far-away or small objects. CCETrack (Yang et al., 2025e) aligns the global context across modalities and performs fusion in the bird's-eye-view space, helping better distinguish similar objects.

In summary, Siamese-based tracking is a widely adopted, efficient, and fast paradigm that focuses on capturing intrinsic geometric information. This method can be combined with VoteNet, voxelization, and transformers. Motion-based tracking takes a different approach by modeling frame-to-frame motion, which is highly robust when the target's appearance varies. Transformer-based tracking uses a simple single-branch framework, which needs fewer parameters and is more efficient(Wang et al., 2024c). LiDAR models face challenges due to sparse, textureless 3D point clouds. For this reason, RGB-LiDAR fusion models have been developed to improve accuracy and robustness.

### 3.4.2  Multi Object Tracking

MOT plays a pivotal role in modern computer vision systems. Consequently, recent research prioritizes methods that balance high tracking accuracy with computational efficiency. As illustrated in Figure 3, these methods can be categorized into three primary paradigms: Tracking by Detection (TBD), and Unified Detection and Tracking.

Seminal works such as AB3DMOT (Weng et al., 2020) demonstrate that streamlined classical approaches can balance accuracy and speed. While probabilistic methods improved precision through uncertainty modeling, recent data-driven architectures achieve superior performance, particularly when handling sparse point cloud data. However, this substantial gain in accuracy often incurs higher computational costs, creating a trade-off between lightweight CPU-based algorithms and heavy, GPU-dependent deep learning models. A comparative summary of these paradigms is provided in Table 4.

**Tracking by Detection**  Tracking by Detection divides the pipeline into sequential detection, association, and path management steps. While traditional baselines rely on Kalman filters, recent methods incorporate learned motion models or geometric constraints to better handle occlusion (Li et al.; Wang et al., 2021f). However, because TBD performance is strictly limited by the detector's quality, current research emphasizes confidence-aware metrics to mitigate the impact of false positives (Pang et al., 2022c; Nagy et al., 2024).

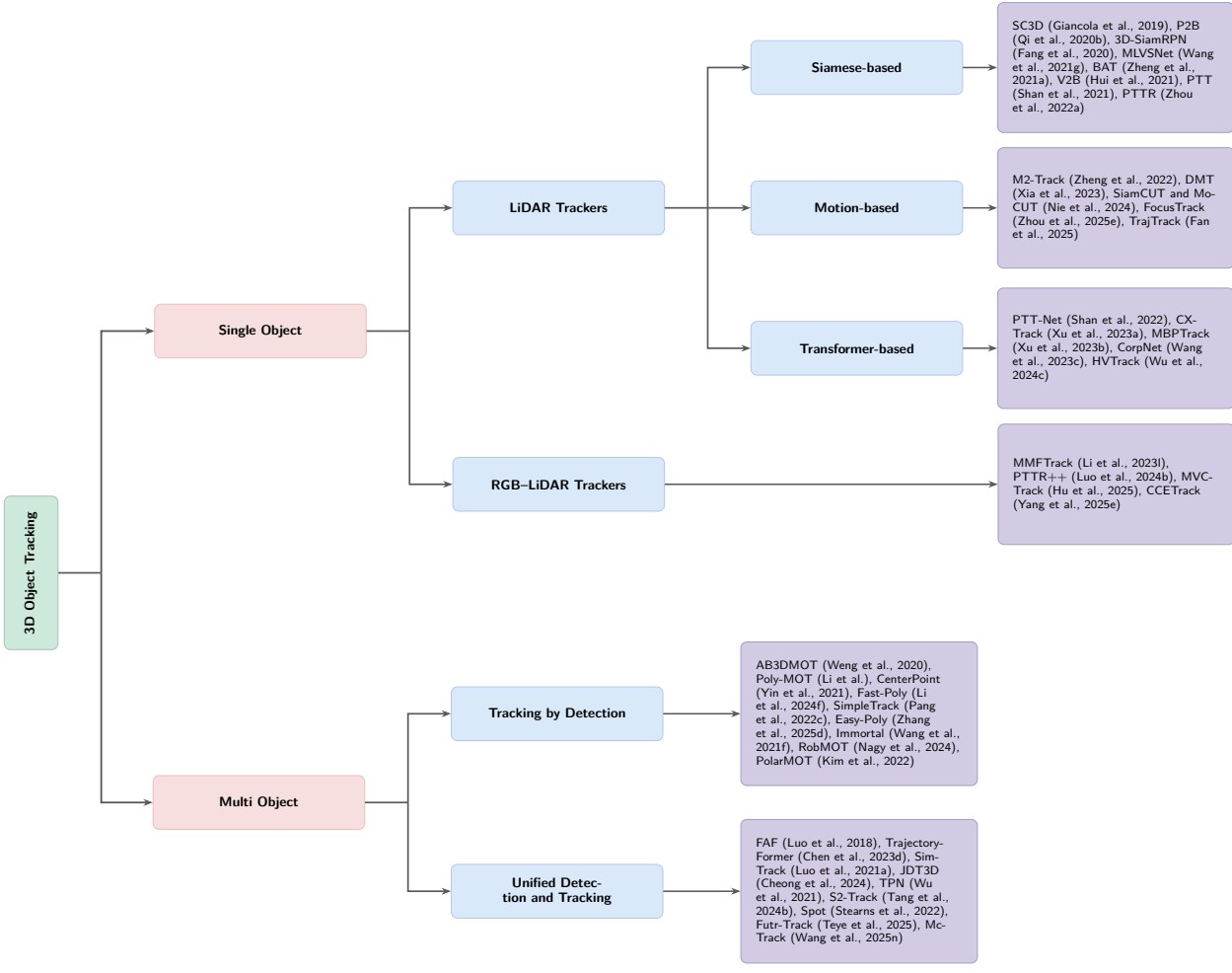

Figure 3: A comprehensive taxonomy of 3D object tracking methods.

Table 4: Advantages and Limitations of 3D MOT Paradigms.

| Paradigm | Advantages | Limitations |
|---|---|---|
| Tracking by Detection | Detectors and trackers optimized independently
Fast classical associations
Traceable failures | Dependent on detection quality
Temporal information loss
Requires heuristics |
| Unified Detection & Tracking | Joint training
Attention handles occlusion
No hand-crafted heuristics | Needs massive datasets
Computationally expensive
Low interpretability |

**Unified Detection and Tracking** Unified frameworks integrate detection and association to prevent the error propagation commonly observed in sequential pipelines. Early approaches, such as FAF (Luo et al., 2018), used shared backbones to improve inference speed but often retained separate post-processing steps. The paradigm later shifted toward transformer-based architectures, such as TrajectoryFormer (Chen et al., 2023d), which use learnable queries to unify tasks without manual matching. More recently, efficient methods like SimTrack (Luo et al., 2021a) and S2-Track (Tang et al., 2024b) have focused on simplifying these architectures by replacing complex heuristics with direct trajectory learning to enhance robustness.

### 3.4.3 Conclusion, Challenges, and Future Directions of Tracking

Despite significant advances, critical challenges remain for real-world deployment. Future research must move beyond closed-set limitations to address open-set tracking of novel objects without retraining (Ishaq et al., 2025). Additionally, cooperative perception (Chiu et al., 2024) offers a promising solution to occlusion, provided that bandwidth and synchronization challenges can be overcome. Finally, bridging the gap between high-performance deep learning models and the strict latency constraints of embedded edge devices remains a priority.

## 4 Geometric Modeling, Alignment, and Pose Estimation

Geometric modeling, alignment, and pose estimation tasks are grouped together because they all focus on the geometric structure and spatial relationships of 3D point clouds rather than only semantic labeling. These tasks involve preserving, generating, recovering, aligning, or interpreting point cloud geometry. This section reviews compression (Section 4.1), generation (Section 4.2), reconstruction (Section 4.3), registration (Section 4.4), and 6DoF pose estimation (Section 4.5) under this shared geometric perspective.

### 4.1 Compression

3D Point Cloud Compression (3DPCC) is a critical task in modern 3D data processing pipelines, particularly as 3D sensors such as LiDAR, depth cameras, and SLAM systems generate massive volumes of spatial data. A point cloud typically comprises a set of 3D coordinates, optionally enriched with attributes such as color or reflectance. The compression process aims to reduce storage and transmission costs while preserving the essential geometric and semantic content for downstream tasks. To systematically analyze existing methods, we categorize 3DPCC models along four dimensions: the type of compression (lossy or lossless), the data component being compressed (geometry, attributes, or joint), the temporal nature of the data (static, dynamic, or dynamically acquired), and the architectural strategy used (e.g., voxel-based, octree-based, point-based, projection-based, or graph-based). Figure 4 illustratesx the categories of 3DPCC.

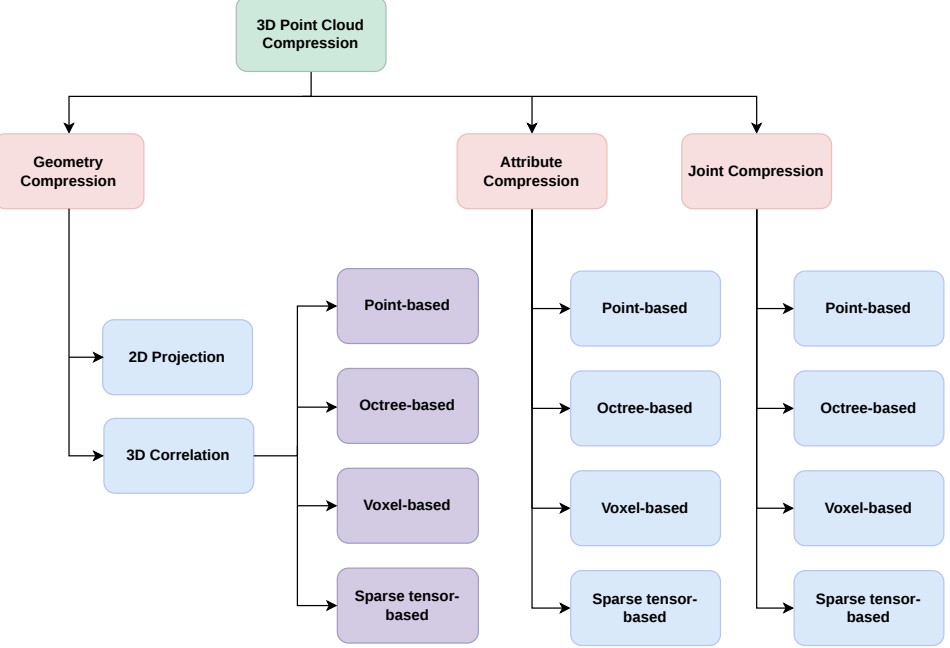

Figure 4: Overview of 3D point cloud compression categories.

Compression type differentiates models that reconstruct point clouds with some allowable error (lossy) from those that reproduce them exactly (lossless). Most current deep learning-based methods fall under the lossy category due to the high compression gains afforded by learned quantization and entropy coding. Data component refers to whether the method focuses solely on geometry (spatial coordinates), attributes such as color, or both jointly. Joint compression remains relatively underexplored but is growing in importance, especially for immersive media and robotics. Data type considers whether the model targets static point clouds, dynamic sequences such as moving humans or real-time, dynamically-acquired data. Finally, architecture type classifies the algorithmic framework, including voxel grids, hierarchical octrees, graph transforms, 2D projections such as range images, and point-wise implicit functions.

**Projection-based** methods convert 3D point clouds into 2D representations, such as planar images, range images, or depth maps, enabling the use of conventional image or video compression codecs to exploit spatial and temporal redundancies (Van Der Hooft et al., 2019; Liu et al., 2020c; 2021a; Gao et al., 2022; Wang & Liu, 2022; Wang et al., 2024d). In contrast, **point-based** methods operate directly on raw point clouds, allowing the processing of irregular and unordered point sets without requiring intermediate structured representations (Huang & Liu, 2019; Wiesmann et al., 2021; You et al., 2024; Xie et al., 2024a). **Octree-based** methods organize point clouds into hierarchical octree structures by recursively subdividing 3D space into occupied cubes, where each node is encoded using compact occupancy codes that can be efficiently compressed with entropy models (Schnabel & Klein, 2006; Wang et al., 2021c; Que et al., 2021; Fu et al., 2022; Pang et al., 2022b; Fang et al., 2022; Song et al., 2023b; Nguyen & Kaup, 2023; Fan et al., 2023; Akhtar et al., 2024; Jin et al., 2024b; Wang et al., 2025o; Zheng et al., 2026a; Jiang et al., 2026). **Voxel-based** methods transform point clouds into regular volumetric grids, representing spatial information through binary voxel occupancy, which facilitates neighbor indexing and the application of 3D convolutions while leveraging previously decoded voxels for improved compression efficiency (Zhang et al., 2014; Thanou et al., 2015; Quach et al., 2019; Wang et al., 2021d; Que et al., 2021; Pang et al., 2022b; Nguyen & Kaup, 2023; Ruan et al., 2024; Rudolph et al., 2024). To address the high computational cost associated with dense voxel representations, **sparse tensor-based** methods employ sparse voxelization and sparse convolutions to process only occupied voxels (Wang & Ma, 2022; Pang et al., 2022b; Nguyen & Kaup, 2023; Wang et al., 2023b; Zhang et al., 2023e; Xie et al., 2024b; Mao et al., 2025; Huo et al., 2025; Fu et al., 2026). These methods significantly reduce computational complexity and have demonstrated strong performance in point cloud compression, typically requiring quantized integer point clouds to enable efficient sparse convolution operations. Table 5 summarizes representative compression methods.

### 4.1.1 MPEG Models

With the rise of deep learning-based point cloud compression (PCC), a key challenge has been how to fairly evaluate and standardize new approaches. Traditional codecs such as G-PCC and V-PCC were insufficient for handling varied point cloud formats, including LiDAR and dense scans. To address this, researchers proposed benchmarking frameworks assessing both compression efficiency and computational cost. Zaghetto et al. (2022) introduced a general evaluation interface, while Fan et al. (2022d) developed D-DPCC, a deep PCC framework for LiDAR using multiscale motion fusion and adaptive interpolation to better model temporal dynamics. These advancements enabled deep models to outperform traditional handcrafted motion compensation techniques.

Fragmentation across implementation frameworks such as TensorFlow and PyTorch has made reproducibility and fair comparison difficult. To overcome this issue, Pang et al. (2022a) introduced pccAI, a PyTorch-based suite for deep PCC model development and evaluation. Improving compression quality also became a major focus. Pang et al. (2022b) proposed GRASP-Net, which separates point cloud compression into base and enhancement layers, combining CNN and MLP modules for richer feature extraction. Zaghetto & Graziosi (2021) and Lodhi et al. (2021), contributed adaptive entropy models that use neural networks for context-based probability estimation, optimizing rate-distortion trade-offs dynamically.

Many deep PCC models struggled to generalize across both sparse LiDAR and dense object clouds. To address this limitation, Wang & Ma (2022) presented SparsePCGC, a unified framework for lossy and lossless compression using sparse tensors. This framework was extended by SparsePCGCv2 (Wang et al., 2022b), which added KNN-based neighborhood attention for better context modeling. For dynamic PCC,

Table 5: Schema of point cloud compression types, categories and data types.

| Type | Category | Data type | Methods |
|---|---|---|---|
| Lossy | Geometry | Static | (Huang & Liu, 2019), (Quach et al., 2019), (Wang et al., 2021c), Voxelcontext-net (Que et al., 2021), (Wang et al., 2021d), Octattention (Fu et al., 2022), GRASP-Net (Pang et al., 2022b), (Song et al., 2023b), OctFormer (Cui et al., 2023), Ecm-opcc (Jin et al., 2024b), (Xie et al., 2024b), (Zheng et al., 2026a) |
| | | Dynamic | (Akhtar et al., 2024), OctMamba (Jiang et al., 2026) |
| | | Dynamically acquired | (Wang & Liu, 2022), (Fan et al., 2023), msLPCC (Wang et al., 2024d) |
| | Attribute | Static | (Zhang et al., 2014), (Wang & Ma, 2022), PCAC-GAN (Mao et al., 2025), (Huo et al., 2025), DeepRAHT (Fu et al., 2026) |
| | Joint | Static | YOGA (Zhang et al., 2023e), (Ruan et al., 2024), (Rudolph et al., 2024) |
| | | Dynamic | (Thanou et al., 2015), (Gao et al., 2022) |
| | | Dynamic + dynamically acquired | (Van Der Hooft et al., 2019), (Liu et al., 2020c), (Liu et al., 2021a) |
| Lossless | Geometry | Dynamic | Octree-STCM (Wang et al., 2025o) |
| | Attribute | Dynamic | (Wang et al., 2023b), (Fang et al., 2022) |
| | Joint | Static | (Schnabel & Klein, 2006), (Nguyen & Kaup, 2023) |

Akhtar et al. (2024) introduced DPCGC-SC, which uses sparse CNNs to propagate inter-frame geometry and predict latent codes, enhancing temporal coherence. Attribute compression has also progressed: PCAC (Wang & Ma, 2022) employed variational autoencoders and sparse convolution for color compression, while Pinheiro et al. (2023) proposed NF-PCAC, a normalizing flow model for high- and low-bitrate settings. Geometry compression was improved by Zhou et al. (2023e), who jointly optimized down-sampling and up-sampling for better reconstruction. Qi & Gao (2024) introduced a variable-rate architecture with scale-aware transformations, and Yu & Gao (2024) developed a plug-in entropy estimator. Zuo et al. (2024a) further enhanced dynamic PCC with a temporal contextual pyramid for cross-frame information fusion at lower computational costs.

MPEG point cloud compression standardization has continued to evolve through improvements to both geometry-based and video-based coding frameworks. Enhanced G-PCC (E-G-PCC) (Group, 2024), developed as MPEG-I Part 38, extends the original geometry-based point cloud compression framework with tools better suited for dynamic point cloud sequences and improved temporal coding (Group, 2024; JTC1/SC29, 2026). In parallel, the video-based branch was updated through (JTC1/SC29, 2025), which specifies Visual Volumetric Video-based Coding (V3C) and Video-based Point Cloud Compression (V-PCC) for immersive volumetric media representation. Together, these standards reflect MPEG's continued focus on improving compression efficiency, dynamic scene support, and interoperability for point cloud and volumetric video applications.

### 4.1.2 Conclusion, Challenges, and Future Directions of Compression

In summary, projection-based methods leverage mature 2D codecs but may introduce geometric distortion during 3D-2D mapping. Point-based approaches preserve irregular structures yet face challenges in entropy modeling. Octree-based models provide efficient hierarchical coding, particularly for geometry compression and lossless settings, while voxel-based methods benefit from 3D convolutions but suffer from high computational cost,

Overall, lossy deep learning-based methods dominate due to their strong rate-distortion performance, whereas lossless, joint geometry-attribute, and dynamic compression remain less explored. Future work should focus

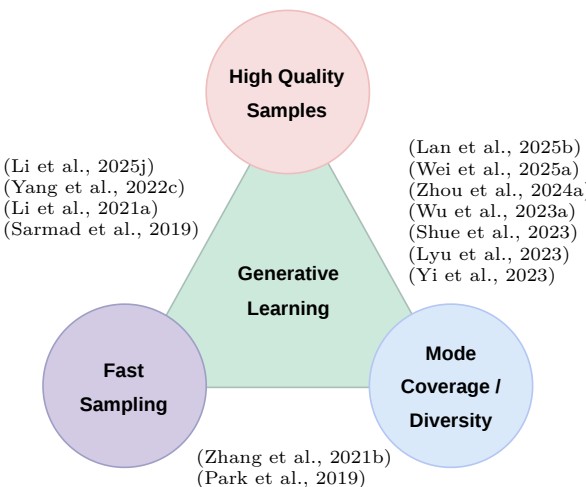

Figure 5: Representative point cloud generative models on the speed–fidelity–coverage trilemma.

on unified and lightweight frameworks, improved joint compression, better temporal modeling, and standardized evaluation for real-world deployment.

## 4.2 Generation

Point cloud generation studies how to synthesize plausible 3D geometry represented as unordered sets of points, optionally with attributes such as color or normals. In this survey, point cloud generation methods are categorized into three main classes according to their conditioning and input assumptions: **unconditional generation**, where models synthesize plausible 3D shapes by sampling from noise or a learned latent space without observing a specific target instance; **point cloud completion**, where models infer missing geometry given a partial, sparse, or noisy observation; and **weakly conditional generation**, where shape synthesis is guided by high-level or indirect cues such as text, images, or sketches. Figure 5 summarizes the common trade-offs among major generative families used across these settings, highlighting the speed, fidelity, and coverage trilemma.

Reliable point cloud generative models serve as priors for 3D content creation such as rapid asset synthesis and editing as well as for ill-posed inverse problems, including denoising, upsampling, and completion. These models help bridge the gap between sparse, noisy sensor measurements and high-fidelity 3D representations.

We distinguish generation from 3D reconstruction. Generation aims to produce a plausible 3D shape from noise or weak conditioning, without requiring fidelity to a specific scene instance. Reconstruction, in contrast, takes one or multiple images (and possibly calibration) and attempts to recover the exact underlying 3D scene consistent with observations. Reconstruction is addressed in a section 4.3 of this survey.

Point cloud generation faces several challenges:

- Permutation invariance and the lack of grid structure, which complicate likelihood modeling and diffusion in Euclidean space.

- Non-uniform sampling and varying density across shape parts or sensors such as LiDAR range-dependent sparsity.

- The trade-off between surface fidelity and global plausibility, especially under severe incompleteness.

- Representation choice, where points compete with implicit fields (SDF/occupancy) and emerging surfel/Gaussian primitives in terms of editability and renderability.

### 4.2.1 Unconditional Generation

Early approaches often used adversarial learning to match shape distributions while preserving point-set properties. SP-GAN (Li et al., 2021a) adopted a sphere-guided design that promotes dense correspondence and part-aware manipulation through a sphere-like proxy that stabilizes generation and editing, while a two-stage category-conditioned generator with a BranchGCN-style design reduced hybrid artifacts and improved controllability (Yang et al., 2022c).

Diffusion methods now dominate due to their robustness and sample quality. FrePolAD (Zhou et al., 2024a) used frequency-rectified latent diffusion to preserve both low-frequency global structure and high-frequency details for scalable, diverse synthesis, and straight flow distillation straightens diffusion trajectories to enable one-step generation while maintaining quality (Wu et al., 2023a). Diffusion methods also targets alternative 3D representations that support point sampling or surface extraction. Triplane diffusion generated 3D neural fields from 2D priors (Shue et al., 2023), and controllable mesh generation diffused sparse semantic latent points and decoded them with a surface-aware procedure for efficient mesh synthesis (Lyu et al., 2023). Together, these works highlighted a shift toward latent and structured spaces that are more diffusion-friendly than raw unordered points.

### 4.2.2 Point Cloud Completion

Point cloud completion aims to recover a full 3D shape from an incomplete or noisy point cloud. Traditional geometric or retrieval-based methods are effective mainly for simple, mostly complete objects and struggle with unseen or complex shapes. Deep learning overcomes these limitations by learning shape priors from large datasets, enabling more reliable inference of missing geometry.

Early deep models such as PCN (Yuan et al., 2018) directly predicted complete point sets, while FoldingNet (Yang et al., 2018b) and AtlasNet (Groueix et al., 2018) introduced flexible surface-generation decoders, and PU-Net (Yu et al., 2018a) improved point density and uniformity. To better capture structure, later works like TopNet (Tchapmi et al., 2019), RL-GAN-Net (Sarmad et al., 2019), and PF-Net (Huang et al., 2020) modeled topology, realism, uniform sampling, and large missing regions using hierarchical decoding, adversarial learning, and fractal-style growth. PoinTr (Yu et al., 2021b) treated completion as set-to-set translation, SeedFormer (Zhou et al., 2022b) used compact seed patches for efficient reconstruction, and AnchorFormer (Chen et al., 2023i) leveraged learned anchor nodes to produce high-fidelity local details.

A parallel multimodal branch emerged, combining RGB images with point clouds. ViPC (Zhang et al., 2021b) introduced image-guided completion, XMFNet (Aiello et al., 2022) strengthened cross-modal attention, EGIINET (Xu et al., 2024a) improved feature alignment, and MAENet (Liu et al., 2025d) enhanced both multimodal accuracy and uniformity. These models use 2D visual cues to guide more accurate 3D completion. Recent works advance point cloud completion with new generative, multimodal, and domain-adaptive strategies: WalkFormer (Zhang et al., 2024i) improved structural reasoning, PCDreamer (Wei et al., 2025a) and other diffusion-based models inferred shapes through multi-view or latent diffusion, DuInNet (Liu et al., 2025h) strengthened image-point fusion, and DAPoinTr (Li et al., 2025l) reduced the synthetic-to-real gap using domain-aligned transformers.

The main challenges in point cloud completion include dealing with large missing regions, uneven sampling, noise, and ambiguity when many plausible completions exist. Additionally, models must generalize from synthetic to real data, handle varying object categories and scales, and produce dense, accurate, and uniform outputs suitable for downstream 3D tasks.

### 4.2.3 Weakly Conditional Generation

Beyond unconditional sampling and completion, recent work emphasizes controllable generation conditioned on language, sketches, images, or sensor modalities. For open-vocabulary 3D asset creation, Gaussian primitives are attractive due to real-time rendering and editability. GaussianDreamer (Yi et al., 2023) generated

3D Gaussian splatting assets from text by fusing 3D and 2D diffusion priors with point cloud guidance, while GaussianAnything (Lan et al., 2025b) used point-cloud-structured latent diffusion to synthesize editable surfel Gaussians from text or single images with interactive control and high quality synthesis. For point-set outputs, sketch-and-text-guided diffusion produced colored point clouds through staged diffusion with attention, enabling controllable geometry and appearance (Wu et al., 2023d).

Sensor-conditioned generation is especially active in LiDAR because of it's structured sparsity and strong viewpoint dependence. RangeLDM (Hu et al., 2024) synthesized realistic LiDAR point clouds by combining Hough-corrected range views with latent diffusion and geometry-aware adversarial training, supporting generation, upsampling, and inpainting. Text2LiDAR (Wu et al., 2024f) enabled text-guided LiDAR generation using an equirectangular transformer to model panoramic structure and improve controllability. When both images and sparse points are available, I-PAttnGAN (Li et al., 2025j) used an image-assisted attention GAN to densify sparse point clouds into dense outputs, targeting completion-like densification rather than freeform synthesis. Collectively, these works show that conditioning modality and sensor geometry (panoramic projection, range-image structure) strongly shape effective generative design.

### 4.2.4 Conclusion, Challenges, and Future Directions of Generation

These works exposed clear trade-offs among model families, conditioning signals, and representations. GAN-based generators offer fast inference but often need strong architectural constraints to avoid mode collapse and preserve local surface fidelity (Li et al., 2021a; Yang et al., 2022c). Diffusion models usually provide better coverage and sample quality and transfer naturally to multi-task and multimodal settings such as completion, inpainting, denoising, but require heavier training, slow sampling unless distillation is applied (Zhou et al., 2024a; Hu et al., 2024; Wei et al., 2025a). For completion, transformer-based set-to-set formulations capture global context effectively, and multimodal fusion adds 2D cues; however, robustness can still degrade under extreme sparsity, noise and synthetic-to-real shifts (Yu et al., 2021b; Aiello et al., 2022; Li et al., 2025l). Representation choice also governs usability: implicit fields yield continuous surfaces and mesh extraction (Park et al., 2019; Chen & Zhang, 2019), while Gaussian/surfel primitives enable differentiable rendering and interactive editing for controllable asset creation (Yi et al., 2023; Lan et al., 2025b).

Evaluation remains narrow beyond Chamfer distance (CD) and Earth Mover's Distance (EMD). Metrics and benchmarks that better reflect perceptual plausibility, topology, sensor realism, especially for LiDAR, and calibrated uncertainty for ambiguous multimodal completion are still needed. Controllability is also limited: even with text/sketch conditioning and editable Gaussian primitives, it is hard to satisfy precise geometric/physical constraints (part-level edits, symmetry, contact/physics, visibility/occlusion) while maintaining cross-view and cross-modality consistency (Lan et al., 2025b; Wu et al., 2023d; 2024f). Generalization across datasets and sensing gaps, such as CAD versus real scans, indoor RGB-D versus outdoor LiDAR, and varying density or noise, remains challenging, motivating sensor-aware objectives, domain-aware priors, and test-time adaptation (Li et al., 2025l; Hu et al., 2024). Finally, building unified and efficient generative systems that cover unconditional, completion, and weakly conditional settings with fast sampling and lightweight deployment is still open, pointing to distilled samplers, shared latent spaces, and modular conditioning (Wu et al., 2023a; Du et al., 2025; Wei et al., 2025a).

### 4.3 Reconstruction

3D reconstruction serves as a fundamental bridge between the physical world and digital space, providing the essential representations—points, surfaces, radiance fields, or explicit primitives—required for advanced 3D vision tasks. These representations enable applications such as virtual reality, augmented reality, robotics, and medical imaging. Historically, the field has evolved from rigid multi-view geometry to flexible neural optimization and generative modeling. Recent surveys offer exhaustive taxonomies covering the transition from classical pipelines to Neural Radiance Fields (NeRF) (Mildenhall et al., 2021) and 3D Gaussian Splatting (3DGS) (Liu et al., 2025f), deep learning based surface extraction (Farshian et al., 2023), and specialized 3DGS-based surface modeling (Xu et al., 2025c). We categorize the landscape into two primary families:

- **3D Reconstruction (Scene-Level):** Focuses on recovering holistic scene representations, including geometry and view-dependent appearance. These methods are optimized for high-fidelity view

synthesis, which involves generating novel views from existing images; navigation through Simultaneous Localization and Mapping (SLAM); and immersive mapping for applications like virtual tours or urban planning.

- **Surface Reconstruction:** Aims to extract explicit, topologically consistent geometry such as triangular meshes or implicit surfaces. These representations are indispensable for Computer-Aided Design (CAD), physical simulation in engineering, and high-precision shape analysis in fields like manufacturing or biomedical engineering.

### 4.3.1 3D Reconstruction

Scene-level reconstruction aims to model the appearance and structure of an entire environment. Drawing inspiration from and extending prior taxonomies (Liu et al., 2025f), this survey groups scene-level reconstruction methods into implicit neural fields, and explicit neural primitives.

**Implicit Reconstruction (NeRF and Neural Fields)**   Implicit neural fields represent a paradigm shift by modeling scenes as continuous functions, typically using Multi-Layer Perceptrons (MLPs). **Neural Radiance Fields (NeRF)** map spatial coordinates and viewing directions to color and density via differentiable volume rendering, enabling photorealistic novel view synthesis (Mildenhall et al., 2021; Yu et al., 2021a; Wang et al., 2021e). Modern research focuses on resolving high-frequency artifacts for improved quality (Barron et al., 2021; Martin-Brualla et al., 2021; Zhang et al., 2020a; Zhou et al., 2023b; Wang et al., 2023d; Li et al., 2023d), accelerating training through multi-resolution hash encodings for improved efficiency (Liu et al., 2020b; Müller et al., 2022; Chen et al., 2022a; Li et al., 2023e; Yang et al., 2022a; Barron et al., 2023; Cao et al., 2024a; Hedman et al., 2021), and generalizing to novel scenes using image-aligned features in sparse-view settings (Yu et al., 2021a; Yang et al., 2023; Seo et al., 2023a;b; Gao et al., 2024). **Dynamic Fields** extend this representation to the temporal domain, capturing non-rigid motion through learnable deformation fields (Park et al., 2021; Li et al., 2021d; Pumarola et al., 2021; Shao et al., 2023; Song et al., 2023a; Klenk et al., 2023). The categorization and examples of these methods are summarized in Table 6.

Table 6: Implicit neural 3D reconstruction (NeRF and related neural fields).

| Category | Methods |
|---|---|
| Base NeRF | NeRF (Mildenhall et al., 2021), PixelNeRF (Yu et al., 2021a), IBRNet (Wang et al., 2021e) |
| Quality / Robustness | Mip-NeRF (Barron et al., 2021), NeRF in the Wild (NeRF-W) (Martin-Brualla et al., 2021), NeRF++ (Zhang et al., 2020a), NeRFLiX (Zhou et al., 2023b), BAD-NeRF (Wang et al., 2023d), UHDNeRF (Li et al., 2023d) |
| Efficiency / Compression | NSVF (Liu et al., 2020b), InstantNGP (Müller et al., 2022), TensoRF (Chen et al., 2022a), NeRFAcc (Li et al., 2023e), Recursive-NeRF (Yang et al., 2022a), Zip-NeRF (Barron et al., 2023), LightningNeRF (Cao et al., 2024a), BakingNeRF (Hedman et al., 2021) |
| Sparse-view / Generalizable | PixelNeRF (Yu et al., 2021a), FreeNeRF (Yang et al., 2023), FlipNeRF (Seo et al., 2023a), MixNeRF (Seo et al., 2023b), HG3-NeRF (Gao et al., 2024) |
| Dynamic / 4D Fields | Nerfies (Park et al., 2021), NSFF (Li et al., 2021d), D-NeRF (Pumarola et al., 2021), Tensor4D (Shao et al., 2023), NeRFPlayer (Song et al., 2023a), E-NeRF (Klenk et al., 2023) |

**Explicit Reconstruction (3D Gaussian Splatting)**   3D Gaussian Splatting (3DGS) introduces an explicit, point-based representation using anisotropic Gaussians. Unlike implicit fields, 3DGS enables real-time rasterization and high-quality view synthesis without expensive volumetric sampling (Kerbl et al., 2023). Current advances target quality by eliminating floaters (Yu et al., 2024b; Xie et al., 2024d; Fan et al., 2024c; Navaneet et al., 2024; Girish et al., 2024; Guédon & Lepetit, 2024; Cheng et al., 2024; Yu et al., 2024c; Yan et al., 2024b; Lu et al., 2024b), sparse-view reconstruction through generative priors (Xiong, 2024; Zhu et al., 2024d; Charatan et al., 2024; Yu et al., 2024a; Xiao et al., 2025c), and large-scale urban modeling via scene partitioning (Lin et al., 2024a; Cui et al., 2024; Jiang et al., 2024a; Sun et al., 2024b; Xu et al., 2025a). This representation is particularly effective for SLAM and interactive editing because of its explicit and deformable nature (Wu et al., 2024a; Lu et al., 2024a; Yang et al., 2024c; Lin et al., 2024c; Gan et al.,

2023; Yan et al., 2024a; Sarikamis & Alatan, 2024; Zhou et al., 2024c; Durvasula et al., 2023), making it suitable for real-time applications. A taxonomy of explicit Gaussian-based methods and their prominent works is provided in Table 7.

Table 7: Explicit 3D reconstruction with 3D Gaussian splatting.

| Category | Methods |
|---|---|
| Core | 3DGS (Kerbl et al., 2023) |
| Quality / Compression | GSDF (Yu et al., 2024b), SuperGS (Xie et al., 2024d), LightGaussian (Fan et al., 2024c), CompGS (Navaneet et al., 2024), EAGLES (Girish et al., 2024), SuGaR (Guédon & Lepetit, 2024), GaussianPro (Cheng et al., 2024), Mip-Splatting (Yu et al., 2024c), Multi-scale 3DGS (Yan et al., 2024b), Scaffold-GS (Lu et al., 2024b) |
| Sparse-view GS | SparseGS (Xiong, 2024), FSGS (Zhu et al., 2024d), PixelSplat (Charatan et al., 2024), LM-Gaussian (Yu et al., 2024a), MCGS (Xiao et al., 2025c) |
| Dynamic / SLAM / Editing | 4DGS (Wu et al., 2024a), DN-4DGS (Lu et al., 2024a), Deformable 3DGS (Yang et al., 2024c), Gaussian-Flow (Lin et al., 2024c), V4D (Gan et al., 2023), GS-SLAM (Yan et al., 2024a), IG-SLAM (Sarikamis & Alatan, 2024), DrivingGaussian (Zhou et al., 2024c), DistWar (Durvasula et al., 2023) |
| Large-scale / Multimodal | VastGaussian (Lin et al., 2024a), LetsGo (Cui et al., 2024), Li-GS (Jiang et al., 2024a), LiDARF (Sun et al., 2024b), DepthSplat (Xu et al., 2025a) |

### 4.3.2 Surface Reconstruction

Surface reconstruction focuses on extracting explicit topology such as meshes or surfels, suitable for geometry-aware applications. We distinguish between Eulerian representations (grid-based, e.g., TSDF) and Lagrangian representations (particle/mesh-based, e.g., TetSphere). Modern trends emphasize Implicit Surfaces (SDFs) for watertight geometry and 3DGS-based surfaces which align gaussian primitives with the scene surface to enable photorealistic yet geometrically accurate extraction (Kazhdan et al., 2006; Bernardini et al., 2002; Boissonnat & Geiger, 1993; Fortune, 2017; Newcombe et al., 2011; Yang et al., 2018b; Yuan et al., 2018; Fan et al., 2017; Groueix et al., 2018; Wang et al., 2018c; Gkioxari et al., 2019; Guo et al., 2024; Park et al., 2019; Mescheder et al., 2019; Chen & Zhang, 2019; Niemeyer et al., 2020; Wolf et al., 2024; Wu et al., 2024b; Ye et al., 2024a; Wang et al., 2025j; Zhang et al., 2024a; 2025k; Fan et al., 2024b; Lyu et al., 2024b; Huang et al., 2024a; Yu et al., 2024d; Chen et al., 2023a; Li et al., 2024b; Liu et al., 2024a; Huang & Yu, 2024; Ma et al., 2025a; Xiong et al., 2024; Huang et al., 2024d; Gao et al., 2025b; Feng et al., 2024).

### 4.3.3 Conclusion, Challenges, and Future Directions of Reconstruction

In conclusion, modern 3D scene and surface reconstruction has increasingly converged toward continuous implicit neural fields (Mildenhall et al., 2021) and explicit neural primitives like 3D Gaussian Splatting (Kerbl et al., 2023) to achieve high-fidelity rendering. However, these neural paradigms still struggle with severe overfitting and geometric noise under sparse-view inputs, frequently lacking global topological coherence and producing artifacts in complex or textureless regions (Xu et al., 2025c). To address these structural bottlenecks, future research trajectories are converging on hybrid representations that unify explicit splats with continuous signed distance fields (SDFs), regularized by 3D-native vision foundation models to resolve geometric ambiguities. Ultimately, optimizing these architectures via advanced vector quantization and lightweight dynamic encodings will be crucial for facilitating real-time, edge-native deployment on standalone platforms.

### 4.4 Registration

3D point cloud registration refers to the problem of estimating a spatial transformation, typically a rigid transformation composed of rotation and translation, that aligns a source point cloud with a target point cloud into a common coordinate system while preserving geometric consistency. Over the past decades, great efforts have been made towards this end, and many solutions, including traditional and deep learning based approaches, have been proposed (Zhang et al., 2024l; Yang et al., 2024b). Basically, registration approaches

can be categorized into traditional geometry-based methods and modern learning based approaches, which are shown in Table 8. In the following, we discuss both traditional and learning-based registration strategies, highlighting their main principles, strengths, and limitations.

- **Classical Registration:** This category encompasses geometry-based classical methods for rigid registration of 3D point clouds, centered on the Iterative Closest Point (ICP) algorithm (Besl & McKay, 1992) and its improved variants (Rusinkiewicz & Levoy, 2001; Yuan et al., 2025). These methods rely on explicit point correspondences or distance minimization principles to estimate rigid transformations and are well suited to scenarios with moderate overlap and low noise. Their strengths include simplicity, interpretability, and efficiency in well-conditioned cases, whereas limitations arise under severe outliers (Bustos & Chin, 2017; Wang et al., 2024b), low-overlap scans (Stechschulte et al., 2019; Denayer et al., 2024), and poor initial poses (Zhou et al., 2016; Di Lauro et al., 2024).

- **Learning-based Registration:** Learning-based approaches leverage data-driven models to extract robust feature representations and predict rigid transformations directly from point cloud data. Supervised methods learn point correspondences or global descriptors from labeled data, enabling accurate alignment even under moderate noise and partial overlap (Ao et al., 2023; Wang et al., 2024g; Wu et al., 2023c; Yao et al., 2024). In contrast, unsupervised and self-supervised approaches optimize alignment objectives without explicit ground truth, often using geometric consistency or probabilistic modeling to handle challenging scenarios such as low overlap, partial observations, and large-scale scenes (Mei et al., 2023; Liu et al., 2024d; Zheng et al., 2024a; Jiang et al., 2024c; Yuan et al., 2023). Their main principle is to learn representations that capture local and global geometric structures, while their strengths include robustness to noise, generalization to diverse scenes, and improved handling of partial overlaps. Limitations remain, including dependence on the training data distribution, higher computational cost, and sometimes limited interpretability compared with classical geometry-based methods.

Table 8: Schema of point cloud registration categories, challenges, and models.

| Category | Challenges | Methods |
|----------|-----------|---------|
| Classical | low overlap | (Stechschulte et al., 2019), (Denayer et al., 2024) |
| | severe outliers | (Bustos & Chin, 2017), (Wang et al., 2024b) |
| | poor initial poses | (Zhou et al., 2016), (Di Lauro et al., 2024) |
| Supervised | low overlap | RORNet (Wu et al., 2023c), PARE-Net (Yao et al., 2024) |
| | noise | Buffer (Ao et al., 2023), (Wang et al., 2024g) |
| Unsupervised | low overlap | UDPReg (Mei et al., 2023), Regiformer (Zheng et al., 2024a) |
| | partial observations | PointMBF (Yuan et al., 2023), GTINet (Jiang et al., 2024c) |
| | large-scale scenes | Regiformer (Zheng et al., 2024a), GTINet (Jiang et al., 2024c) |

### 4.4.1 Conclusion, Challenges, and Future Directions of Registration

Point cloud registration aims to align two 3D point clouds into a common coordinate system and remains essential for many 3D vision applications. Classical methods such as ICP are simple and efficient when point clouds have sufficient overlap, limited noise, and good initial alignment, while learning-based methods can handle more challenging cases by learning robust features from data. However, registration is still difficult under low overlap, noise, outliers, partial observations, and poor initial poses. Classical methods often fail in these conditions, whereas learning-based methods may require large training datasets and high computational resources. Future research should therefore focus on developing registration methods that are more accurate, efficient, robust to real-world noise and low-overlap scenarios, less dependent on labeled data, and easier to interpret and deploy in large-scale 3D scenes.

### 4.5  6DoF Pose Estimation

In this section, we discuss pose estimation in 3D space. For rigid objects, pose estimation usually aims to recover the object's spatial transformation with respect to the camera coordinate frame, whereas human pose estimation is commonly formulated in terms of articulated joints, body structure, or mesh recovery (Liu et al., 2024b). We divide this topic into two main parts: object pose estimation and human pose estimation. In the following, we explore and discuss these two topics in detail.

#### 4.5.1  Object Pose Estimation

Object pose estimation involves recovering an object's 3D translation and rotation with respect to the camera coordinate frame, which is commonly formulated as a rigid 6-DoF pose estimation problem in SE(3) (Xiang et al., 2017). In many applications, once an object has been identified, we need to estimate its precise pose. Depending on the formulation, the task may also include estimating object size or scale, especially in category-level pose-and-size estimation settings. However, point clouds generated from depth images or LiDAR are often noisy, and real-world settings typically provide only a partial point cloud, which is one of the main challenges (Wang et al., 2019a). Point density may vary across different parts of the object, and occlusion can cause missing regions (He et al., 2020b). In some cases, geometric or rotational symmetries of the object can also make pose estimation ambiguous (Hodan et al., 2020). In this survey, we organize object pose estimation methods into three groups: instance-level, category-level, and unseen or novel object-level methods. In the following, we describe each of these categories.

- **Instance-level:** In instance-level settings, the goal is to estimate the pose of a known object instance. For example, if the target object is a cup, the goal is to estimate the pose of one specific cup instance rather than to handle all possible cups. Many works in this area assume known object instances and learn instance-specific RGB-D or point-cloud cues for accurate pose estimation (He et al., 2021b). Some methods also reduce dependence on real annotated data by synthesizing point-cloud segments from 3D object models (Gao et al., 2021).

- **Category-level:** In category-level settings, the goal is to estimate the pose of objects within a given category. The primary challenge is intra-category variation, namely differences in shape and appearance among objects belonging to the same class (Zou et al., 2024a). A common strategy to address this issue is to map objects into a shared canonical space or normalized object coordinate representation, where 9-DoF pose-and-size estimation can be performed by estimating rotation, translation, and scale (Wang et al., 2019b). Other category-level methods further improve robustness by learning stronger shape, geometric, or contextual priors across object instances (Chen et al., 2020b; Li et al., 2025h).

- **Unseen object-level:** In unseen or novel object-level settings, the target instance has not been observed during training, and in some cases the test objects may also lie outside the training categories. A CAD model or object model is often assumed to be available at test time, although some recent settings relax this requirement by using reference observations or model-free formulations (Wen et al., 2024; Lee et al., 2025c). At this level, the emphasis is on generalization and robustness to novel objects, clutter, occlusion, and limited supervision (Cai et al., 2022b; Chen et al., 2024a; Caraffa et al., 2024). Because these methods relax the strong instance-specific assumptions used by traditional instance-level methods, they usually involve a trade-off between generalization and instance-specific accuracy.

#### 4.5.2  Human Pose Estimation

Human pose estimation aims to localize and model the articulated configuration of the human body by estimating the spatial positions of key joints or body parts from visual data, typically in 2D or 3D (Kappan et al., 2026; Galaaoui et al., 2025). Unlike object pose estimation, which focuses on recovering a rigid or piecewise-rigid transformation of an object in space, human pose estimation addresses a highly non-rigid and deformable structure with complex kinematic constraints, self-occlusion, and large intra-subject

motion variability. This distinction makes human pose estimation fundamentally more challenging and requires specialized representations and learning strategies. The task is central to numerous applications, including human-computer interaction, motion capture, action recognition, animation, sports analytics, and healthcare. Comprehensive surveys have reviewed the evolution of human pose estimation methods from classical pictorial structure models to modern deep learning approaches (Liu et al., 2024e; Kappan et al., 2026).

Existing human pose estimation methods that operate on point cloud inputs can be broadly categorized into three main methodological paradigms, reflecting the treatment of geometric structure, temporal information, and model priors. These categories are summarized in Table 9. **body-scanned point clouds** learn representations directly from raw 3D points and regress human joint locations (Zhou et al., 2020b; Chen et al., 2022b; Ballester et al., 2025) and **liDAR point clouds** are designed for real-time applications with sparse, irregular point clouds and adapt networks to LiDAR scanning characteristics (Weng et al., 2023; Kovács et al., 2024; Ye et al., 2024b; Zhang et al., 2024g; An et al., 2025a). Finally **joint pose and body shape estimation approaches** estimate both human joint positions and full-body meshes from unstructured point clouds using cascaded or model-fitting frameworks (Ren et al., 2024b; Cai et al., 2023).

Table 9: Point Cloud-Based Human Pose Estimation.

| Category | Methods |
|---|---|
| Body-Scanned Point Clouds | SPiKE (Ballester et al., 2025), (Zhou et al., 2020b), (Chen et al., 2022b) |
| LiDAR-based Point Clouds | LidPose (Weng et al., 2023), (Kovács et al., 2024), LPFormer (Ye et al., 2024b), LiDARCapV2 (Zhang et al., 2024g), DAPT (An et al., 2025a) |
| Joint Pose and Body Shape Estimation | LiveHPS (Ren et al., 2024b), PointHPS (Cai et al., 2023) |

### 4.5.3 Conclusion, Challenges, and Future Directions of 6DoF Pose Estimation

Pose estimation aims to determine the position and orientation of objects or humans in 3D space and is important for many 3D vision applications. Object pose estimation typically focuses on rigid objects by estimating their rotation and translation, while human pose estimation is more challenging because the human body is articulated and can take many different configurations. Despite recent progress, pose estimation remains difficult under noise, occlusion, partial observations, and missing data. Object pose estimation is further complicated by shape variation and object symmetries, whereas human pose estimation faces challenges such as body motion, self-occlusion, and sparse LiDAR measurements. Future research should focus on developing methods that are more accurate, robust in real-world scenes, generalizable to unseen objects and complex human poses, less dependent on large labeled datasets, and efficient enough for real-time deployment.

## 5 Foundation Models and Scene Understanding

Foundation models and scene understanding are discussed together because both move point cloud research beyond conventional geometry-focused perception toward semantic and multimodal 3D understanding. This section reviews foundation models (Section 5.1) and scene understanding (Section 5.2), emphasizing how point clouds are increasingly integrated with images, language, and large pretrained models for open-vocabulary recognition, grounding, captioning, question answering, and reasoning.

### 5.1 Foundation Models

Foundation Models (FMs) are large-scale pretrained models that learn general-purpose representations from diverse data and can be reused across tasks, domains, and datasets with minimal task-specific fine-tuning.

Rather than being optimized for a single objective, they capture broad structural, semantic, or reasoning priors that transfer effectively to downstream problems (Deng et al., 2025c).

In the context of 3D vision, the concept of FMs extends beyond models trained directly on 3D data to include those pretrained on 2D images and language, whose learned representations can be leveraged for 3D perception, representation learning, and reasoning. Consequently, the FM paradigm has reshaped how 3D vision systems are designed, even in domains traditionally driven by geometry. Three classes of FMs have become particularly influential in computer vision and point cloud research.

Vision Language Models (VLMs) jointly model visual and textual information, enabling open-vocabulary perception, semantic grounding, and cross-modal alignment (Xu et al., 2023c; Luan et al., 2025). These models have become central to modern vision systems as language emerges as a dominant interface for supervision, querying, and semantic abstraction. Large Language Models (LLMs) provide powerful capabilities for reasoning, planning, and task decomposition; in 3D vision pipelines, LLMs are increasingly used to reason over structured perceptual outputs, direct multi-stage systems, and enable flexible interaction with 3D environments through language (Xu et al., 2025b). Vision Foundation Models, conversely, offer strong visual and geometric priors without involving language. Large-scale visual models for segmentation, feature extraction, or class-agnostic perception provide transferable cues for objectness, boundaries, and structure, and have become increasingly influential in recent 3D pipelines (Jung et al., 2025a).

Together, these FMs form the backbone of current approaches that integrate point clouds with large-scale pretrained knowledge. As FMs have become dominant across vision and language, a growing body of work has explored how point clouds interact with FM pipelines. While prior surveys have documented individual techniques or model classes, they have largely focused on cataloging applications of 2D or language FMs to 3D tasks (Ma et al., 2024b; Thengane et al., 2025; Shao et al., 2025; Liu et al., 2025g). As a result, the relationship between point clouds and FMs has remained conceptually fragmented.

This section addresses how foundation knowledge flows with point cloud representations through two usage categories: Point Cloud Native (PC-Native) approaches, where point clouds serve as the primary modality for input, representation learning, or reasoning; and Point Cloud Scaffolded (PC-Scaffolded) approaches, where point clouds provide geometric context as intermediate structures. Table 10 presents a comprehensive taxonomy of 3D foundation models.

### 5.1.1 Representation-Defining Foundation Models

In these works, the FM is the representation learner. The models are designed to intrinsically generate dense 3D representations directly from raw data, such as multi-view images, in a unified feed-forward pass.

**Geometry-centric 3D Foundation Models (GFMs)**  GFMs have emerged from the success of large-scale foundation models in 2D vision and language. The main goal of these predictive, end-to-end models is to generate dense 3D representations such as depth maps or point maps, from multi-view image inputs.

Their key innovation is to eliminate dependence on camera parameters and replace complex, multi-step reconstruction pipelines with a fast, unified feed-forward process. DUSt3R (Wang et al., 2024e), MUSt3R (Cabon et al., 2025), Fast3R (Yang et al., 2025b), and VGGT (Wang et al., 2025f) are examples of GFMs that focus on transformer-based feed-forward architectures, enabling the parallel processing of multiple views. These methods infer a common, rich representation of the scene that encodes the 3D appearance features and perspective relationships simultaneously. This unified representation then serves as a foundation for all desired output options including camera parameters, depth maps, point maps, novel view synthesis, and point tracks. However, challenges such as acceptable accuracy, generalization to different cases, and efficiency under real-time conditions provide the basis for the development of more comprehensive evaluation criteria (Cong et al., 2025; Leblanc & Poullis, 2026).

**Point Cloud Adapted Vision-based Foundation Models**  These approaches overcome traditional limitations in depth estimation and scene understanding, including expensive hardware (e.g., LiDAR) and poor generalization in pure vision-based methods. They achieve this by utilizing large-scale pre-trained models with robust transfer capabilities (Xu et al., 2026c). Frameworks such as Seal (Liu et al., 2023e) and

Table 10: A comprehensive taxonomy of 3D Foundation Models.

| Category | SubCategory | Methods |
|---|---|---|
| Representation-Defining FMs | Geometry-centric 3D FMs (GFMs) | DUSt3R (Wang et al., 2024e), MUSt3R (Cabon et al., 2025), Fast3R (Yang et al., 2025b), VGGT (Wang et al., 2025f), E3D-Bench (Cong et al., 2025), Distill3R (Leblanc & Poullis, 2026) |
| | Point Cloud Adapted Vision-based FMs | Seal (Liu et al., 2023e), Uni-Adapter (Tamjidi et al., 2025), UniQA-3D (Zuo et al., 2025) |
| | 3D Vision-Language FMs | AiDe (Wang & Czarnecki, 2025), 3D-LLaVA (Deng et al., 2025a), GreenPLM (Tang et al., 2025b), FoundationMorph (Pan et al., 2025b), TotalFM (Yamamoto & Kikuchi, 2026), ProxyTransformation (Peng et al., 2025), LF-3DVG (Wu et al., 2025b), FAC++ (Liu et al., 2025b) |
| | Action- or Embodiment-Centric 3D FMs | Pointvla (Li et al., 2026a), 3D CAVLA (Bhat et al., 2025), Geovla (Sun et al., 2025a), Spatialvla (Qu et al., 2025a), Evo-0 (Lin et al., 2025) |
| Representation-Guiding FMs | - | GFS-VL (An et al., 2025b), EAIL (Zhang et al., 2025c), PCN-TI (Song et al., 2023c), Ges3ViG (Mane et al., 2025), Text2Loc++ (Xia et al., 2025b), GSP (Chen et al., 2025a) |
| | 2D VLMs As Semantic Priors | SemAbs (Ha & Song, 2022), Fine-tuning with UAPs (Rudner et al., 2024) |
| | Vision FMs As Semantic Priors | BALViT (Hindel et al., 2025), Partslip (Liu et al., 2023d), Vipocc (Feng et al., 2025a), HFIT (Guo et al., 2026b) |
| Reasoning-Centric FMs | - | LARM (Li et al., 2025o), ZeroKey (Gong et al., 2025), Reason3D (Huang et al., 2025c), 3D-AffordanceLLM (Chu et al., 2025), SeqAfford (Yu et al., 2025a), LiDAR-LLM (Yang et al., 2025d), RadarLLM (Lai et al., 2025), PULLM (Zhang et al., 2025i), 3D-R1 (Huang et al., 2025e) |
| Point Cloud-Scaffolded FMs | - | ScanReQA (Zhang et al., 2025f), 3D-LOTUS++ (Garcia et al., 2025), Gpt4scene (Qi et al., 2025), VisionCube (Wang et al., 2025b) |

imaging-based methods of 2D projections into 3D space enable segmentation and point cloud understanding with high scalability, stability, and generalizability (Dong et al., 2025). However, 3D vision-language-based models often require adaptive strategies such as Uni-Adapter (Tamjidi et al., 2025) to improve robustness when faced with noisy or out-of-distribution data. Despite these advances, evaluations such as UniQA-3D (Zuo et al., 2025) show that these models are still far from achieving robust 3D reasoning on par with human understanding, pointing the way for future research towards developing deeper and more reliable foundation models for 3D vision.

**3D Vision-Language Foundation Models** In this category, language participates directly in representation learning. Point cloud features are explicitly aligned with language through contrastive, distillation (Hinton et al., 2015; Mansourian et al., 2025), or alignment objectives (Wang & Czarnecki, 2025; Deng et al., 2025a; Tang et al., 2025b). The learned embedding space is therefore language-aware. Removing language during training would fundamentally change the representation, indicating that language is part of the representation itself. This foundational language-aligned 3D representation enables a suite of sophisticated downstream tasks, including medical image analysis (Pan et al., 2025b; Yamamoto & Kikuchi, 2026), visual grounding (Peng et al., 2025; Wu et al., 2025b; Liu et al., 2025a), and robotic interaction (Liu et al., 2025b), where language serves as the direct interface for querying, controlling, or reasoning about the 3D world.

**Action or Embodiment-Centric 3D Foundation Models** These models have been developed to overcome the inherent limitations of Vision-Language-Action (VLA) models in understanding spatial relation-

ships and the geometry of the physical world. They improve spatial reasoning and generalizability in robotic tasks by injecting 3D representations into existing architectures, such as low-perturbation point cloud injection (Li et al., 2026a), depth awareness and chain reasoning (Bhat et al., 2025), independent geometry encoder (Sun et al., 2025a), Ego3D position encoding and Adaptive Action Grids (Qu et al., 2025a), and implicit depth perception without additional sensors (Lin et al., 2025).

### 5.1.2 Representation-Guiding Foundation Models

3D FMs improve their ability to understand and represent 3D spaces by leveraging the rich knowledge of pretrained VLMs. In this regard, various approaches have been developed to guide 3D representations. For example, the GFS-VL framework (An et al., 2025b) improves point cloud segmentation by combining the knowledge of a new class of 3D vision-language models (3D VLMs) with accurate multi-shot samples. Some studies emphasize the use of 3D representation as a spatial anchor for other data modalities (Zhang et al., 2025c). In point cloud completion, Song et al. (2023c) and Zhou et al. (2025a) both use the CLIP (Radford et al., 2021) model to guide the geometric reconstruction of incomplete shapes. Mane et al. (2025) introduces a new approach in which human pointing gestures, in addition to linguistic descriptions, guide the model toward the target, significantly improving recognition accuracy. Similarly, the hierarchical architecture in Text2Loc++ uses natural language descriptions for localization in point clouds, enabling effective alignment between language and 3D spaces (Xia et al., 2025b). Finally, the training-free approach proposed by Chen et al. (2025a) improves the efficiency of vision-language models for detecting out-of-distribution data in 3D spaces. However, in all these works, FMs play a guiding role in the perception process rather than defining the main point cloud representation themselves. As a result, the point cloud encoder remains unimodal.

**2D VLMS As Semantic Priors** Here, language acts as a semantic prior, typically at inference time. The foundation model is frozen, the point cloud encoder is trained for a specific task, and no attempt is made to align point cloud representations with language. In this setting, language functions as external guidance rather than as a representational component (Ha & Song, 2022; Rudner et al., 2024).

**Vision Foundation Models As Semantic Priors** Since collecting 3D data with accurate labels is expensive and time-consuming, researchers have developed methods to transfer knowledge to the 3D domain to reduce the need for large-scale annotations. For example, in point cloud segmentation, predictions from these models are mapped to 3D space through pooling and pseudo-label generation (Dong et al., 2025). The BALViT architecture proposed by Hindel et al. (2025) uses frozen image models as encoders and combines them with LiDAR data via 2D-3D adapters to create robust representations. For object part segmentation, knowledge from the GLIP vision-language model is transferred to point clouds through recognition on 3D renderings and label transfer (Liu et al., 2023d). In 3D structure prediction from images, relative depth information obtained from these models is converted into metric depth by a depth alignment module to improve accuracy (Feng et al., 2025a). The HFIT architecture proposed by Guo et al. (2026b) also improves performance in driving scenes by extracting and merging heterogeneous features from RGB and depth data without requiring Vision Transformers to be retrained.

### 5.1.3 Reasoning-Centric Foundation Models

The integration of LLMs with 3D point cloud vision represents a shift toward reasoning-centric systems. In these approaches, the foundation model operates on top of the point cloud system; it does not modify or learn point-level features directly. Instead, it functions as a high-level reasoning engine that interprets semantic concepts and user intentions, while specialized encoders handle geometric details. This enables machines to perceive 3D structures while reasoning about physical scenes using human language and commonsense knowledge. Recent works demonstrate that LLMs can augment point cloud understanding in several critical ways. Models like LARM (Li et al., 2025o) and ZeroKey (Gong et al., 2025) leverage the vast prior knowledge embedded in LLMs to enrich category prototypes and detect salient keypoints, overcoming the limitations of scarce 3D training data. Frameworks such as Reason3D (Huang et al., 2025c), 3D-Affordance LLM (Chu et al., 2025), and SeqAfford (Yu et al., 2025a) reformulate segmentation and affordance detection tasks into instruction-reasoning problems, enabling models to comprehend complex user intentions and decompose

them into actionable segmentation masks. This reasoning capability is further extended to challenging real-world data by systems such as LiDAR-LLM (Yang et al., 2025d) and RadarLLM (Lai et al., 2025). This reasoning-centric paradigm extends even to foundational 3D tasks. For instance, PULLM (Zhang et al., 2025i) reimagines point cloud upsampling by leveraging LLMs to guide the preservation of fine geometric details. Meanwhile, frameworks like 3D-R1 (Huang et al., 2025e) push the boundaries further by employing reinforcement learning and dynamic view selection to enhance the reasoning and generalization capabilities of 3D VLMs. Together, these works point to a clear direction: the future of point cloud vision depends on models that combine precise 3D perception with the reasoning abilities of language.

### 5.1.4 Point cloud Scaffolded Foundation Models

In these approaches, point clouds are typically reconstructed or employed to provide the necessary geometric context, although their role is that of an external structural input than an original representation learned or generated by the model itself. Zhang et al. (2025f) critically questions whether point clouds truly enhance spatial reasoning in LLMs, challenging their assumed value. In contrast, 3D-LOTUS++ (Garcia et al., 2025) introduces a framework that integrates point cloud-based motion planning with language and vision models to improve robotic manipulation. However, Qi et al. (2025) proposes a purely vision-based alternative, suggesting that point clouds may not be necessary for 3D scene understanding. Meanwhile, VisionCube (Wang et al., 2025b) demonstrates that structured 3D representations reconstructed from images can effectively scaffold complex multi-step spatial reasoning tasks.

### 5.1.5 Conclusion, Challenges, and Future Directions of Foundation Models

Foundation models have profoundly reshaped 3D vision, yet the field remains in a formative stage. While representation-defining approaches such as GFMs and 3D Vision-Language FMs have begun to integrate geometry and semantics directly, they still face challenges in scalability, generalization, and robustness across sensor modalities. A further bottleneck is the computational cost of large VLMs and LLMs, which limits real-time deployment on edge devices and robots. Moreover, generalization across diverse sensor types (LiDAR, RGB-D, stereo) and unseen environments still lags behind that of 2D foundation models. Future progress will require more unified 3D foundation models that learn geometry and semantics jointly from large-scale multimodal data, complemented by rigorous benchmarks that test spatial reasoning, embodiment, and out-of-distribution robustness. Developing lighter, sample-efficient architectures will be critical to transitioning from laboratory demonstrations to reliable, deployed systems in robotics, and augmented reality.

### 5.2 Scene Understanding

3D scene understanding aims to extract semantic, relational, and contextual knowledge from three-dimensional environments by jointly reasoning over geometry and high-level concepts. In this survey, we organize 3D scene understanding tasks into four complementary categories: 3D captioning 5.3, 3D question answering (3DQA) 5.5, 3D grounding 5.4, and 3D reasoning 5.6. These tasks differ in their output structure and reasoning requirements, ranging from holistic scene-level description (3D captioning) to interactive query-based understanding (3DQA), precise language-conditioned localization (3D grounding), and multi-step inference over spatial, semantic, or physical relationships (3D reasoning). This categorization provides a structured view of the field by grouping tasks according to the form of language supervision and the depth of reasoning involved, enabling clearer comparison of methodologies and facilitating analysis of how different models trade off expressiveness, interpretability, and computational complexity across 3D scene understanding problems. An overview of this taxonomy of 3D scene understanding tasks is illustrated in Fig 6.

### 5.3 Captioning

3D captioning combines 3D visual data with natural language. The model takes a 3D point cloud representing the appearance of a scene and automatically generates text descriptions for objects in that scene. 2D captioning surveys are generally categorized based on deep learning methods (Hossain et al., 2019; Stefanini

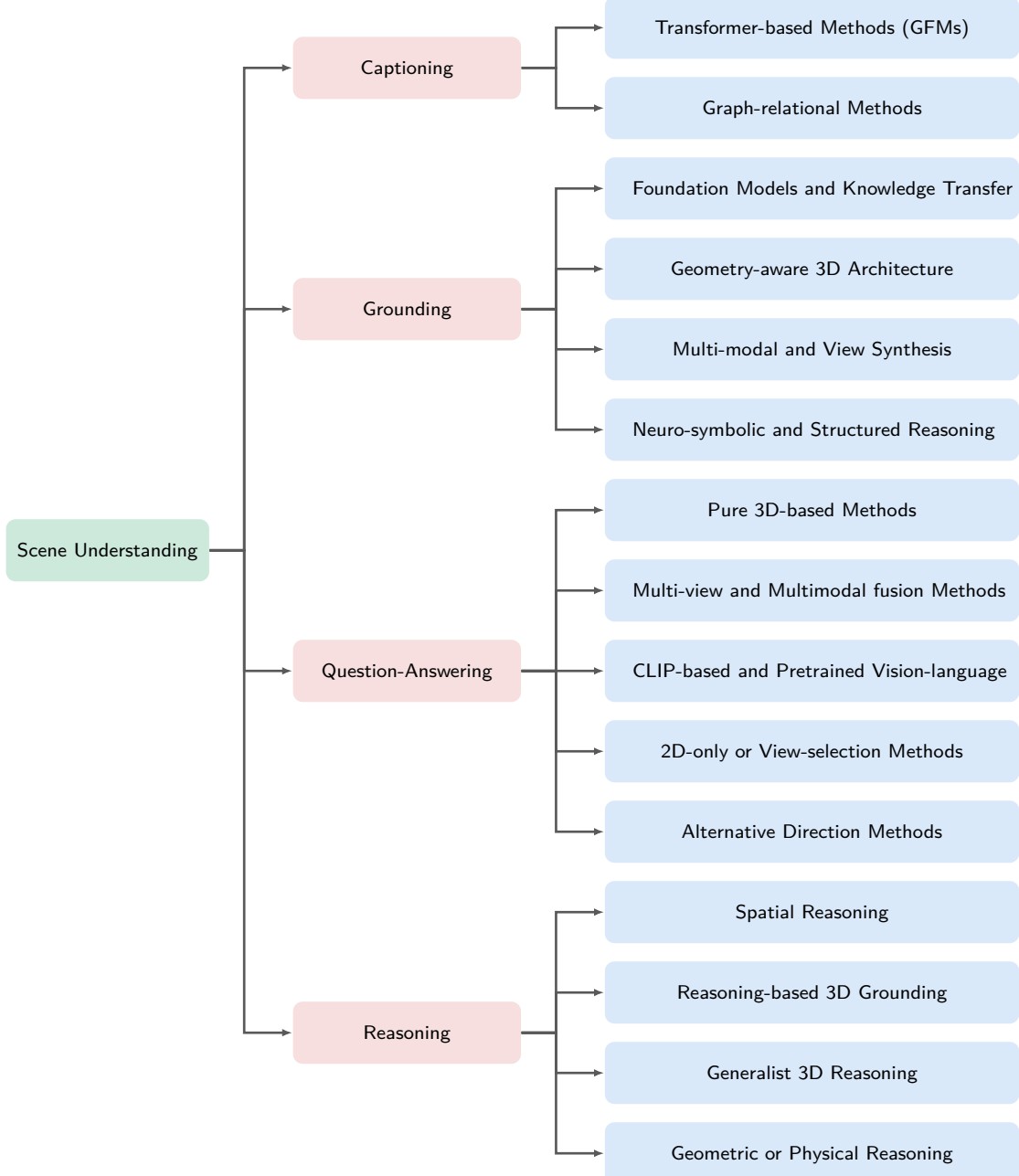

Figure 6: Taxonomy of 3D Scene Understanding methods.

et al., 2022; Ghandi et al., 2023). In contrast, 3D captioning focuses more on applications and tasks related to dense captioning, and the categorization provided by (Yu et al., 2023c) is too general. For this reason, in this paper, two main groups of articles in this field are categorized based on their architecture and methodology, and a more detailed analysis is provided based on each of the methodologies.

**Transformer-based methods** in 3D captioning have been developed in three main directions: first, integrated detection and description approaches such as Vote2Cap-DETR (Chen et al., 2023b) and ToD3Cap (Jin et al., 2024a) that perform localization and caption generation operations in parallel with a query-based architecture. Second, context-aware and geometry-aware transformers (Wang et al., 2022a; Kim et al., 2024b) that model the relationships between objects and scene structure with specialized attention mechanisms.

Third, multimodal and pretrained transformers such as CG-MLLM (Huang & Xu, 2026), X-Trans2Cap (Yuan et al., 2022), 3D-VisTA (Zhu et al., 2023b), and 3D CoCa (Huang et al., 2025e; Tang et al., 2026a) that reduce the gap between 3D representation and language by transferring knowledge from the 2D domain and adversarial alignment. In addition, advanced methods like Generative Adversarial Networks (GAN) have also been applied in the transformer framework to increase the quality, variety, and realism of the produced captions (Luo et al., 2023b; Yu et al., 2023b).

**Graph-relational methods** such as the studies by (Chen et al., 2021b) and (Jiao et al., 2022) directly used graph neural networks (GNNs) and structured scene graphs to model object relationships, a base idea that has even been applied to tasks such as 3D caption changing (Liao et al., 2021). Next, research moved towards attention-based relational models (Kim et al., 2024b) that extract relationships without the need to explicitly construct graphs. A newer generation consists of hybrid models (Cai et al., 2022a) that combine graph-based reasoning with transformer attention mechanisms; for example, graph networks that exploit multiple 2D views to improve dense captioning (Ma et al., 2025d). This evolution has led to more flexible and powerful relational modeling (Jia et al., 2023). Such hybrid frameworks provide a solid foundation for relational reasoning, which has recently been enhanced with advanced diffusion models (Luo et al., 2023a; Cioni et al., 2023; Luo et al., 2024a; Shu et al., 2025) to increase the detail and quality of the output by gradually converting the scene representation into captions.

### 5.3.1 Conclusion, Challenges, and Future Directions of Captioning

3D captioning has evolved impressively, but it still faces real challenges. Dense captioning in cluttered scenes remains error-prone, most methods struggle with fine-grained spatial relationships, and transferring 2D language priors to 3D often introduces domain gaps. Scalability is also an issue, as current benchmarks are limited in scene diversity and annotation richness. Looking ahead, we need more lightweight, geometry-aware models that work in real time, better integration of multiview and RGB-D data, and stronger evaluation metrics that go beyond n-gram matches. Exploring large 3D-language foundation models and few-shot adaptation to novel object categories would also be valuable. These steps will help move 3D captioning from research prototypes to practical, human-friendly scene understanding systems.

### 5.4 Grounding

Visual grounding (Rohrbach et al., 2016; Mao et al., 2016; Li et al., 2022b; 2023k; Wang et al., 2025l; Li et al., 2025n) is a central task in the field of integrating computer vision and natural language processing. In 3D environments, the goal is to identify and accurately locate a specific object in a complex scene, based solely on its linguistic description. Because of these capabilities, visual grounding is considered an essential intermediate component, enabling and facilitating many more advanced systems. Broad applications such as comprehensive understanding of 3D scenes (Man et al., 2024b; Zhi et al., 2025; Ost et al., 2026), creating interactive virtual or augmented reality systems (Gopakumar et al., 2024; Jang et al., 2024; Zhai et al., 2025) and controlling robots with natural language commands (Huang et al., 2025a; Tang et al., 2025a) are among the areas that directly benefit from advances in this field.

This review organizes the developing literature in 3D grounding by categorizing recent studies according to their core methodological approaches. We have grouped the leading work into four principal paradigms, providing a structured analysis of their fundamental technical strategies.

**Foundation models and knowledge transfer** directly leverages pretrained 2D Vision-Language Models (VLMs) or Large Language Models (LLMs) to enhance the 3D visual grounding pipeline. These approaches eliminate the costly process of training on massive 3D datasets from scratch. Instead, they adapt rich visual-language representations by transferring knowledge from foundation models pretrained on vast internet-scale 2D image datasets (Bakr et al., 2022; Wang et al., 2023h), employing LLMs as semantic reasoners (Yang et al., 2024a), or using foundational models as robust feature extractors (Chen et al., 2024c; Liu et al., 2025i). **Geometry-aware 3D architectures** tackle the geometric, structural, and sparsity challenges of point cloud data, including transformer adaptations (Zhao et al., 2021b; He et al., 2021a; Chen et al., 2023h), novel attention modules (Li et al., 2025e; Zhang et al., 2025h; Dey et al., 2025), and enhanced relational encoders (Chen et al., 2020a; Yuan et al., 2021; Xiao et al., 2024; Wang et al., 2025m) that produce robust 3D

feature representations well suited to grounding tasks. **Multimodal and view synthesis** methods combine and enrich the 3D point cloud principle with information from other sensory methods to achieve stronger groundings. This additional information can include 2D images (Lyu et al., 2024a; Xiao et al., 2025a), multi-view projections (Huang et al., 2022b; Guo et al., 2023; Geng et al., 2025), segmentation masks (Chen et al., 2024b), and CLIP embeddings (Hegde et al., 2023; Zhang et al., 2024m; Li et al., 2025c). This method combines 3D geometric information with other semantic and visual data to create a better understanding of the scene. As a result, both object detection accuracy and consistency across different viewpoints are improved. **Neuro-symbolic and structured reasoning** integrates neural networks with symbolic logic or structured knowledge bases, such as spatial commonsense, which is an emerging direction for complex reasoning. This hybrid method facilitates explicit reasoning, allowing models to compositionally understand the elements within a 3D scene. This capability is essential for tasks like interpreting long-horizon, logically constrained linguistic queries and ensuring verifiable predictions. (Zellers et al., 2021; Chen et al., 2022c; Hsu et al., 2023; Zhu et al., 2024a; Jahangard et al., 2025; Chen et al., 2025d; Huang et al., 2026a). It addresses key limitations of purely data-driven models, such as their poor generalization to new scenarios and lack of interpretability.

### 5.4.1 Conclusion, Challenges, and Future Directions of Grounding

Current 3D grounding techniques demonstrate a strong capacity for precise object localization by jointly reasoning about geometric relationships and distinguishing visual features. However, future advances depend on solving challenges such as handling noisy 3D data, achieving more advanced linguistic understanding and reasoning, and defining metrics that better capture the complexity of real-world environments.

## 5.5 Question Answering

Recently, 3D question answering (3DQA) has emerged as a significant research topic, as it involves answering questions about 3D scenes. Unlike 2D question answering (Antol et al., 2015; Kim et al., 2025), which relies on 2D images, 3DQA captures richer spatial relationships between objects. In recent years, several datasets have been introduced for this task (Etesam et al., 2022; Azuma et al., 2022; Zhao et al., 2022; Yan et al., 2023b; Qian et al., 2024; Szymańska et al., 2024; Guan et al., 2026; Ishihara et al., 2026). 3DVQA-ScanNet (Etesam et al., 2022) was the first dataset proposed for 3DQA. Later, ScanQA (Azuma et al., 2022) introduced a new dataset and an end-to-end model for the task that jointly encodes the 3D scene and the question, and is trained with a combination of classification, grounding, and answer prediction losses. FE-3DGQA (Zhao et al., 2022) contributed a large-scale 3D grounded VQA dataset with free-form questions. More recently, CLEVR3D (Yan et al., 2023b) introduced a compositional scene-manipulation strategy to generate questions from augmented 3D scenes; NuScenes-QA (Qian et al., 2024) proposed the first multimodal 3D VQA dataset for driving scenes; Space3D-Bench (Szymańska et al., 2024) focused on questions regarding spatial relationships among indoor objects; and City-3DQA (Sun et al., 2024a) proposed a 3D multimodal question-answering benchmark for city-level scene understanding.

With the growth of annotated benchmarks for this task, several new methods have been proposed to address 3D question answering (Ye et al., 2022; Luo et al., 2025; Zhou et al., 2025d; Zhang et al., 2026b; Xu et al., 2026a). 3DQA-TR (Ye et al., 2022) introduced the first transformer-based approach, fusing appearance, geometry, and linguistic features. DSPNet (Luo et al., 2025) integrates multi-view and point cloud features, while HCNQA (Zhou et al., 2025d) incorporates supervision signals that include intermediate reasoning checkpoints on the path to the final answer. In addition, various works have explored alternative directions for 3DQA, including state-space models (Qorsham et al., 2025), Gaussian splatting (Thai et al., 2025), active learning (Zhou et al., 2025c), and self-supervised learning (Li et al., 2023h).

Furthermore, building on the advances in 2DQA, several methods have been proposed to perform 3DQA using existing 2D question-answering models. Some approaches attempt to bridge the gap between 2D and 3D by selecting diverse views from the 3D scene and fusing 2D and 3D modalities (Mo & Liu, 2024; Guo et al., 2025b). Other methods leverage CLIP by training 3D variants of CLIP for 3DQA (Parelli et al., 2023; Delitzas et al., 2023), while additional works investigate the zero-shot capabilities of existing

2D vision-language models when applied to 3D scenes (Singh et al., 2024; Huang et al., 2025b; Wang et al., 2025c).

### 5.5.1 Conclusion, Challenges, and Future Directions of Question Answering

In summary, 3DQA approaches can be broadly distinguished by how they represent and reason over 3D scenes. Pure 3D-based methods explicitly model geometry and spatial structure, making them well suited for questions requiring precise localization or spatial relationships, but they often suffer from limited scalability and high annotation cost. Multi-view and multimodal fusion methods improve robustness by combining 2D appearance and 3D geometry, achieving better contextual understanding at the expense of increased system complexity and data requirements. CLIP-based and pretrained vision-language approaches offer strong semantic generalization and reduced training cost by leveraging large-scale 2D pretraining, though they may lose fine-grained 3D spatial accuracy. 2D-only or view-selection methods provide efficient and often zero-shot solutions, but their performance depends heavily on view coverage and may struggle with occluded or geometry-intensive queries. Overall, method selection depends on the balance among spatial precision, semantic generalization, and computational efficiency required by the target application.

## 5.6 Reasoning

3D reasoning refers to the capability of a model to infer non-trivial properties, relations, or outcomes in three-dimensional environments that are not directly observable from raw geometric input, such as point clouds or multi-view observations (Zhu et al., 2024a). Unlike standard 3D perception tasks that focus on recognition or localization, 3D reasoning requires compositional, multi-step inference over spatial relations, semantic concepts, geometry, or physical constraints. Depending on the nature of the inferred knowledge and the role reasoning plays in the task, existing works can be broadly categorized into four main subcategories: spatial reasoning, reasoning-based grounding, generalist 3D reasoning, and geometric or physical reasoning.

**Spatial reasoning** focuses on inferring relative spatial relationships among objects—such as distance, orientation, containment, or visibility often requiring relational or viewpoint-dependent inference beyond direct perception (Chen et al., 2022c; Hong et al., 2023a; Wang et al., 2023e; He et al., 2024; Ma et al., 2024a; Zhu et al., 2024e; Man et al., 2024a; Ma et al., 2025c;b; Zha et al., 2025; Chen et al., 2026). **Reasoning-based 3D grounding** uses compositional reasoning to decompose complex linguistic descriptions into semantic attributes and spatial constraints in order to produce structured 3D outputs, such as object instances or part-level masks, particularly in open-vocabulary settings (Zhu et al., 2024a; Kareem et al., 2024; Chen et al., 2024b; Sun et al., 2024a; Huang et al., 2025c; Feng et al., 2026). **Generalist 3D reasoning** aims to build unified 3D vision–language models capable of handling diverse tasks, such as question answering, grounding, and instruction following within a single framework, where reasoning is largely implicit and task-agnostic (Hong et al., 2023b; Jiang et al., 2024b; Fu et al., 2025; Huang et al., 2025e; Kang et al., 2025; Deng et al., 2025a; Zhu et al., 2025; Xue et al., 2026; Yang et al., 2026). In contrast, **geometric or physical reasoning** emphasizes inference grounded in 3D geometry or physical principles, such as shape, pose, stability, support, occlusion, or physical feasibility, often relying on explicit geometric representations or physics-inspired constraints (Chaudhuri et al., 2011; Jia et al., 2013; Schwing et al., 2013; Suwajanakorn et al., 2018; Qin et al., 2019b; Liu et al., 2019b). Table 11 summarizes representative works in each category.

### 5.6.1 Conclusion, Challenges, and Future Directions of Reasoning

In summary, different 3D reasoning paradigms address distinct requirements of scene understanding. Spatial reasoning is most effective for tasks dominated by explicit geometric relationships, offering strong interpretability but limited semantic flexibility. Reasoning-based grounding is better suited for compositional and open-vocabulary queries, as it explicitly aligns language with 3D structure, albeit at higher computational cost. Generalist 3D reasoning models provide broad task coverage and strong generalization through large-scale multimodal pretraining, though they often rely on implicit reasoning and may lack fine-grained geometric precision. Geometric and physical reasoning excels in tasks requiring accurate shape, pose, or physical feasibility, but is less applicable to language-driven or semantic reasoning tasks. Overall, the choice

Table 11: Summary of 3D reasoning methods.

| Category | Methods |
|---|---|
| Spatial reasoning | ViL3DRel (Chen et al., 2022c), 3D-CLR (Hong et al., 2023a), Super-CLEVR-3D (Wang et al., 2023e), LLM-TPC (He et al., 2024), SpatialPIN (Ma et al., 2024a), PQ3D (Zhu et al., 2024e), SIG3D (Man et al., 2024a), SpatialReasoner (Ma et al., 2025c), 3DSRBench (Ma et al., 2025b), CVP Chen et al. (2026) |
| Reasoning-based grounding | Scanreason (Zhu et al., 2024a), PARIS3D (Kareem et al., 2024), Reasoning3D (Chen et al., 2024b), Sg-CityU (Sun et al., 2024a), Reason3D (Huang et al., 2025c), CoRe (Feng et al., 2026) |
| Generalist reasoning | 3D-LLM (Hong et al., 2023b), MORE3D (Jiang et al., 2024b), Scene-llm (Fu et al., 2025), 3D-R1 (Huang et al., 2025e), Robin3D (Kang et al., 2025), 3D-LLaVA (Deng et al., 2025a), LLaVA-3D (Zhu et al., 2025), UniBVR(Yang et al., 2026), Descrip3D (Xue et al., 2026) |
| Geometric/Physical reasoning | (Chaudhuri et al., 2011), (Jia et al., 2013), Box-in-Box (Schwing et al., 2013), KeypointNet (Suwajanakorn et al., 2018), MonoGRNet (Qin et al., 2019b), SoftRas (Liu et al., 2019b) |

of category depends on whether the task prioritizes geometric precision, semantic flexibility, or multi-task generalization.

# 6 Robustness, Generalization, and Reliability

Robustness is a critical requirement for reliable 3D point cloud understanding systems operating in real-world environments. Due to sensor noise, partial observations, distribution shifts, and potential adversarial manipulation, 3D machine learning models must maintain stable and consistent performance under challenging conditions. This section examines robustness from multiple perspectives. It first discusses security threats in 3D ML pipelines, highlighting potential vulnerabilities and attack surfaces. It then analyzes robustness to noise and occlusion, which commonly arise in practical sensing scenarios. Next, rotation invariance and equivariance in point cloud processing are reviewed, as they are essential for handling arbitrary object orientations. Domain adaptation strategies are also explored to address distribution shifts across datasets and environments. Finally, anomaly detection is covered as a mechanism for identifying unexpected or out-of-distribution samples and improving system reliability.

## 6.1 3D ML Pipeline Security Threats

The pipeline of processing a 3D object, from the scanning point to processing it with a ML model, is susceptible to various types of security risks; Hence, there is a need for robustness in each section of this pipeline to different types of attacks or defending against them. To frame adversarial machine learning in 3D vision, it is useful to start from a pipeline-level view and then zoom into finer subcategories. Inspired by (Kim & Kaur, 2024), and as illustrated in Figure 7, threats can be grouped by where they occur in the pipeline: (i) attacks on sensors (e.g., cyber/physical LiDAR manipulation), and (ii) attacks on perception/decision models.

Compared to attacks on sensors, model-layer attacks are broader and more studied, and each branch comes with corresponding defenses as well. In the following sections, we firstly focus on the model-layer branch and summarize Evasion and Poisoning attacks for point cloud data type in sections 6.1 and 6.1 followed by their corresponding defenses for each section, then revisit the same pipeline-level view in the LiDAR setting in section 6.1. It is worth mentioning that, another type of attack called "Model Extraction/Model Theft" explored in the 2D literature to an admissible extent, to the best of our knowledge, appears under-explored for point cloud models unlike Evasion and Poisoning attacks and existing point cloud security literature rarely treats it as a first-class threat. However, dataset/model ownership verification and watermarking for point clouds are emerging and can support post-hoc attribution in potential model-stealing scenarios, which will be addressed in section 6.1.

**Evasion Attacks' Attributes and Methodologies** Evasion attacks, craft test-time inputs - called Adversarial Examples - that remain plausible to humans, but induce incorrect, often high-confidence predictions.

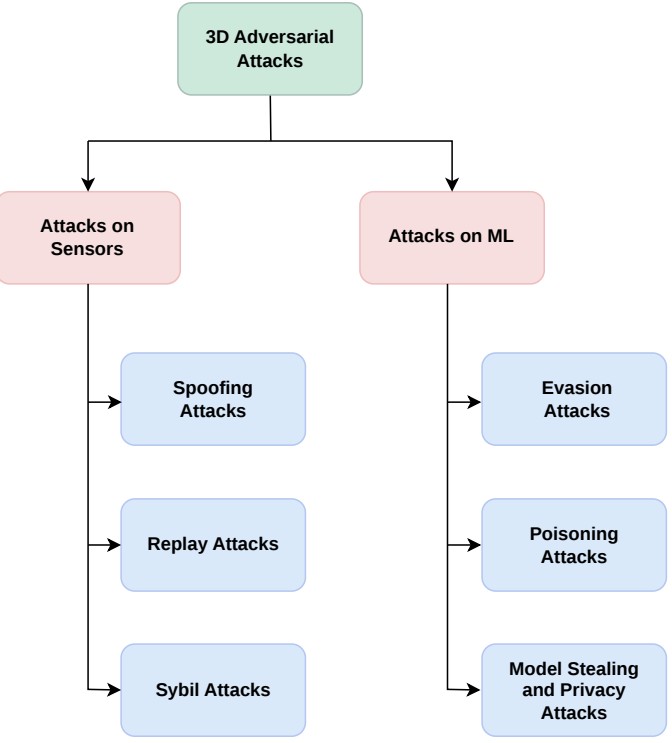

Figure 7: A high-level overview of 3D ML pipeline security threats.

First time demonstrated for 2D ConvNets by (Szegedy et al., 2013), adversarial examples were extended to 3D settings by (Liu et al., 2019a) and (Yang et al., 2019a). Building upon the taxonomies of (Naderi & Bajić, 2023) and (Li et al., 2024h), we can categorize adversarial attacks by attack attributes and attack methodologies, as discussed below:

- **Evasion Attack Attributes:** Each attack can have its own characteristics according to the threat model it is practiced in, such as the data representation type it is being applied on, the amount of adversarial knowledge the adversary holds, the type of target label strategy the attacker has, etc.

- **Evasion Attack Methodologies:** Based on the attack methodology, attacks can be categorized into optimization-based attacks (Lou et al., 2024; Tang et al., 2024a; Xiang et al., 2019; Naderi et al., 2026; Bu & Liu, 2026), generative-based attacks (Tang et al., 2023; Zhou et al., 2020a; Hamdi et al., 2020; Wang et al., 2026b), transform-based attacks (Liu et al., 2023a; Zhao et al., 2020b; Zhang et al., 2024h; Tang et al., 2026b), perturbation-based attacks (Liu & Hu, 2023; Huang et al., 2022a; Liu et al., 2020a), heuristic attacks (Cao et al.; Wicker & Kwiatkowska, 2019), simulation-based attacks (Tu et al., 2020b; Cao et al.), and physical system-based attacks (Cao et al., 2019; Sato et al., 2025).

**Evasion Defense Methodologies** Defenses against adversarial example attacks, can operate on two entities: data preprocessing (in either the train (Miao et al., 2024; Zhang et al., 2022a; Sun et al., 2021a) or test phase (Sun et al., 2023; Lin et al., 2024b; Zhou et al., 2019; Li et al., 2026b; Fan et al., 2026; Luo et al., 2026)), or modifying the model architecture or weights with different approaches (Zhang et al., 2023d; Chu et al., 2022; Li et al., 2022a; Huang et al., 2026c; Gui et al., 2026).

**Poisoning Attacks and Defenses**    Poisoning attacks, also known as Backdoor attacks, are training-phase attacks where an adversary poisons the learning process - more specifically, by injecting triggered samples into the training set in the common setting - so that the victim model behaves normally on benign inputs, but shows attacker-chosen behavior in the presence of the aforementioned specific trigger pattern. The lack of a unified taxonomy in existing studies is addressed by our taxonomy below.

- **3D Backdoor Attacks and their Attributes**

  3D backdoor attacks can be characterized based on four attributes:

  - **Attack Goal**: Whether the attacker wants to change the label of the injected triggered data in the training set (Xiang et al., 2021b; Gao et al., 2025a; Fan et al., 2024a; Guo et al., 2026a) or keep the label clean (Li et al., 2021c; Tian et al., 2021; Wen et al., 2021), or even having the goal of poisoning the utility of the model, so that the model fails even on clean test inputs (Wang et al., 2024f). Also some studies on Backdoors aim to leverage them as a watermarking technique and not as an adversary (Wei et al., 2024).
  - **Trigger Carrier**: Triggers can be embedded in geometric channels of the data (Xiang et al., 2021b; Bian et al., 2024; Gao et al., 2025a; Guo et al., 2026a), other non-geometric channels (Ning et al., 2024), some latent space representation of the data (Feng et al., 2025b; Jiang et al., 2025; Fan et al., 2024a) or operate at sensor-level (Chaturvedi et al., 2025; Zhang et al., 2022b).
  - **Realizability**: We can have purely digital triggers, defined in data space (Xiang et al., 2021b; Bian et al., 2024; Gao et al., 2025a; Guo et al., 2026a) or physical triggers, implemented in real world (Zhang et al., 2022b; Li et al., 2023f).
  - **Trigger Behavior**: We can have a single trigger mapped to a single type of misbehavior (Xiang et al., 2021b; Bian et al., 2024; Gao et al., 2025a; Guo et al., 2026a), or a trigger causing multiple misbehaviors (Shan et al., 2025; Wang et al., 2024f).

- **3D Backdoor Defenses**

  We can detect, leverage or defend against backdoors by acting at different pipeline stages, similar to section 6.1; For example, try and fix the input test data if it is corrupted with some kind of trigger through Reconstruction techniques (Lian et al., 2025; Xiang, 2022), detecting backdoors at inference time by different monitoring approaches (Hu et al., 2023; Xiang, 2022) or moving the pipeline state towards safer states by tampering with different sections of it (Karim et al., 2024a;b; Xiang, 2022). We could even use backdoor-aware training if access exists, or utilize outlier suppression methods (Lan et al., 2025a; Xiang, 2022).

**An Overview on LiDAR Attacks and Defenses**    LiDAR is an active sensor: it emits laser pulses and measures the time-of-flight of the reflections to build a 3D point cloud, often augmented with intensity/reflectivity and timestamp information. A useful way to stay oriented is to group threats by what the attacker can change, namely, what the LiDAR measures, by interfering with the optical signal, what the perception model "believes", by manipulating the point cloud inputs or model behavior, and finally, what the system learns over time, by poisoning/backdoors in training or updates. Guesmi & Shafique (2024) provide a broad view of physically realizable attacks on LiDAR perception and also the survey by (Kim & Kaur, 2024) provides a good starting point as well.

- **LiDAR Attacks**

  Most LiDAR attacks ultimately try to cause either Phantom objects a.k.a "ghosts", by injecting returns, so the system detects something that is not there, or Object hiding/removal, by suppressing or distorting returns, so a real object is missed. Below we have a more detailed taxonomy on attacks on LiDARs:

  - **Optical / sensor-layer spoofing**: *"Making the sensor see wrong points"*, physical attacks that tamper with LiDAR measurements by injecting extra light or disrupting real returns, impacting detection and even upstream tasks (Sun et al., 2020a; Sato et al., 2024; Nagata et al., 2025; Capraru et al., 2026).

- **Physically realizable adversarial objects**: *"Making the world look adversarial"*, instead of shining lasers, an attacker can place or modify real objects to induce a harmful point cloud (Tu et al., 2020a).
- **Training-time poisoning and backdoors**: *"Making the model fail when triggered"*, implanting behaviors in the model that only activate under specific triggers, if having access to the training or fine-tuning process (Li et al., 2023f; Chaturvedi et al., 2026).
- **Availability attacks on 3D point-cloud pipelines**: *"Making perception unreliable or slow"*, beyond targeted misdetections, attacks aiming to degrade the system's overall availability (Zhu et al., 2024c; Wang et al., 2026b).

- **LiDAR Defenses**

  If we mirror the processing pipeline, we can have defenses in different stages similar to previous sections, such as protecting the sensor signal, detecting anomalies at runtime and hardening learning/training. Below the LiDAR defense taxonomy is addressed in a more specific manner:

  - **Sensor- and signal-level hardening**: *"Reducing what an attacker can inject"*, using physics-based checks, like whether occlusions/shadows make sense, because fake points usually can't match these patterns perfectly (Sun et al., 2020a), or adding hardware-level protections, but these type of protections aren't a guaranteed fix (Sato et al., 2024). Finally, applying simple safety rules in software, like rejecting impossible distances/time-of-flight, and requiring consistency across multiple returns or across time.
  - **Runtime detection and monitoring**: *"Catching spoofed frames/objects"*, these defenses assume some attacks will still get through and focus on *detecting* inconsistencies fast enough to trigger safe fallback behavior, such as catching spoofing by checking what should stay consistent in the real world, by flagging fake objects (Sun et al., 2020a), looking for point-level stability (Cho et al., 2023; Wang et al., 2026a), and detecting injected "ghost" objects by monitoring motion trajectories (You et al., 2021a).
  - **Robust learning for 3D perception**: *"Making attacks harder to succeed"*, by improving adversarial robustness of 3D perception via smoothing and robustification techniques (Srinivasan et al., 2021), or using adversarial training and augmentation.
  - **Defending against poisoning/backdoors**: *"Securing the learning pipeline"*, for backdoors, the most reliable mitigations are usually process controls, either by auditing data using dataset/version control, signed updates and separation of training vs. evaluation data, or by trigger hunting/model inspection, running backdoor detection tests for rare "shortcut" features (Li et al., 2023f).

  It's worth mentioning that most runtime defenses ultimately need a system response like reducing speed, requesting additional confirmations from other sensors or switching to a more conservative planning mode, as a defense without a safe fallback plan is only half a defense.

### 6.1.1 Conclusion, Challenges, and Future Directions of 3D ML Pipeline Security Threats

The security of 3D machine learning pipelines extends beyond the robustness of individual neural networks. Vulnerabilities can arise at multiple stages, including sensor-level manipulation, adversarial evasion during inference, and poisoning or backdoor attacks during training. Although important progress has been made in analyzing and mitigating these threats, many defenses remain limited to specific threat models, datasets, or architectures. Therefore, trustworthy 3D perception requires coordinated defenses across the full processing pipeline, especially in safety-critical domains such as driving, robotics, and intelligent infrastructure.

Several challenges remain open. First, there is still a gap between digital evaluations and real-world deployment, particularly for physically realizable attacks affected by sensor properties, environmental conditions, and system-level interactions. Second, the emergence of 3D foundation models, multimodal perception systems, large-scale pretraining, and diverse point-cloud representations introduces new attack surfaces that are not yet well understood. Third, current 3D robustness benchmarks are often fragmented and may not capture adaptive adversaries, multi-stage attacks, or physical constraints, making fair comparison difficult. Finally,

topics such as model extraction, intellectual property protection, ownership verification, and watermarking remain underexplored for 3D models. Future research should therefore develop unified security frameworks, certified robustness guarantees, secure training and deployment pipelines, provenance and ownership verification mechanisms, and cross-modal defenses for resilient 3D perception in adversarial environments.

## 6.2 Robustness to Noise and Occlusion

Real-world 3D point clouds are inevitably affected by measurement noise, quantization artifacts, spurious returns, and outliers, while also exhibiting structured missing regions due to limited viewpoints, clutter, and self-occlusion. Robustness to such degradations is critical because downstream tasks, including classification, detection, segmentation, registration, and completion rely on geometric cues that become unreliable under partial visibility and corruption. Unlike images, point clouds are irregular and often sparse; consequently, perturbations or missing subsets of points may remove entire semantic parts or distort local geometry, leading to performance degradation. Robustness should therefore be treated as an explicit design consideration in 3D perception systems rather than assumed to emerge automatically from architectural scaling or training data size.

Historically, robustness research originated from classical geometric estimation for registration and surface modeling. The Iterative Closest Point (ICP) framework (Besl & McKay, 1992) and its robust variants addressed noise and partial overlap through correspondence pruning and outlier rejection, while RANSAC-style (Fischler & Bolles, 1981) consensus estimation became a standard mechanism for handling heavily contaminated matches. In parallel, Moving Least Squares (MLS) surface fitting (Alexa et al., 2001; Fleishman et al., 2005) provided principled local smoothing under noisy measurements, and handcrafted local descriptors such as Spin Images (Johnson & Hebert, 2002), FPFH (Rusu et al., 2009), and SHOT (Salti et al., 2014) enabled matching and recognition under clutter and occlusion by relying on partial neighborhood statistics. These approaches established robustness primarily through explicit geometric modeling and filtering.

With the advent of learning-based methods, robustness increasingly shifted toward data-driven modeling of corruptions. Early works on robust normal estimation (Boulch & Marlet, 2016) and PCPNet (Guerrero et al., 2018) learned noise-aware local geometry directly from data, while PointCleanNet (Rakotosaona et al., 2020) demonstrated learned denoising and outlier removal for dense point clouds. The modern deep learning era for raw point sets began with PointNet (Qi et al., 2017a) and its hierarchical extension PointNet++ (Qi et al., 2017b). Although not designed explicitly as robustness mechanisms, their permutation-invariant set processing and multi-scale aggregation biases have been shown to mitigate sensitivity to non-uniform sampling and partial visibility. Similarly, convolutional operators such as KPConv (Thomas et al., 2019) stabilize feature extraction under irregular sampling by enforcing structured local support, indirectly contributing to robustness in downstream tasks.

Transformer-based architectures subsequently improved contextual reasoning under incomplete observations. Point Transformer (Zhao et al., 2021a) captures long-range dependencies that can compensate for missing regions, while masked point modeling approaches such as Point-BERT (Yu et al., 2022) train encoders to infer masked structure from surrounding context and are often used for pretraining in partial-input tasks. In parallel, completion-oriented pipelines interpret occlusion as reconstruction, exemplified by PCN (Yuan et al., 2018). At the evaluation level, benchmark-driven analyses exposed substantial robustness gaps: ModelNet40-C (Sun et al., 2022a), Robo3D (Kong et al., 2023), and ModelNet-O (Fang et al., 2024) demonstrated that performance measured on clean datasets frequently overestimates real-world reliability under common corruptions and occlusion patterns.

Building on these insights, recent work has explored explicit training-time and inference-time robustness mechanisms. **Ensemble-based** aggregation and augmentation strategies such as EPiC (Levi & Gilboa, 2023) and Set-Mixer (Zhang et al., 2024b) improve classification stability, while sampling or filtering protocols including PointCVaR (Li et al., 2024g) and PointSP (Li et al., 2025b) target resilience through risk-aware point selection. Parameter-efficient prompting strategies such as UPP (Ai et al., 2025) extend adaptation across multiple tasks without modifying backbone weights. Generative priors further extend robustness to severe occlusion: **diffusion-based** registration via PointDifformer (She et al., 2024) and completion using ComPC (Huang et al.) demonstrate improved recovery from extreme partiality. Complementarily, certifiable

Table 12: Pipeline-oriented taxonomy of robustness mechanisms against noise and occlusion in 3D point cloud processing. Representative methods are organized by task and stage at which robustness is introduced, including training, architectural design, inference-time adaptation, geometric estimation, and generative reconstruction.

| Task | Training Strategy | Model / Arch. | Preprocess. / Adapt. | Geom. Solvers | Generative |
|---|---|---|---|---|---|
| Classification | EPiC (ens) (Levi & Gilboa, 2023) PointCutMix-R (aug) (Zhang et al., 2022a) Point-BERT (pretrain) (Yu et al., 2022) | Set-Mixer (arch) (Zhang et al., 2024b) | PointSP (samp) (Li et al., 2025b) PointCVaR (filt) (Li et al., 2024g) UPP (adapt) (Ai et al., 2025) | – | – |
| Detection | Corruption aug. (aug) (Dong et al., 2023) | – | SOR (filt) (Rusu et al., 2008) | – | – |
| Segmentation | – | KPConv (arch) (Thomas et al., 2019) RandLA-Net (arch) (Hu et al., 2020) Point Transformer (arch) (Zhao et al., 2021a) | – | – | – |
| Normal/Denoise | – | Boulch (model) (Boulch & Marlet, 2016) PCPNet (model) (Guerrero et al., 2018) | PointCleanNet (filt) (Rakotosaona et al., 2020) MLS (filt) (Alexa et al., 2001) | – | – |
| Registration | – | – | FPFH (feat) (Rusu et al., 2009) | ICP (solver) (Besl & McKay, 1992) RANSAC (solver) (Fischler & Bolles, 1981) TEASER (solver) (Yang et al., 2020a) | PointDifformer (model) (She et al., 2024) |
| Completion | – | PCN (model) (Yuan et al., 2018) | MLS (samp) (Alexa et al., 2001) | – | ComPC (gen) (Huang et al.) |

*Abbreviations:* aug = augmentation, ens = ensemble, pretrain = self-supervised pre-training, arch = architecture, model = learned model, samp = sampling, filt = filtering/outlier removal, adapt = test-time adaptation, feat = feature descriptor, solver = geometric solver, gen = generative model.

solvers such as TEASER (Yang et al., 2020a) provide theoretical guarantees for registration under extreme outlier ratios, highlighting the continued relevance of classical robust estimation within modern pipelines.

Table 12 provides a task-oriented view of robustness mechanisms against noise and occlusion, organizing representative methods by both application domain and intervention strategy. Each row corresponds to a core point cloud task such as classification, detection, segmentation, normal estimation, denoising, registration, and completion. While columns categorize techniques according to where robustness is introduced within the processing pipeline: (1) training-time strategies such as data augmentation, ensemble learning, and self-supervised pretraining; (2) model and architecture designs with built-in robustness properties; (3) inference-time preprocessing and adaptation, including sampling protocols, outlier filtering, and test-time adaptation; (4) classical geometric solvers; and (5) generative priors that leverage learned shape distributions for reconstruction.

The distribution of entries reveals several patterns. Classification benefits from the broadest range of mechanisms, spanning ensemble training (Levi & Gilboa, 2023), augmentation (Zhang et al., 2022a), self-supervised pretraining (Yu et al., 2022), robust architecture design (Zhang et al., 2024b), and diverse inference time strategies including adaptive sampling (Li et al., 2025b), gradient-based outlier removal (Li et al., 2024g), and point-level prompting (Ai et al., 2025). Registration remains anchored in classical geometric solvers (ICP, RANSAC, and TEASER), now complemented by learned models such as PointDifformer (She et al., 2024) that exploit neural diffusion equations for robust feature extraction. Completion tasks combine learned reconstruction networks (Yuan et al., 2018) with generative priors that harness 2D diffusion models (Huang et al.). In contrast, detection and segmentation exhibit sparser coverage across all columns, indicating domains where future robustness research may be especially valuable.

### 6.2.1 Conclusion, Challenges, and Future Directions of Robustness to Noise and Occlusion

Despite rapid progress, dependable robustness to noise and occlusion remains unresolved. Benchmarks such as ModelNet40-C (Sun et al., 2022a) and Robo3D (Kong et al., 2023) reveal that strong clean-set accuracy may substantially overestimate reliability under realistic corruptions, yet constructing corruption models that are physically plausible and semantics-preserving across sensors and tasks remains challenging. Occlusion realism constitutes a second bottleneck: although ModelNet-O (Fang et al., 2024) formalizes controlled self-occlusion, discrepancies between synthetic partiality and deployment-time visibility patterns limit transferability when viewpoints or acquisition conditions shift.

Algorithmically, robustness mechanisms are not yet unified into efficient end-to-end pipelines. Multi-stage enhancement can introduce domain mismatch or additional failure modes, and parameter-efficient adaptation may not explicitly suppress severe input defects, leading to degraded representations on low quality scans (Ai et al., 2025). Multimodal and generative pretraining partially address data discrepancy (Nguyen et al., 2025a; Huang et al.), yet raise open questions regarding scalability, computational cost, and calibrated uncertainty when reconstructed geometry is plausible but incorrect. Finally, compound degradations remain underexplored: detection accuracy can drop sharply under combined corruptions (Li et al., 2023g), and completion models trained on synthetic data struggle when noise and occlusion co-occur (Tesema et al., 2025), motivating evaluation protocols that explicitly target multi-factor sensor failures.

### 6.3 Rotation Invariance and Equivariance

**Rotation invariance** in 3D point cloud analysis refers to the property that a model produces identical outputs for an input point cloud and any of its rotated versions. This property is essential due to the arbitrary poses induced by sensor viewpoints, object placement, or scene acquisition processes (Fei & Deng, 2024). Early studies on rotation-robust point cloud analysis primarily focused on achieving rotation invariance through geometric descriptors, such as 3D Zernike descriptors, which encode the overall shape in a rotation-invariant manner (Novotni & Klein, 2004). Recent networks for learning rotation-invariant geometric features in 3D point cloud analysis can be grouped into several architectural paradigms.

- **Invariant point-feature-based** methods extract invariant features by transforming raw coordinates into rotation-invariant representations. Notable examples include PPFNet (Deng et al., 2018b) and its unsupervised variant PPF-FoldNet (Deng et al., 2018a), which learn robust local descriptors for tasks such as matching and registration. Other representative works, such as Parot (Zhang et al., 2023a), PRIN (You et al., 2021b), SGMNet (Xu et al., 2021a), and (Luo & Gao, 2024), further improve rotation invariance by disentangling shape and pose or by leveraging learned local reference frames to normalize point neighborhoods.

- **Rotation-invariant convolution** methods design convolution operators that aggregate local geometric information in a manner insensitive to global orientation. Representative works include RIConv (Zhang et al., 2019a), RIConv++ (Zhang et al., 2022d), RI-GCN (Kim et al., 2020),RISurConv (Zhang et al., 2024n) and (Lin et al., 2026) .

- **Rotation-invariant transformer** methods leverage self-attention mechanisms and feature decoupling strategies to maintain consistent representations under arbitrary rotations. Representative models include RITNet (Yang et al., 2022b), RoITr (Yu et al., 2023a), MaskLRF (Furuya, 2024), self-supervised invariant transformers (Furuya et al., 2024), and RotInv-PCT (He et al., 2025a).

**Rotation equivariance** requires that a model's output transforms predictably under rotations of the input such that features rotate consistently with the point cloud. This property provides a strong inductive bias for 3D geometry (Fei & Deng, 2024). Rotation equivariance methods (SO(3)/SE(3)) for point clouds and related 3D data can be grouped as:

- **Equivariant point convolutions / kernel operators** build SO(3)- or SE(3)-equivariant layers on point sets so that features rotate consistently with the input. Representative examples include

(Finzi et al., 2020), EPN (Chen et al., 2021a), E2PN (Zhu et al., 2023a), and CSEConv (Kim et al., 2024a).

- **Equivariant message passing (geometric GNNs)** enforces equivariance through geometric update rules for nodes and, in some cases, coordinates. Representative examples include EGNN (Satorras et al., 2021), SEGNN (Brandstetter et al.), and ClofNet (Du et al., 2022).

- **Equivariant attention / transformers** integrate tensor or irreducible representation features into attention mechanisms so that aggregation remains SE(3)-consistent. Representative examples include SE(3)-Transformer (Fuchs et al., 2020), Equiformer (Liao & Smidt, 2022), EquiformerV2 (Liao et al., 2024), and BITR (Wang & Jörnsten, 2024).

- **Equivariant priors for pose, registration, and fitting** use SE(3)-equivariant representations to improve robustness in downstream alignment and pose recovery (Li et al., 2021b; Feng et al., 2023; Kang et al., 2024; Yao et al., 2024; Li et al., 2025a).

- **Equivariant generative models on SE(3)** define diffusion or flow models on SE(3) so that denoising respects rotational symmetry, supporting robust pose estimation and alignment (Jiang et al., 2023a; Min et al., 2025; Tie et al., 2025).

### 6.3.1 Conclusion, Challenges, and Future Directions of Rotation Invariance and Equivariance

Rotation invariance and rotation equivariance are important properties for robust 3D point cloud learning. Rotation-invariant methods aim to produce the same output when the input point cloud is rotated, while rotation-equivariant methods preserve feature transformations consistently with the rotation of the input. Despite their importance, handling arbitrary rotations in real-world point clouds remains challenging, especially when models must also remain accurate, efficient, and scalable. Some rotation-aware methods introduce additional architectural complexity or computational cost compared with standard point cloud networks. Future research should therefore focus on developing methods that generalize better across rotations, sensor viewpoints, and 3D tasks while maintaining high accuracy, low computational cost, and practical usability in real-world applications.

## 6.4 Domain Adaptation

Deep learning models for 3D data often degrade under domain shifts caused by sensor variations, environmental changes, and synthetic-to-real gaps. Domain Adaptation (DA) adapts models to target domains, while Domain Generalization (DG) builds models robust to unseen domains. This section reviews recent advances organized by paradigms and tasks.

3D domain adaptation extends concepts from 2D computer vision but introduces distinct challenges. Wilson & Cook (2020) reviewed unsupervised domain adaptation in general, while Oza et al. (2023) concentrated on object detection. Theoretical insights were discussed by Redko et al. (2020). Surveys tailored to 3D tasks included Sohail et al. (2025), which focused on point cloud transfer learning, and Triess et al. (2021), which covered LiDAR perception. Additionally, Xiao et al. (2025b) provided an overview of prompt-based adaptation.

Four paradigms characterize 3D DA. **Feature alignment** minimizes distribution discrepancies via adversarial or metric learning. **Self-training with pseudo-labeling** generates target predictions as pseudo-labels for iterative refinement. **Domain mixing** creates hybrid samples between domains. **Generative and reconstruction-based** methods use generative models or reconstruction. Table 13 summarizes these approaches.

Here, This survey describes domain adaptation in 3D across major tasks, including classification, segmentation, and detection.

Table 13: Paradigms in 3D Domain Adaptation with Representative Methods and Key Characteristics

| Paradigm | Core Mechanism | Representative Methods and Key Contributions | Primary Tasks |
|---|---|---|---|
| Feature Alignment | Minimize distribution discrepancy between source and target feature spaces using adversarial training, metric learning, or moment matching. | **PointDAN** (Qin et al., 2019a): Aligns multi-scale local and global features via adversarial adaptation. **PC-Adapter** (Park et al., 2023b): Lightweight adapter modules for parameter-efficient tuning. **SADA-3D** (Huang et al., 2025d): Incorporates structural consistency losses to preserve geometric relationships. | Segmentation, Classification |
| Self-Training & Pseudo-Labeling | Generate pseudo-labels on target data using source model predictions, then refine models through iterative training with filtered or corrected labels. | **ConUDA** (Li et al., 2025f): Confidence-guided sampling to select reliable pseudo-labels. **RefRec** (Cardace et al., 2021): Refines pseudo-labels through shape reconstruction consistency. **GAST** (Zou et al., 2021): Geometry-aware self-training with rotation prediction as pretext task. | Segmentation, Detection |
| Domain Mixing & Intermediate Domains | Create intermediate or hybrid feature/input spaces between domains to smooth the transition. | **ConDA** (Kong et al., 2021): Regularized concatenation of source and target features. **UMDMix** (Nihal et al., 2025): Swaps urban structural patches of varying scales. **BeyondMix** (Chen et al., 2025e): Leverages structural priors and long-range dependencies. | Segmentation |
| Generative & Reconstruction-Based | Use generative models (GANs, VAEs, Diffusion) or reconstruction to learn domain-invariant representations or directly modify data. | **DiffRefine** (Shin et al., 2025): Diffusion models to densify sparse object proposals. **RefRec** (Cardace et al., 2021): Shape reconstruction as auxiliary task. **DPGLA** (Li et al., 2025g): Aligns geometric patterns between synthetic and real data. | Detection, Segmentation |

### 6.4.1 Point Cloud Classification

Point cloud classification, typically applied to individual objects rather than scenes, serves as a foundation for adaptation techniques. Feature alignment methods have demonstrated the applicability of techniques from 2D vision to 3D domains. Qin et al. (2019a) proposed PointDAN, which aligned both global shape features and local geometric features using multi-scale adversarial adaptation with self-adaptive attention. Cardace et al. (2021) introduced RefRec, which refined pseudo-labels through consistency with shape reconstruction from multiple viewpoints, leveraging geometric consistency as a supervisory signal.

Self-supervised learning techniques learned domain-invariant representations without labels through pretext tasks. Fan et al. (2022a) developed methods that learned by predicting geometric transformations applied to point clouds, forcing models to understand underlying structure rather than superficial patterns. Zou et al. (2021) presented Geometry-Aware Self-Training (GAST), which integrated geometric pretext tasks like rotation prediction with pseudo-label refinement to encourage orientation and scale invariance.

Efficient adaptation approaches addressed the growing size of pretrained models. Park et al. (2023b) proposed PC-Adapter, inserting lightweight adapter modules into frozen pretrained backbones for parameter-efficient tuning while leveraging rectified pseudo-labels for adaptation. Khoche et al. (2025) introduced BlendCLIP, leveraging vision-language models for zero-shot classification across domains using textual descriptions to align visual features.

Emerging directions included holistic scene understanding (Knights, 2025), cross-modal transfer between different sensors (BRIDGING), digital twin generation Shahbaz & Agarwal (2025), and specialized applications such as primitive segmentation (Wang et al., 2025k) and infrastructure monitoring (Xu et al., 2026b).

### 6.4.2 Semantic Segmentation

3D semantic segmentation faces significant domain shift challenges due to its dense prediction nature and sensitivity to variations in point density, occlusion patterns, and scene layouts. The synthetic-to-real gap is particularly pronounced, as simulators typically produce clean, dense point clouds that differ substantially from sparse, noisy real-world sensor data.

Cross-modal approaches leveraged complementary information from other sensors, most commonly 2D cameras. Jaritz et al. (2022) used 2D segmentation predictions to guide 3D adaptation by projecting 2D labels onto 3D point clouds. Wu et al. (2023b) proposed a bidirectional fusion-then-distillation framework that fused 2D and 3D features at multiple scales before distilling knowledge between domains to preserve rich contextual information.

Domain mixing strategies created intermediate training samples to smooth transitions between domains. Kong et al. (2021) introduced ConDA, which constructed an intermediate domain through regularized concatenation of source and target features, encouraging the learning of domain-agnostic representations. For urban scenes, Nihal et al. (2025) proposed UMDMix, which swapped structural patches between domains to enhance robustness to diverse urban layouts. Chen et al. (2025e) extended this approach with BeyondMix, leveraging structural priors and modeling long-range dependencies through transformer architectures.

Self-training has become a dominant paradigm for unsupervised domain adaptation in segmentation. ConUDA, proposed by Li et al. (2025f), addressed noise accumulation through confidence-guided pseudo-label sampling, while SADA-3D, developed by Huang et al. (2025d), incorporated structure-aware consistency losses to ensure that pseudo-labels respect object geometries. Yi et al. (2021) took a different approach with Complete & Label, using point cloud completion as a canonical pretraining task to learn geometry-invariant features robust to sparsity variations.

Specialized methods addressed the synthetic-to-real gap directly. Zhao et al. (2021c) proposed ePointDA, which simulated realistic sensor dropout and noise patterns during training on synthetic data. Li et al. (2023a) employed adversarial noise injection to make synthetic data mimic real-world occlusion patterns, while Li et al. (2025g) focused on aligning geometric patterns between synthetic and real LiDAR scans in DPGLA.

### 6.4.3 Object Detection

3D object detection which involves localizing and classifying objects such as vehicles and pedestrians within point clouds, faces distinct domain shift challenges. Detection performance depends heavily on point density within object regions, occlusion patterns, and object scale distributions, all of which vary significantly across domains.

Source-free domain adaptation (SFDA) addressed practical constraints in which source data are unavailable during adaptation. Saltori et al. (2020) proposed SF-UDA3D, which leveraged temporal coherence in LiDAR sequences to refine detection proposals and reduce false positives. Wang et al. (2025h) introduced Vicinal Gaussian Transform (VGT), which modeled target feature distributions and enforced label consistency through covariance contraction informed by source domain statistics.

Generative approaches addressed sparse point clouds within detection proposals. Shin et al. (2025) proposed DiffRefine, employing conditional diffusion models to densify sparse object proposals and reconstruct canonical shapes for improved classification. Yang et al. (2025a) developed CounterPC, a counterfactual reasoning framework that disentangled domain-specific geometric variations from semantic content by asking what objects would look like in alternative domains.

Sensor alignment methods addressed hardware-induced domain shifts directly. Tanaka et al. (2025) developed adaptation methods for varying LiDAR configurations, while Tsai et al. (2022) proposed See Eye to Eye, a LiDAR-agnostic framework that normalized scan pattern differences across multiple sensor types. Luo et al. (2021b) enforced multi-level consistency across network stages to improve robustness to domain variations. Huch et al. (2023) provided empirical analysis quantifying the sim-to-real gap, identifying key factors including intensity distribution differences and ray drop patterns.

### 6.4.4 Conclusion, Challenges, and Future Directions of Domain Adaptation

Domain adaptation is an important direction for improving the generalization of point cloud models across different sensors, environments, datasets, and deployment conditions. Since real-world 3D data often differs from the training distribution, models must remain reliable under changes. Key challenges include extreme sparsity, multi-modal shifts, and real-time adaptation requirements. Future directions encompass unified

frameworks, foundation model adaptation, test-time adaptation, and theoretical advances. Robust domain adaptation remains essential for real-world 3D vision deployment across diverse conditions.

## 6.5 Anomaly Detection

Anomaly detection plays a key role in ensuring stable performance, preventing errors, and improving the efficiency of industrial systems. Earlier approaches relied on manual inspection, which is labor-intensive and prone to human error. With the advent of deep learning and computer vision, the field has shifted towards automated visual inspection systems, substantially increasing inspection speed and reliability. RGB image-based anomaly detection systems achieve high accuracy and generalization across industrial and real-world scenarios by using knowledge distillation to train student-teacher networks (Rudolph et al., 2023; Gu et al., 2023), adapting pretrained vision models for efficient feature transfer (Reiss et al., 2021; Zhou et al., 2023d; Pan et al., 2025c), and synthesizing artificial anomalies for robust optimization (Cao et al., 2023; Zavrtanik et al., 2024a). 3D anomaly detection, using data such as point clouds and depth maps, overcomes the limitations of 2D methods and enables the identification of structural, geometric, and internal defects that are difficult or impossible to detect with 2D techniques.

Unsupervised methods are proposed as a practical approach that is trained only on normal data. These methods fall into three main technical categories: First, reconstruction-based such as autoencoders (Masuda et al., 2021), diffusion models (Zhou et al., 2024d), and dual reconstruction architectures (Li et al., 2024k;d; Hoang et al., 2025; Liang et al., 2025b), detect anomalies through high reconstruction error. Second, feature-based methods, including memory banks (Sabokrou et al., 2017; Chu et al., 2023; Wang et al., 2023g; Bhunia et al., 2024), self-supervised learning (Tu et al., 2024), and teacher-student architectures (Sun et al., 2024c), learn compact and normalized feature representations. Finally, group-level contrastive learning identifies inconsistencies as anomalies by comparing samples in latent space (Zhu et al., 2024b).

In production environments, supervised anomaly detection methods train models to detect abnormal samples using limited, usually customized, labeled data (Wang et al., 2025a; Kaji et al., 2022; Xu et al., 2024b; Bolourian et al., 2023; Hwang & Kang, 2023).

Few-Shot methods in 3D point clouds anomaly detection address the challenge of lacking anomalous data through several key approaches. Data synthesis methods generate new anomalous samples (Li et al., 2024c; 2025m; Zavrtanik et al., 2024b; Xiang et al., 2025a). Feature adaptation models leverage knowledge from other domains (Liu et al., 2024c; Zhou et al., 2024b; 2025b). Prompt learning methods aim to teach models the concept of anomalous data using textual or visual prompts (Zuo et al., 2024b).

As shown in Table 14, the development of 3D anomaly detection is strongly driven by practical demands across diverse application domains. This applied focus has led to the creation of tailored benchmark datasets, which are generally categorized into pure point cloud and RGB-D formats (Bergmann et al., 2021; Bonfiglioli et al., 2022; Zhou et al., 2023c; Liu et al., 2023c; Wang et al., 2024a; McHard et al., 2025; Li et al., 2025i; Cheng et al., 2026).

Table 14: 3D anomaly detection applications.

| Application Domain | Methods |
|---|---|
| Industrial Manufacturing | 3DRÆM (Zavrtanik et al., 2024b), F2PAD (Tao et al., 2024), 3D-CSAD (Cao et al., 2024b), MVR (Sun et al., 2025b), CPS3D-AD (Guo et al., 2025a), PLANE (Wang et al., 2025g), 3DMulti-FPFHI (Jing et al., 2025), MC4AD (Liang et al., 2025a) |
| Autonomous Systems and Robotics | PAAD (Ji et al., 2022), PointAD (Zhou et al., 2024b), MADFlow (Li et al., 2025k), MiniShift (Cheng et al., 2026), Splatpose (Kruse et al., 2024), Splatpose+ (Liu et al., 2024f), PIAD (Yang et al., 2025c) |
| Healthcare | MADGAN (Han et al., 2021), SimpleSliceNet (Zhang & Mohsenzadeh, 2025) |

### 6.5.1 Conclusion, Challenges, and Future Directions of Anomaly Detection

Despite significant progress, several challenges persist. Generalizing to unseen defect types remains difficult, supervised paradigms demand costly fine-grained labels, and balancing real-time inference speed with accuracy on edge devices is non-trivial. Multimodal RGB-3D fusion is still underutilized, and existing benchmarks lack the diversity of real-world production environments. Promising future directions include unified foundation models for zero-shot anomaly detection, scalable self-supervised pre-training strategies that reduce annotation dependence, enhanced explainability for safety-critical applications, and dynamic evaluation protocols that track concept drift. Addressing these issues will be essential for deploying robust, scalable anomaly detection in smart manufacturing.

## 7 Evaluation and Benchmarking

To provide a comprehensive comparison of existing approaches in 3D point cloud learning, this section presents an evaluation framework that highlights the benchmark datasets, evaluation metrics, and performance analyses commonly reported in the literature. In this survey, widely used datasets are first presented, highlighting their associated tasks, data representations, scale, and data sources. Next, the evaluation metrics commonly adopted to measure performance across different 3D tasks are summarized. Finally, a performance analysis of representative deep learning approaches for point cloud understanding is provided, focusing on major tasks such as classification, segmentation, and object detection. This structured evaluation helps clarify benchmarking practices in the literature and facilitates a more consistent comparison of existing methods.

### 7.1 Datasets

3D point cloud datasets are very diverse. They cover many tasks, such as classification, segmentation, detection, registration, completion, reconstruction, and generation. They also come from different sources, such as synthetic CAD models or real scans captured by sensors. So they can differ substantially in size and in how detailed their labels are.

Another key difference is the 3D representation used in each dataset, such as meshes, CAD surfaces, point clouds, voxels, or multi-view images. This choice affects how methods are built, how data are labeled, and how results are evaluated. To describe datasets in a clear and consistent way, we summarize each one using the same attributes: task, representation, bumber of classes, number of samples, type (source). This helps readers quickly compare datasets along several axes: data quality, label richness, synthetic versus real data, and dataset scale. Table 15 lists these fields, which helps compare datasets and choose the right benchmarks for a specific 3D setting.

Table 15: Summary of the 3D Point cloud datasets for each task

| Dataset | Task | Representation | #Classes | #Samples | Type |
|---|---|---|---|---|---|
| ShapeNet (2015) (Chang et al., 2015) | Classification, Reconstruction | Mesh | 55 | 51,190 | Synthetic |
| ModelNet10 (2015) (Wu et al., 2015) | Classification, Generation, Reconstruction | Mesh | 10 | 4,899 | Synthetic |
| ModelNet40 (2015) (Wu et al., 2015) | Classification, Generation, Reconstruction | Mesh | 40 | 12,311 | Synthetic |
| ScanNet (2017) (Dai et al., 2017) | Classification, Semantic Segmentation, Reconstruction | RGB-D + Meshes | 20 | 1,513 scans, 12,283 samples | Indoor |
| ScanNet200 (2022) (Rozenberszki et al., 2022) | Semantic Segmentation, Instance Segmentation | RGB-D + Meshes | 200 | 1,513 scans, 12,283 samples | Indoor |
| Semantic3D (2017) (Hackel et al., 2017) | Semantic Segmentation | LiDAR Point Cloud | 8 | 4B | Outdoor |

**Table 15 Continued**

| Dataset | Task | Representation | #Classes | #Samples | Type |
|---|---|---|---|---|---|
| S3DIS (Area 5) (2017) (Armeni et al., 2016) | Semantic Segmentation | RGB + Point Cloud | 13 | 695.9M | Indoor |
| SemanticKITTI (2019) (Behley et al., 2019) | Semantic Segmentation | LiDAR Point Cloud | 19 | 4.549B points | Outdoor |
| nuScenes (2019) (Caesar et al., 2020) | Detection, Semantic Segmentation; Tracking | LiDAR + Cameras | 16 | 1,000 scenes, ~40k keyframes | Outdoor |
| SemanticPOSS (2020) (Pan et al., 2020) | Semantic Segmentation | LiDAR Point Cloud | 14 | 216M points | Outdoor |
| BlendedMVS (2020) (Yao et al., 2020) | Foundation Models | RGB-D | – | 17,000 images | Synthetic |
| CO3D (2021) (Reizenstein et al., 2021) | Foundation Models | RGB + Point Cloud | 50 | 1.5 million frames | Real-world |
| KITTI (2012) (Geiger et al., 2012) | Detection, Tracking | LiDAR + GPS/IMU | 9 | 14,999 | - |
| Waymo (2019) (Sun et al., 2020b) | Detection, Tracking; Foundation Models | LiDAR + cameras + radar | 4 detection; 23–28 segmentation | 390,000 frames | Real-world |
| H3D (2019) (Patil et al., 2019) | Tracking | Point Cloud | 8 | 27,721 | - |
| Owlii (2017) (Xu et al., 2017) | Compression | Mesh | - | 4 sequences × 600 frames | Real-world |
| 8iVSLF (2018) (d'Eon et al., 2017) | Compression | Voxel | - | 1 sequence × 300 frames + 6 single-frame point clouds | Real-world |
| 8iVFB v2 (2017) (d'Eon et al., 2017) | Compression | Voxel | - | 4 sequences × 300 frames | Real-world |
| MVUB (2016) (Loop et al., 2016) | Compression | Voxel | - | 5 subjects; 7–10 s at 30 fps | Real-world |
| PKU-DPCC (2024) (Xie et al., 2024c) | Compression | Mesh | - | 50 sequences × 250 frames | Real-world |
| Matterport3D (2017) (Chang et al., 2017) | Generation, Reconstruction | RGB-D + Mesh | - | 194,400 RGB-D images | Real-world |
| Completion3D (2019) (Tchapmi et al., 2019) | Generation | Point Cloud | 8 | 30,974 3D models | Synthetic |
| VIPC (2021) (Zhang et al., 2021b) | Generation | RGB + Point Cloud | 13 | ~919,872 | Synthetic |
| Google Scanned Objects (2022) (Downs et al., 2022) | Generation | Mesh + SDF | 17 | 1030 | Real-world |
| Sun3D (2013) (Xiao et al., 2013) | Reconstruction | RGB + Point Cloud | - | 415 sequences | Real-world |
| ABC dataset (2019) (Koch et al., 2019) | Reconstruction | Mesh | - | 1 million 3D models | Synthetic |
| 3D-FUTURE (2020) (Fu et al., 2021) | Reconstruction | Mesh | - | 20,240 images | Synthetic |
| LineMod (2012) (Hinterstoisser et al., 2012) | 6DoF Pose Estimation | RGB-D | 15 | ~1,200 per class | Indoor |
| LineMod-Occluded (2014) (Brachmann et al., 2014) | 6DoF Pose Estimation | RGB-D | 15 | 1214 | Indoor |
| T-less (2017) (Hodan et al., 2017) | 6DoF Pose Estimation | RGB-D | 30 | ~1,600 per class | Industrial |
| YCB-Video (2018) (Xiang et al.) | 6DoF Pose Estimation | RGB-D | 21 | 92 videos | Indoor |
| ScanObjectNN (2019) (Uy et al., 2019) | Robustness | Point Cloud | 15 | 15,000 | Real-world |
| RobustPointSet (2020) (Taghanaki et al., 2020) | Robustness | Point Cloud | 40 | 17,276 | - |
| ModelNet40-C (2022) (Sun et al., 2022b) | Robustness | Point Cloud | 40 | 185,100 | Synthetic |
| Point Cloud-C (2022) (Ren et al., 2022) | Robustness | Point Cloud | 40 classification + 16 part labels | 186,970 | Synthetic |

**Table 15 Continued**

| Dataset | Task | Representation | #Classes | #Samples | Type |
|---|---|---|---|---|---|
| ShapeNet-C (2022) (Ren et al., 2022) | Robustness | Point Cloud | 16 | 16,881 | Synthetic |
| ModelNet40-E (2025) (Alonso et al., 2025) | Robustness | Point Cloud | 40 | 12,311 | Synthetic |
| MVTec 3D-AD (2019) (Bergmann et al., 2019) | Anomaly Detection | RGB + Point Cloud | 15 | 5,354 | Industrial |
| Real3D-AD (2023) (Liu et al., 2023c) | Anomaly Detection | Point Cloud | 12 | 1,254 | - |
| MAD (2023) (Zhou et al., 2023c) | Anomaly Detection | Multi-view RGB | 20 | 4,000 | - |
| Real-IAD (2024) (Wang et al., 2024a) | Anomaly Detection | RGB-D | 30 | 150,000 | Industrial |
| Anomaly-ShapeNet (2024) (Li et al., 2024d) | Anomaly Detection | Point Cloud | 40 | 1,600 | Industrial |
| ScanRefer (2020) (Chen et al., 2020a) | Captioning; Grounding | RGB-D scans | 806 | 11,046 | Indoor |
| ScanQA (2022) (Azuma et al., 2022) | Captioning; Question Answering | Mesh | - | 40,000 QA pairs | Indoor |
| ScanNet++ (2023) (Yeshwanth et al., 2023) | Captioning | DSLR; RGB-D | 460 | 280k DSLR images | Indoor |
| Multi3DRefer (2023) (Zhang et al., 2023g) | Captioning; Grounding | Point Cloud scenes | 265 | 11,609 | Indoor |
| ExCap3D (2025) (Yeshwanth et al., 2025) | Captioning | DSLR images | 2000 | 34,700 objects | Indoor |
| ReferIt3D (2020) (Achlioptas et al., 2020) | Grounding | Point Cloud | 1,273 scenes | 11,375 object contexts | Indoor |
| ALFRED (2020) (Shridhar et al., 2020) | Grounding | 3D scans (AI2-THOR) | 120 scenes; 84 classes | 428,322 | Indoor |
| 3DVQA (2022) (Etesam et al., 2022) | Question Answering | Point Cloud | - | 484,359 questions | Indoor |
| 3DQA (2022) (Ye et al., 2022) | Question Answering | Point Cloud | - | 6,000 questions | Indoor |
| FE-3DGQA (2022) (Zhao et al., 2022) | Question Answering | Point Cloud | - | 20,125 QA | Indoor |
| CLEVR3D (2023) (Yan et al., 2023b) | Question Answering | Point Cloud | - | 171,000 QA | Synthetic and Indoor |
| Space3D-Bench (2024) (Szymańska et al., 2024) | Question Answering | Mesh + Point Cloud + RGB-D | - | 1,000 QA | Indoor |
| 3DSRBench (2025) (Ma et al., 2025b) | 3D Reasoning | 2D Image | - | 2,772 QA | - |
| SURPRISE3D (2025) (Huang et al., 2026b) | 3D Reasoning | Mesh + Point Cloud | - | 200,000 QA | Indoor |

## 7.2 Metrics

Choosing appropriate evaluation metrics is essential in 3D learning, since performance must be measured differently across tasks. For instance, classification is judged by whether labels are predicted correctly, whereas segmentation, detection, registration, reconstruction, and generation are evaluated using measures such as region overlap, geometric distance, or alignment error. To present these metrics in a clear and consistent way, we summarize each metric by its metric name, the target task, the mathematical Formula, and a short description. This organization helps readers quickly identify which metrics are commonly used for each 3D task and how they are computed (see Table 16).

Table 16: Summary of the 3D point cloud metrics

| Metric | Task | Formula | Description |
|---|---|---|---|
| Accuracy | Classification, Tracking, Robustness | $\frac{TP+TN}{TP+TN+FP+FN}$ | Fraction of all predictions that are correct. |
| Precision | Classification, Reconstruction, Robustness | $\frac{TP}{TP+FP}$ | Of the predicted positives, how many are truly positive. |
| Recall | Classification, Reconstruction | $\frac{TP}{TP+FN}$ | Of the true positives, how many are successfully found (penalizes misses). |
| F1-Score | Classification, Generation, Reconstruction, Foundation Models | $2 \cdot \frac{\text{Precision} \cdot \text{Recall}}{\text{Precision} + \text{Recall}}$ | Harmonic mean of precision and recall, balancing false positives and false negatives. |
| IOU | Segmentation, Grounding, Foundation Models | $IoU(A,B) = \frac{|A \cap B|}{|A \cup B|}$ | Mean IoU across classes, averaging per-class overlap to mitigate class imbalance. |
| mAP | Detection, Segmentation, Robustness | $\frac{1}{C} \sum_{c=1}^{C} \sum_n \left( R_{c,n} - R_{c,n-1} \right) P_{c,n}$ | AP summarizes the precision–recall curve; mAP is the mean AP over all classes. |
| MIOU | Segmentation, Robustness | $\frac{1}{C} \sum_{c=1}^{C} \frac{TP_c}{TP_c + FP_c + FN_c}$ | Mean intersection-over-union across classes. |
| Success | Tracking | $\frac{\text{Area}(B_p \cap B_{gt})}{\text{Area}(B_p \cup B_{gt})}$ | Percentage of frames where IoU exceeds a threshold. |
| Precision | Tracking | $\sqrt{(x_p - x_{gt})^2 + (y_p - y_{gt})^2}$ | Precision evaluates the tracker's ability to locate the center of the target accurately. |
| HOTA (Higher Order Tracking Accuracy) | Tracking | $\sqrt{\frac{TP}{TP+FP+FN} \cdot \frac{1}{TP} \sum_{i=1}^{TP} \frac{|A_i \cap \hat{A}_i|}{|A_i \cup \hat{A}_i|}}$ | HOTA balances Detection Accuracy (DetA) and Association Accuracy into a single score. |
| Point to point PSNR | Compression | - | D1 (point-to-point) PSNR measures geometry accuracy by comparing nearest neighbor distances. |
| Point to plane PSNR | Compression | - | D2 PSNR measures geometry distortion by point-to-plane error with normals. |
| Bits per point | Compression | $\frac{R}{N}$ | Bits per point (bpp) is a metric that averages number of coded bits per point. |
| BD-PSNR | Compression | $\frac{1}{x_{\max} - x_{\min}} \left[ \frac{\Delta a}{4} x^4 + \frac{\Delta b}{3} x^3 + \frac{\Delta c}{2} x^2 + \Delta d\, x \right]_{x_{\min}}^{x_{\max}}$ | BD-PSNR (Bjøntegaard Delta PSNR) is a rate-distortion summary metric averaged over a range of bitrates. |
| Bits per occupied voxel | Compression | $\frac{R}{N_{\text{occ}}}, \quad R \in \mathbb{N} \text{ (total bitstream size in bits)}, \ N_{\text{occ}} = |\mathcal{V}_{\text{occ}}|$ | Bits per occupied voxel (bpov) is a metric normalized by occupied voxels. |
| Chamfer Distance (CD) | Generation, Reconstruction, Robustness | $\sum_{x \in S_1} \min_{y \in S_2} \|x - y\|_2^2 + \sum_{y \in S_2} \min_{x \in S_1} \|x - y\|_2^2$ | Measures the mean of the minimum squared distances between two point sets (prediction and ground truth). Lower is better. |
| Density-aware Chamfer Distance (DCD) | Generation | $\frac{1}{|X|} \sum_{x \in X} \frac{1 - \exp\left( -\frac{\|x - \pi_Y(x)\|_2^2}{\gamma} \right)}{(n_{\pi_Y(x)})^\tau} + \frac{1}{|Y|} \sum_{y \in Y} \frac{1 - \exp\left( -\frac{\|y - \pi_X(y)\|_2^2}{\gamma} \right)}{(n_{\pi_X(y)})^\tau}$ | A Chamfer-like distance penalizing density mismatch. |
| Earth Mover's Distance (EMD) | Generation, Reconstruction | $\min_{\phi: S_1 \to S_2} \sum_{x \in S_1} \|x - \phi(x)\|_2$ | Minimum sum of distances under a bijection. Lower is better. |
| Jensen–Shannon Divergence (JSD) | Generation, Reconstruction | $\frac{1}{2} D_{\text{KL}}(P \| M) + \frac{1}{2} D_{\text{KL}}(Q \| M), \quad M = \frac{1}{2}(P + Q)$ | Measures similarity between point distributions. Lower is better. |

**Table 16 Continued**

| Metric | Task | Formula | Description |
|---|---|---|---|
| Point-cloud Fréchet Inception Distance (P-FID) | Generation | $\|\mu_X - \mu_Y\|_2^2 + \mathrm{Tr}\left(\Sigma_X + \Sigma_Y - 2(\Sigma_X \Sigma_Y)^{1/2}\right)$ | FID-style distance on point-cloud embeddings. Lower is better. |
| Point-cloud Kernel Inception Distance (P-KID) | Generation | $\frac{1}{m(m-1)}\sum_{i\neq j} k(u_i, u_j) + \frac{1}{n(n-1)}\sum_{i\neq j} k(v_i, v_j) -$ $\frac{2}{mn}\sum_{i=1}^{m}\sum_{j=1}^{n} k(u_i, v_j), \quad u_i = f(x_i),\ v_j = f(y_j)$ | Kernel-based distribution similarity (MMD variant). Lower is better. |
| Fréchet Pointcloud Distance (FPD) | Generation | $\|m_P - m_Q\|_2^2 + \mathrm{Tr}\left(\Sigma_P + \Sigma_Q - 2(\Sigma_P \Sigma_Q)^{1/2}\right)$ | Computed on PointNet feature vectors from real vs generated sets. Lower is better. |
| Minimum Matching Distance (MMD) | Generation, Reconstruction | $\frac{1}{|Y|}\sum_{y\in Y} \min_{x\in X} d(x,y)$ | Measures average nearest-neighbor distance from real to generated. Lower is better. |
| Hausdorff Distance (HD) | Reconstruction, Robustness | $\max\{\sup_{x\in S_1} d(x, S_2),\ \sup_{y\in S_2} d(y, S_1)\}$ | Measures the maximum mismatch between two point sets. Lower is better. |
| Normal Consistency (NC) | Reconstruction | $\frac{1}{2|\partial \hat{M}|}\int_{\partial \hat{M}} |\langle n(p), n(\pi_2(p))\rangle|\ dp +$ $\frac{1}{2|\partial M|}\int_{\partial M} |\langle n(\pi_1(q)), n(q)\rangle|\ dq$ | Mean absolute dot product of corresponding normals. Higher is better. |
| Light Field Descriptor (LFD) | Reconstruction | $\min_i \sum_{k=1}^{10} d(I_{1k}, I_{2k})$ | Measuring visual similarity between 3D shapes based on rendered views. Lower is better. |
| Average Distance of Distinguishable Model Point (ADD) | 6DoF Pose estimation | $\frac{1}{|M|}\sum_{x\in M} \|(Rx + T) - (\hat{R}x + \hat{T})\|, \quad x \in M$ | Average distance between model points transformed by predicted and ground-truth pose. Lower is better. |
| Average Distance of Distinguishable Model Point - Symmetric (ADD-S) | 6DoF Pose estimation | $\frac{1}{|M|}\sum_{x_1\in M} \min_{x_2\in M} \left\|(Rx_1 + T) - (\hat{R}x_2 + \hat{T})\right\|$ | For symmetric objects: closest-point variant. Lower is better. |
| Attack Success Rate (ASR) | Robustness | - | Proportion of generated adversarial examples that successfully fool the model, considering only originally correctly classified samples. |
| Defense Accuracy | Robustness | - | Ratio of input samples correctly classified by the model, to the whole number of input samples in the test phase. |
| Corruption Error (CE) | Robustness | $\frac{1-\mathrm{OA}_i}{1-\mathrm{OA}_{\mathrm{baseline}}}$ | CE normalizes the error under corruption i by the baseline model's error. |
| Relative mean CE (RmCE) | Robustness | $\frac{\mathrm{mCE}_{\mathrm{model}}}{\mathrm{mCE}_{\mathrm{baseline}}}$ | Quantifies a model's degradation under noise corruptions relative to a baseline by averaging the ratio of error increases. |
| AUROC | Anomaly Detection | $\int R_{\mathrm{TP}}\ dR_{\mathrm{FP}}$ | AUROC evaluates the performance of a classification model for all thresholds. |
| AUPRO | Anomaly Detection | $\int \frac{1}{N}\sum_{i=1}^{N} \frac{|P\cap C_i|}{|C_i|}, \quad dR_{\mathrm{FP}}$ | AUPRO evaluates segmentation anomaly detection via per-region overlap vs false positive rate. |
| BLEU | Captioning | $\mathrm{BP}\cdot \exp\left(\frac{1}{4}\sum_{n=1}^{4} \log p_n\right)$ | BLEU (Bilingual Evaluation Understudy) compares generated captions to references using n-gram precision. |
| METEOR | Captioning | $(1 - \mathrm{Penalty})\cdot F_{\mathrm{mean}}$ | Metric for Evaluation of translation with explicit ordering. |
| CIDEr | Captioning | $\frac{1}{m}\sum_j \frac{\boldsymbol{g}^n(c_i)\cdot \boldsymbol{g}^n(s_{ij})}{\|\boldsymbol{g}^n(c_i)\|\,\|\boldsymbol{g}^n(s_{ij})\|}$ | Consensus-based Image Description Evaluation |

Table 17: Quantitative comparison of 3D object classification methods on the ModelNet10 and ModelNet40 benchmarks.

| Category | SubCategory | Method | Year | ModelNet40 (OA) | (mAcc) | ModelNet10 (OA) | (mAcc) |
|---|---|---|---|---|---|---|---|
| Projection-based | – | MVCNN (Su et al., 2015) | 2015 | 90.10 | – | – | – |
| | | GVCNN (Feng et al., 2018) | 2018 | 93.10 | 84.50 | – | – |
| | | MHBN (Yu et al., 2018b) | 2018 | 93.10 | **94.70** | 95.00 | 95.00 |
| | | MVTN (Hamdi et al., 2021) | 2021 | 93.50 | 92.20 | – | – |
| | | MVACPN (Wang et al., 2022c) | 2022 | 93.64 | 91.53 | – | – |
| | | DTV-CNN (Xia, 2023) | 2023 | 89.20 | – | 94.00 | – |
| | | CLIP2Point (Huang et al., 2023a) | 2023 | 94.20 | – | – | – |
| Volumetric-based | – | VoxNet (Maturana & Scherer, 2015) | 2015 | – | 83.00 | – | 92.00 |
| | | FPNN (Li et al., 2016) | 2016 | 88.40 | – | – | – |
| | | OctNet (Riegler et al., 2017) | 2017 | – | 85.50 | – | 91.00 |
| | | O-CNN (Wvoting) (Wang et al., 2017) | 2017 | 89.90 | – | – | – |
| | | Kd-Network (depth 15) (Klokov & Lempitsky, 2017) | 2017 | 91.80 | 88.50 | 94.00 | 93.50 |
| | | MRCNN (Ghadai et al., 2019) | 2019 | – | 86.20 | – | 91.30 |
| | | $(AF)^2$-S3Net (Cheng et al., 2021) | 2021 | 93.16 | – | – | – |
| | | PV-Ada (Zhu et al., 2022) | 2022 | 92.30 | – | – | – |
| Point-based | MLP-based | DeepSets (Zaheer et al., 2017) | 2017 | 82.00 | – | – | – |
| | | PointNet (Qi et al., 2017a) | 2017 | 89.20 | 86.20 | – | – |
| | | PointNet++ (Qi et al., 2017b) | 2017 | 90.70 | – | - | – |
| | | SRN (Duan et al., 2019b) | 2019 | 91.50 | – | – | – |
| | | PointWeb (Zhao et al., 2019) | 2019 | 92.30 | 89.40 | – | – |
| | | Mo-Net (Joseph-Rivlin et al., 2019) | 2019 | 92.40 | 90.30 | – | – |
| | | PointNeXt (Qian et al., 2022) | 2022 | 93.20 | 90.80 | – | – |
| | | PointMLP (Ma et al., 2022) | 2022 | 94.50 | 91.40 | – | – |
| | | PointGL (Li et al., 2024a) | 2024 | 93.00 | 90.40 | – | – |
| | | DualMLP (Paul et al., 2024) | 2024 | 93.70 | – | – | – |
| | | POINTMIL (CurveNet) (De Vries et al., 2025) | 2025 | 93.50 | 90.50 | – | – |
| | | Point-KAN (Shi et al., 2025) | 2025 | 93.70 | – | – | – |
| | Convolution-based | PointWiseCNN (Hua et al., 2018) | 2018 | 86.10 | 81.40 | – | – |
| | | PointCNN (Li et al., 2018) | 2018 | 92.20 | 88.10 | – | – |
| | | PCNN (Atzmon et al., 2018) | 2018 | 92.30 | – | 94.90 | – |
| | | A-CNN (Komarichev et al., 2019) | 2019 | 92.60 | 90.30 | 95.50 | **95.30** |
| | | KPConvrigid (Thomas et al., 2019) | 2019 | 92.90 | – | – | – |
| | | ShellNet (Zhang et al., 2019b) | 2019 | 93.10 | – | – | – |
| | | DensePoint (Liu et al., 2019c) | 2019 | 93.20 | – | **96.60** | – |
| | | RS-CNN (Liu et al., 2019d) | 2019 | 93.60 | – | – | – |
| | | ConvPoint (Boulch, 2020) | 2020 | 91.80 | 88.50 | – | – |
| | | DeltaConv (Wiersma et al., 2022) | 2022 | 93.80 | 91.20 | – | – |
| | | RepSurf (Ran et al., 2022) | 2022 | **94.70** | 91.70 | – | – |
| | | CompositeNets (Conv) (Floris et al., 2024) | 2024 | 91.30 | 87.10 | – | – |
| | | DC-CCNN (Dang et al., 2026) | 2026 | 93.50 | – | – | – |
| | Graph-based | ECC (Simonovsky & Komodakis, 2017) | 2017 | 87.40 | 83.20 | 90.80 | 90.00 |
| | | PointGCN (Zhang & Rabbat, 2018) | 2018 | 89.51 | 86.05 | 91.91 | 91.57 |
| | | RGCNN (Te et al., 2018) | 2018 | 90.50 | 87.30 | – | – |
| | | KCNet (Shen et al., 2018) | 2018 | 91.00 | – | 94.40 | – |
| | | SpecGCN (Wang et al., 2018a) | 2018 | 92.10 | – | – | – |
| | | DGCNN (Wang et al., 2019c) | 2019 | 93.50 | 90.70 | – | – |
| | | 3D-GCN (Lin et al., 2020) | 2020 | 92.10 | – | – | – |
| | | PointNGCNN (Lu et al., 2020) | 2020 | 92.80 | 89.90 | – | – |
| | | PointManifold (Yang & Gao, 2020) | 2020 | 93.00 | 90.10 | – | – |
| | | LDGCNN (Zhang et al., 2021a) | 2021 | 92.90 | 90.30 | – | – |
| | | PAConv (Xu et al., 2021b) | 2021 | 93.90 | – | – | – |
| | | CurveNet (Xiang et al., 2021a) | 2021 | 94.20 | – | 96.30 | – |
| | | PointView-GCN (Mohammadi et al., 2021) | 2021 | 95.40 | – | – | – |
| | | PointViG (Zheng et al., 2024b) | 2024 | 94.30 | 91.20 | – | – |
| | | Point-SkipNet (Saeid et al., 2025) | 2025 | 92.29 | 89.84 | – | – |
| | Transformer-based | PAT (Yang et al., 2019b) | 2019 | 91.70 | – | – | – |
| | | Point Transformer (Zhao et al., 2021a) | 2021 | 93.70 | 90.60 | – | – |
| | | PCT (Guo et al., 2021) | 2021 | 93.20 | – | – | – |
| | | PTv2 (Wu et al., 2022) | 2022 | 94.20 | – | – | – |
| | | LCPFormer (Huang et al., 2023b) | 2023 | 93.60 | 90.70 | – | – |
| | | PPT (Wang et al., 2025i) | 2025 | 93.90 | 91.20 | – | – |
| | | SPT (Q-SDE768) (Wu et al., 2025a) | 2025 | 91.22 | 88.45 | 94.76 | 93.69 |

## 7.3 Performance Analysis

This section analyzes the performance of representative deep learning methods for point cloud understanding, focusing on classification, segmentation, and object detection. These tasks are selected because they have widely used benchmark datasets and commonly reported evaluation metrics, making task-specific comparison more meaningful. The reported values are taken from the original papers rather than reproduced through re-training. Re-training and testing every method under a unified setting is often impractical due to differences in implementations, hyperparameter choices, training strategies, hardware requirements, and computational cost. Therefore, numerical results are compared only within compatible settings, such as the same task, dataset, and metric. Sections 7.1 and 7 further review the benchmark datasets and evaluation metrics commonly used in point cloud learning.

### 7.3.1 Classification

Point cloud classification aims to assign a semantic label to an entire 3D object or scene and has served as one of the most fundamental benchmarks for evaluating 3D deep learning models. Table 17 summarizes the performance of representative classification methods on the ModelNet40 and ModelNet10 datasets using overall accuracy (OA) and mean class accuracy (mAcc). The compared approaches are broadly categorized into projection-based, volumetric-based, and point-based methods, where point-based approaches are further divided into MLP-based, convolution-based, graph-based, and transformer-based architectures.

Early projection-based and volumetric approaches applied deep learning to 3D data by converting point clouds into image views or voxel grids. MVCNN (Su et al., 2015) achieved promising early results, but these representations often introduce information loss due to projection or discretization. As shown in Table 17, later projection-based models such as CLIP2Point (Huang et al., 2023a) improved performance on ModelNet40, reaching 94.2% OA.

Point-based methods later became dominant because they operate directly on raw point sets and better preserve geometric information. PointNet (Qi et al., 2017a) and PointNet++ (Qi et al., 2017b) established the foundation for this paradigm, and many subsequent architectures improved performance through better local feature learning. For example, RepSurf (Ran et al., 2022) achieves 94.7% OA on ModelNet40, while PointView-GCN (Mohammadi et al., 2021) reports 95.4% accuracy. Transformer-based models such as PTv2 (Wu et al., 2022) also achieve competitive results by capturing long-range dependencies.

### 7.3.2 Segmentation

3D point cloud segmentation aims to partition a three-dimensional environment by assigning a semantic label or instance identifier to every individual point, serving as a critical prerequisite for spatial understanding in navigation, robotics, and augmented reality. To evaluate these capabilities, various models are benchmarked across diverse datasets, such as ScanNet200 (Rozenberszki et al., 2022) and S3DIS (Armeni et al., 2016), utilizing metrics including mean Intersection over Union (mIoU) and Average Precision (AP). The compared approaches are broadly categorized into fully supervised baselines, weakly supervised methods, and unsupervised domain adaptation frameworks, where recent advancements are further driven by the integration of multi-modal foundation models and open-vocabulary architectures.

The benchmarks in Table 18 indicate a major shift in 3D scene understanding toward open-vocabulary generalization, multi-modal integration, and cross-domain robustness. State-of-the-art models are increasingly leveraging 2D foundation models and vision-language priors—such as SAS (Li et al., 2025p) integrating 2D priors—to achieve strong zero-shot, few-shot, and class-agnostic segmentation on highly complex, long-tail datasets like ScanNet200. Additionally, there is a distinct advancement in unsupervised domain adaptation; frameworks like CACE (Chen et al., 2025b), UniDxMD (Liang et al., 2025c), and D3CTTA (Zhao et al., 2025a) successfully bridge severe distribution shifts from synthetic to real-world environments or across adverse weather conditions without requiring target-domain annotations.

### 7.3.3 Object Detection

3D object detection has been extensively evaluated on benchmarks such as KITTI (Geiger et al., 2012), nuScenes (Caesar et al., 2020), and Waymo Open Dataset (Sun et al., 2020b). While earlier works primarily reported results on KITTI, recent methods increasingly focus on nuScenes and Waymo due to their larger scale, richer annotations, and more diverse driving scenarios. Therefore, in this work, we present comparisons on nuScenes and Waymo to align with contemporary literature.

Since detection performance varies significantly depending on the input modality, we categorize methods into camera-based, LiDAR-based, and multi-modal fusion approaches to enable a more meaningful comparison. As shown in Table 19, the reported results are collected from the original papers, using the test set of nuScenes and the validation set of Waymo (car category), which is the common evaluation protocol adopted by most prior works for fair comparison. Performance is evaluated using standard metrics, including mean Average Precision (mAP) and nuScenes Detection Score (NDS) for nuScenes, and Level 1/Level 2 mAP and mAPH for Waymo, where mAPH additionally accounts for heading accuracy.

Table 18: Segmentation performance comparison. Metric variants are reported per model.(Certain methods were excluded from the table because their unique evaluation criteria prevent direct comparison.)

| Category | Model | Metric | S3DIS | ScanNet | ScanNet200 | nuScenes | Evaluation Context / Task Focus |
|---|---|---|---|---|---|---|---|
| Instance Segmentation | Any3DIS (Nguyen et al., 2025b) | AP | – | – | 25.8 | – | Open-Vocabulary generalization to novel classes. |
| | Any3DIS (Nguyen et al., 2025b) | AP | – | – | 19.1 | – | Open-Ended without predefined vocabularies. |
| | Sketchy-3DIS (Deng et al., 2025b) | $AP_{50}$ | 64.6 | 65.8 | – | – | Weakly supervised using S1 "sketchy" boxes. |
| | SAM2Object (Zhao et al., 2025b) | mAP | – | 34.0 | 13.3 | – | Zero-shot, training-free class-agnostic instance seg. |
| Method Based | 3DVLM Pseudo-labeling (An et al., 2025b) | hIoU | – | 61.9 | 43.1 | – | 5-shot generalized few-shot vision-language. |
| | HIPO (Sur et al., 2025) | mIoU | 31.1 | 11.7 | – | – | Incremental learning / continual adaptation. |
| | SAS (Li et al., 2025p) | mIoU | – | 61.9 | – | 47.5 | Zero-shot segmentation with integrated 2D priors. |
| | Mosaic3D (Lee et al., 2025b) | f-mIoU | – | 84.4 | 28.3 | – | Annotation-free / zero-shot, trained on Mosaic3D-5.6M. |
| | BFANet (Zhao et al., 2025d) | mIoU | – | 78.0 | 37.3 | – | Fully supervised semantic segmentation. |
| | 3D-AVS (Wei et al., 2025b) | mIoU | – | 40.5 | 14.6 | 36.2 | Standard semantic segmentation. |
| | CDSegNet (Qu et al., 2025b) | mIoU | – | 77.9 | 36.3 | 81.2 | Single-step inference conditional network. |
| | GHEA (Zhang et al., 2025e) | mIoU | 79.5 | 76.6 | – | – | Generative hard example augmentation on baselines. |
| | PointNet-KAN-MLP (Kashefi, 2025) | mIoU | 51.5 | – | – | – | 3D Semantic Segmentation. |
| | LogoSP (Zhang et al., 2025j) | mIoU | 46.5 | 35.8 | – | 20.1 | Cross-dataset generalization. |
| | D3CTTA (Zhao et al., 2025a) | mIoU | – | – | – | 24.1 | Online continual test-time adaptation across corruptions. |
| | FastAdapter (Sun & Yan, 2025) | mIoU | 78.0 | 78.2 | – | – | Fast downsampling adapter on PTV3 backbone. |
| | Alpha-CLIP (Jung et al., 2025b) | mAP | 22.6 | – | 25.8 | – | Top-1 evaluation protocol (2D+3D consensus). |
| Cross-modality | VDG-Uni3DSeg (Han et al., 2025) | mAP | 63.7 | 59.3 | 29.5 | – | Cross-modal unified 3D segmentation |
| | VDG-Uni3DSeg (Han et al., 2025) | mIoU | 73.2 | 71.5 | 29.7 | – | Fully supervised semantic/panoptic segmentation. |
| | AiDe (Wang & Czarnecki, 2025) | hIoU | 42.2 | 72.8 | – | 62.2 | Open-vocabulary (hIoU of base/novel splits). |
| | GFS3DSeg (Li et al., 2025o) | hIoU | 56.04 | – | – | – | 1-shot generalized few-shot segmentation. |
| | UniDxMD (Liang et al., 2025c) | mIoU | – | – | – | 74.3 | Cross-modal unsupervised domain adaptation. |
| | CACE (Chen et al., 2025b) | mIoU | 57.1 | 58.2 | – | – | Sim-to-real unsupervised domain adaptation. |
| | MM-FSS (An et al., 2024) | mIoU | 54.2 | 50.1 | – | – | 1-way 5-shot few-shot segmentation. |

From the results, camera-based methods generally achieve lower performance compared to LiDAR-based approaches, primarily due to the lack of accurate depth information. LiDAR-based methods demonstrate strong and consistent performance, benefiting from precise 3D geometric cues, with voxel- and point-based architectures achieving competitive results across both datasets. Fusion-based methods further improve performance by leveraging complementary information from both camera and LiDAR modalities, often achieving state-of-the-art results. However, these gains come at the cost of increased model complexity and computational requirements. Overall, the comparison highlights the trade-off between accuracy and sensor dependency, with multi-modal fusion providing the best performance when multiple sensors are available.

# 8 Overall Conclusion, Open Challenges and Future Directions

This section summarizes the main insights of this survey, discuss the remaining challenges in deep point cloud learning, and outline future research directions. The conclusion first highlights the overall scope and organization of the survey. The challenges section then identifies common limitations that appear across different tasks, representations, and architectures. Finally, the future directions section discusses promising paths toward more robust, efficient, generalizable, and trustworthy 3D point cloud systems.

## 8.1 Overall Conclusion

This survey reviewed deep point cloud models across a broad range of 3D tasks, with particular emphasis on how representation choices and architectural paradigms shape model design and performance. The literature was organized into coherent task families, including core tasks, geometric modeling, alignment and pose estimation, foundation models and scene understanding, and robustness. The survey also discussed benchmark datasets, evaluation metrics, and performance analysis with emphasis on comparable task-specific settings.

## 8.2 Overall Challenges

Despite this progress, several general challenges remain unresolved. A major limitation is robustness under real-world sensing conditions. Many models achieve strong performance on clean or synthetic benchmarks but degrade under noise, occlusion, sparsity, density variation, clutter, missing regions, and motion distortion. This issue affects nearly all 3D tasks. In addition, generalization across domains remains a significant challenge. Point cloud data can differ substantially across sensors, environments, viewpoints, sampling densities, and acquisition settings, often causing performance degradation when models are deployed outside their training distribution. Differences between synthetic and real-world data, as well as between indoor RGB-D scans, outdoor LiDAR data, and industrial sensing systems, continue to pose challenges for reliable deployment.

Another major challenge is the trade-off between representation power and efficiency. Transformer-based, graph-based, diffusion-based, and multimodal foundation models often provide stronger reasoning or generation capabilities, but they also introduce high computational cost, memory consumption, and latency. This is problematic where real-time processing is required.

A further limitation is the lack of realistic and standardized evaluation protocols. Many benchmarks focus on specific tasks, clean datasets, or simplified metrics that do not fully reflect real-world performance. Common metrics may overlook factors such as robustness, uncertainty, spatial reasoning, and physical consistency. In addition, fragmented benchmarks make fair comparison across tasks, datasets, sensors, and methods difficult.

Security and trustworthiness are also becoming increasingly important. As 3D models are deployed in safety-critical applications, they become vulnerable to adversarial perturbations, sensor spoofing, poisoning, backdoor attacks, model extraction, and cross-modal attacks. Existing robustness and security studies are often limited to specific datasets, architectures, or attack assumptions, while physically realizable and system-level threats remain underexplored. Reliable deployment therefore requires robustness, interpretability, uncertainty estimation, provenance verification, and secure training and inference pipelines.

### 8.3 Overall Future Directions

Future research should move toward robust, efficient, generalizable, and trustworthy 3D intelligence. A key direction is developing models that remain reliable under noise, sparsity, occlusion, and domain shifts through realistic training conditions, adaptation techniques, and multimodal fusion of geometry, vision, language. Another important direction is the design of unified and lightweight architectures. Efficient transformers, sparse computation, distillation, and parameter-efficient adaptation can help bridge the gap between powerful 3D foundation models and real-time deployment while supporting multiple 3D tasks within a single framework.

The field also needs better evaluation frameworks. Future benchmarks should include realistic noise, occlusions, dynamic scenes, domain shifts, adversarial settings, and open-world scenarios. Evaluation should go beyond task-specific metrics to measure robustness, uncertainty, efficiency, and real-world performance. Standardized benchmarks and protocols would enable fairer comparisons across methods.

Finally, future research should move toward open-world and embodied 3D understanding. Next-generation models must reason about unseen objects, changing environments, physical interactions, and human instructions while remaining robust and efficient. Advances in foundation models, multimodal learning, self-supervised training, and secure deployment will be key to building reliable and scalable 3D perception systems.

Table 19: Quantitative comparison of representative 3D object detection methods on the nuScenes (test set) and Waymo (validation set) benchmarks. Methods are grouped by modality (camera, LiDAR, and fusion).

| Method | Classification | nuScenes | | Waymo car | | | |
|---|---|---|---|---|---|---|---|
| | | mAP | NDS | L1 mAP | L1 mAPH | L2 mAP | L2 mAPH |
| CenterNet (Duan et al., 2019a) | Camera-Monocular | 33.80 | 40.00 | – | – | – | – |
| DETR3D (Wang et al., 2022e) | Camera-Multi-View | 41.20 | 47.90 | – | – | – | – |
| PolarFormer (Jiang et al., 2023b) | Camera-Multi-Camera | 49.30 | 57.20 | – | – | – | – |
| BEVDepth (Li et al., 2023j) | Camera-Depth | 52.00 | 60.90 | – | – | – | – |
| CAPE (Xiong et al., 2023) | Camera-Multi-View | **52.50** | **61.00** | – | – | – | – |
| PV-RCNN (Shi et al., 2020a) | Lidar-Point-Voxel | – | – | 77.51 | 76.89 | 68.98 | 68.41 |
| PV-RCNN++ (Shi et al., 2023b) | Lidar-Point-Voxel | – | – | 80.17 | 79.70 | 72.17 | 71.70 |
| PVTransformer (Leng et al., 2024) | Lidar-Point-Voxel | – | – | 82.20 | 81.70 | 74.30 | 73.90 |
| RSN (Sun et al., 2021b) | Lidar-Range Image | – | – | 78.40 | 78.10 | 69.50 | 69.10 |
| PillarNet (Shi et al., 2022a) | Lidar-Pillars | 66.00 | 71.40 | 79.09 | 78.59 | 70.92 | 70.46 |
| BiProDet (Zhang et al., 2023f) | Lidar-BEV | – | – | 78.36 | 77.91 | 69.45 | 69.04 |
| HEDNet (Zhang et al., 2023c) | Lidar-BEV | 67.70 | 72.00 | 81.10 | 80.60 | 73.20 | 72.70 |
| MGTANet (Koh et al., 2023) | Lidar-BEV | 67.50 | 72.70 | – | – | – | – |
| BEVDilation (Zhang et al., 2026a) | Lidar-BEV | **75.40** | 73.10 | – | – | – | |
| DetZero (Ma et al., 2023) | Lidar-Point | – | – | **83.07** | **82.57** | **75.72** | **75.24** |
| Li3DeTr (Erabati & Araujo, 2023) | Lidar-Point | 61.30 | 67.60 | – | – | – | – |
| SEED (Liu et al., 2024h) | Lidar-Point | – | – | 79.80 | 79.30 | 71.90 | 71.50 |
| Lion (Liu et al., 2024g) | Lidar-Point | 69.80 | **79.30** | 80.30 | 79.90 | 72.00 | 71.60 |
| GeoFormer (Jin et al., 2025b) | Lidar-Point | 69.80 | 73.70 | 80.48 | 79.98 | 72.21 | 71.75 |
| CneterPoint (Yin et al., 2021) | Lidar-Voxel | – | – | 76.70 | 76.20 | 68.80 | 68.30 |
| SST (Fan et al., 2022b) | Lidar-Voxel | – | – | 77.04 | 76.56 | 68.50 | 68.08 |
| UVTR (Li et al., 2022c) | Lidar-Voxel | 63.90 | 69.70 | – | – | – | |
| FocalConvs (Chen et al., 2022e) | Lidar-Voxel | 70.10 | 73.60 | – | – | – | – |
| MSSVT++ (Li et al., 2023b) | Lidar-Voxel | – | – | 79.96 | 79.43 | 71.30 | 70.86 |
| LargeKernel3D (Chen et al., 2023e) | Lidar-Voxel | 65.40 | 70.60 | 78.07 | 77.61 | 69.81 | 69.38 |
| VoxelNetXt (Chen et al., 2023f) | Lidar-Voxel | 66.20 | 71.40 | 78.20 | 77.70 | 69.90 | 69.40 |
| OCTR (Zhou et al., 2023a) | Lidar-Voxel | – | – | 78.12 | 77.63 | 69.79 | 69.34 |
| DSVT (Wang et al., 2023a) | Lidar-Voxel | 68.40 | 72.70 | 82.10 | 81.60 | 74.50 | 74.10 |
| VoxelMamba (Zhang et al., 2024d) | Lidar-Voxel | 69.00 | 73.00 | 80.80 | 80.30 | 72.60 | 72.20 |
| FSHNet (Liu et al., 2025e) | Lidar-Voxel | 71.70 | 68.10 | 82.20 | 81.70 | 74.50 | 74.00 |
| DSTR (Cai et al., 2025) | Lidar-Voxel | 72.30 | 74.40 | – | – | – | – |
| WinMamba (Zheng et al., 2026b) | Lidar-Voxel | – | – | 78.00 | 77.50 | 69.60 | 69.10 |
| MapFusion (Fang et al., 2021) | Fusion (multi-view + point cloud) | 60.61 | 67.97 | – | – | – | – |
| PointAugmenting (Wang et al., 2021a) | Fusion (monocular + point cloud) | 66.80 | 71.00 | 67.41 | – | 62.70 | – |
| EPNet++ (Liu et al., 2022) | Fusion (monocular + point cloud) | – | – | 76.57 | 76.10 | 68.29 | 67.86 |
| MSMDFusion (Jiao et al., 2023) | Fusion (multi-view + point cloud) | 71.50 | 74.00 | – | – | – | – |
| LoGoNet (Li et al., 2023i) | Fusion (multi-view + point cloud) | – | – | **83.21** | **82.72** | **75.84** | **75.38** |
| SAFDNet (Zhang et al., 2024c) | Fusion (multi-view + point cloud) | 68.30 | 72.30 | 80.60 | 80.10 | 72.70 | 72.30 |
| IS-Fusion (Yin et al., 2024) | Fusion (multi-view + point cloud) | **73.00** | 75.20 | – | – | – | – |
| Wang et al. (2025d) | Fusio (multi-view + point cloud) | 72.70 | **75.90** | – | – | – | – |
| UniMamba (Jin et al., 2025a) | Fusion (multi-view + point cloud) | 70.20 | 74.00 | 80.60 | 80.06 | 72.28 | 71.77 |

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
