# OpenReview forum: "A Comprehensive Survey on 3D Deep Point Cloud Models"
_TMLR — Under review for TMLR_

### Review · Reviewer_gMLF · 2026-06-30

**Summary Of Contributions:**

This manuscript provides a detailed summary of current deep learning models for 3D point clouds. It organizes the literature along two complementary axes: (1) point cloud representations, and (2) architectural paradigms, and attempts to characterize how "representation × architecture" combinations shape model behavior across different tasks. The paper also organizes point cloud learning into several task families: core tasks, geometric modeling and pose, foundation models and scene understanding, and robustness, generalization, and reliability. It further provides a summary of datasets and evaluation metrics, presents a performance comparison and analysis for three tasks (classification, segmentation, and detection), and finally outlines open challenges and future directions.

**Main strengths**

- The breadth of coverage is notable and the work is highly up-to-date, incorporating a large body of recent works, including foundation models and VLM/embodied directions. Compared with existing surveys, it integrates directions that are less commonly covered together—such as "robustness and security" and "foundation models and scene understanding"—within a single framework.
- The dual-axis "representation × architecture" organizing perspective offers some novelty, and Table 1 provides a clear comparison with existing surveys.
- The systematic summaries of datasets (Table 15) and metrics with formulas (Table 16) provide important reference value for readers seeking to understand the point cloud learning field.

**Main weaknesses**

- Despite the large length, some subsections lean toward "taxonomic enumeration / list-like" presentation, lacking critical synthesis and cross-method insight.
- The "representation–architecture interaction" analysis, which is positioned as the core selling point, is notably thin—essentially one paragraph plus a single table—and does not deliver on the "systematic analysis of interactions" promised in the introduction.
- Some tables mix results obtained under different supervision settings / datasets / protocols, which can easily mislead readers.
- Some recent references are missing.

**Audience:**

Yes

**Audience Explanation:**

3D point clouds are a core data modality for 3D vision, robotics, autonomous driving, and AR/VR, with a broad audience. A substantial portion of the TMLR readership would be interested.

**Broader Impact Concerns:**

The current manuscript does not include a Broader Impact statement. We recommend that the authors add a brief Broader Impact / ethics and limitations statement, addressing at least: (i) the consequences of point cloud perception failures in safety-critical scenarios; (ii) dual-use considerations regarding the attack/adversarial techniques discussed in the survey; and (iii) privacy considerations (e.g., 3D scanning of human bodies / scenes).

**Claims And Evidence:**

Yes

**Claims Explanation:**

The descriptive claims of the paper are generally accurate and clearly organized. However, the depth of analysis and the accuracy of the taxonomy weaken its persuasiveness, and we recommend that the authors revise accordingly.

**Requested Changes:**

1. Provide a "representation × architecture" grid, specifying for each cell the typical combinations, their advantages / failure modes, and on which tasks they empirically prevail, supporting this as far as possible with the performance data already summarized in the paper.

2. Add a Broader Impact / limitations statement.

3. At least one typo has been found (Section 7.1, "bumber of classes" should be "number"); a thorough proofreading of the entire manuscript is recommended.

4. The figure captions are overly brief; please provide complete, self-contained captions.

5. Some additional point cloud tasks are missing. For example:
   - **3D Gaussian Splatting tasks:**
     - A survey on 3D Gaussian Splatting applications: Segmentation, editing, and generation.
     - DiffStyle3D: Consistent 3D Gaussian Stylization via Attention Optimization.
     - FantasyStyle: Controllable stylized distillation for 3D Gaussian splatting.
     - ReferSplat: Referring segmentation in 3D Gaussian splatting.
   - **Parameter-efficient fine-tuning for point clouds:**
     - Parameter-efficient fine-tuning in spectral domain for point cloud learning.
   - **Few-shot point cloud semantic segmentation:**
     - Reasoning beyond points: A visual introspective approach for few-shot 3D segmentation.
     - Biologically-inspired evolutionary domain symbiosis for few-shot and zero-shot point cloud semantic segmentation.
     - DyPolySeg: Taylor Series-Inspired Dynamic Polynomial Fitting Network for Few-shot Point Cloud Semantic Segmentation.
     - Taylor series-inspired local structure fitting network for few-shot point cloud semantic segmentation.

---

### Review · Reviewer_otWr · 2026-07-09

**Summary Of Contributions:**

Summary:
This paper surveys deep learning methods for 3D point clouds, covering representations, architectural paradigms, core tasks, geometric modeling, foundation models, scene understanding, robustness, datasets, metrics, and performance comparisons. The paper aims to organize the field through representation-architecture interactions and broad task families rather than focusing on a single task or model type.

Strengths:
1) The paper has broad coverage, spanning classification, segmentation, detection, tracking, compression, generation, reconstruction, registration, pose estimation, foundation models, scene understanding, and robustness.
2) The topic is timely, especially with the inclusion of recent trends such as 3D foundation models, open-vocabulary 3D understanding, 3D reasoning, security threats, and robustness.
3) The “representation + architecture” organization is reasonable and has more potential than a purely task-by-task listing.
4) The sections on datasets, metrics, and performance analysis help readers quickly understand commonly used benchmarks.

Weaknesses:
1) The manuscript would benefit from stronger editorial unification and a cleaner taxonomy. Several sections appear to be written in different styles, with inconsistent levels of technical depth and claim calibration across tasks. More importantly, the central thread is not always clear, as the manuscript repeatedly mixes representation, architecture, training paradigm, modality, and application domain. For example, foundation models are sometimes treated as representations, sometimes as architectures, and sometimes as reasoning modules.
2) Many paragraphs merely list recent works one by one, without sufficient abstraction, comparison across methods, or a clear account of the key technical developments.
3) Section 2.1 is titled “Point Cloud Representations,” but the main text only discusses point-based and voxel-based representations. Table 2 later includes projection-based, Pillar/BEV, graph-based, token/foundation, and hybrid fusion representations. This makes the taxonomy appear incomplete, and the table feels as if it appears abruptly.
4) The paper also lacks discussion of recent work on parameter-efficient fine-tuning for point clouds, such as IDPT [1] and DAPT [2]. These methods adapt foundation models through parameter-efficient fine-tuning and are highly useful for practical point cloud applications.

[1] Zha, Yaohua, et al. "Instance-aware dynamic prompt tuning for pre-trained point cloud models." Proceedings of the IEEE/CVF International Conference on Computer Vision. 2023.
[2] Zhou, Xin, et al. "Dynamic adapter meets prompt tuning: Parameter-efficient transfer learning for point cloud analysis." Proceedings of the IEEE/CVF Conference on Computer Vision and Pattern Recognition. 2024.

**Audience:**

Yes

**Audience Explanation:**

The paper covers a broad and timely area in 3D point cloud learning, including foundation models, scene understanding, and robustness, which would be relevant to researchers working on 3D vision and multimodal learning.

**Claims And Evidence:**

Yes

**Claims Explanation:**

Most of the major claims are supported by relevant citations and broad literature coverage. The paper discusses a wide range of tasks, representations, architectures, datasets, metrics, and recent trends, and the claims are generally grounded in prior work.

**Requested Changes:**

1) Unify the taxonomy and clearly separate representations, architectures, training paradigms, modalities, and application domains.
Strengthen the central narrative around the proposed “representation + architecture” perspective.
Revise sections with inconsistent writing style, technical depth, and claim strength to improve editorial coherence.

2) Expand Section 2.1 to cover all representation types later used in Table 2, including projection-based, Pillar/BEV, graph-based, tokenized, and hybrid representations.

3) Add discussion of recent parameter-efficient fine-tuning methods for point clouds, such as IDPT and DAPT.

---

### Review · Reviewer_D45k · 2026-07-11

**Summary Of Contributions:**

This paper provides a broad survey of deep learning methods for 3D point clouds. It organizes the literature from both representation and architecture perspectives and covers classification, segmentation, detection, tracking, compression, generation, reconstruction, registration, pose estimation, foundation models, scene understanding, and robustness. It also summarizes commonly used datasets, evaluation metrics, and benchmark results

Strengths:
The paper has broad coverage and includes several topics that are often missing from earlier surveys, especially robustness, security, foundation models, and language-based 3D scene understanding (Secs. 5–6; Table 1).
The representation–architecture taxonomy is useful. Connecting point-based, voxel-based, projection-based, graph-based, token-based, and hybrid representations with different network families gives readers a clear overview of the field (Sec. 2; Table 2).
The dataset and benchmark summaries are also useful as reference material, and the authors generally avoid directly comparing results obtained under incompatible settings (Sec. 7.3).

Weaknesses:
The main weakness is that the paper is very broad but not always sufficiently analytical. Many sections mainly list recent papers with short descriptions, without clearly comparing their assumptions, computational costs, failure cases, or practical trade-offs (e.g., Secs. 3.2 and 5.1). As a result, parts of the manuscript read more like an extended bibliography than a critical survey.
The discussion of point-cloud distance metrics is particularly shallow. Chamfer Distance is mainly introduced through its definition, while important issues such as sensitivity to outliers, uneven point density, and inaccurate local correspondences are not discussed in enough detail (Sec. 7.2; Table 16). The paper should also distinguish more clearly between using Chamfer Distance as a training loss and using it as an evaluation metric. Newer alternatives, such as HyperCD, InfoCD, and GPS, should be introduced and compared with CD, DCD, and EMD.
The survey methodology is unclear. No direct evidence was found in the manuscript describing the databases searched, keywords, inclusion criteria, screening procedure, or literature cutoff date. This makes it difficult to assess whether the coverage is systematic and balanced.
The manuscript also needs substantial proofreading. There are frequent grammatical issues, inconsistent terminology, and several awkward or broken cross-references throughout the paper.

**Audience:**

Yes

**Audience Explanation:**

Point cloud is one of the major data format in 3D vision, certainly a good survey is always needed

**Claims And Evidence:**

Yes

**Claims Explanation:**

See Summary Of Contributions and Requested Changes

**Requested Changes:**

question:
How were the large number of recent 2025–2026 papers and their results verified?
Clarifying the verification process would help assess the reliability of the benchmark tables.

Can the authors substantially expand the discussion of point-cloud distance metrics?
In particular, it would be useful to compare CD, DCD, EMD, HyperCD, InfoCD, and GPS in terms of robustness, computational cost, density sensitivity, and suitability for training and evaluation. This revision would address one of my main concerns.

Requested Changes:
Add a short survey-methodology subsection describing the literature search, selection criteria, coverage period, and verification process.
Expand Sec. 7.2 with a dedicated comparison of point-set distances. The comparison should cover computational complexity, differentiability, sensitivity to outliers, sensitivity to density variation, correspondence quality, and typical use cases. HyperCD, InfoCD, and GPS should be included alongside CD, DCD, and EMD.
Reduce long lists of methods and add more comparative tables or synthesis, especially in the segmentation, foundation-model, and scene-understanding sections.
Perform a full language, terminology, and cross-reference check before publication.